# Edge curvature drives endoplasmic reticulum reorganization and dictates epithelial migration mode

Simran Rawal[1], Pradeep Keshavanarayana[2,3], Diya Manoj[1], Purnati Khuntia [1], Sanak Banerjee[1], Basil Thurakkal[1], Rituraj Marwaha[1], Fabian Spill [2]✉ & Tamal Das [1]✉

From single-cell extrusion to centimetre-sized wounds, epithelial gaps of various sizes and geometries appear across organisms. Their closure involves two orthogonal modes: lamellipodial crawling at convex edges and purse string-like movements at concave edges. The mechanisms driving this curvature-dependent migration remain unclear. Here we perform an intracellular cartography to reveal that in both micropatterned and naturally arising gaps, the endoplasmic reticulum (ER) undergoes edge curvature-dependent morphological reorganizations, forming tubules at convex edges and sheets at concave edges. This reorganization depends on cytoskeleton-generated protrusive and contractile forces. Mathematical modelling reveals that these morphologies minimize strain energy under their respective geometric regime. Functionally, ER tubules at the convex edge favour perpendicularly oriented focal adhesions, supporting lamellipodial crawling, while ER sheets at the concave edge favour parallelly oriented focal adhesions, supporting purse string-like movements. Altogether, ER emerges as a central mechanotransducer, integrating signals from cytoskeletal networks to orchestrate two orthogonal modes of cell migration.

Epithelial gap closure is essential for maintaining tissue integrity as disruptions occur at various scales, from individual cell loss to large wounds, throughout the life of an organism[1–6]. The epithelial gap closure happens by a collective movement of the surrounding cells[7], which involves two orthogonal migration modes: lamellipodia-mediated cell crawling and actomyosin contractility-driven purse string-like movements[8–12]. These modes involve different actin and focal adhesion dynamics and assembly: branched actin polymerization and perpendicularly oriented focal adhesions in lamellipodia-mediated cell crawling versus actin bundle formation and parallelly oriented focal adhesions in purse string-like movements. Importantly, their relative contribution to epithelial gap closure depends on various biochemical and biophysical aspects of the

tissue, including the curvature of the gap edge[6]. While previous studies have identified several molecules that sense or produce nanoscale membrane curvatures[13,14], how cells coordinate their response towards curvature at cellular and multicellular length scales remains largely elusive. Nevertheless, epithelial cells preferentially use lamellipodial crawling at convex edges and purse string contraction at concave edges[15]. Also, a previous study showed that curvature-dependent migration is linked to actin flow, with anterograde flow at concave edges and retrograde flow at convex edges during lamellipodia formation[16]. However, it remains unknown how other intracellular structures organize in response to large-scale curvature, how such organization is regulated and whether this organization influences the mode of cell migration.

[1]Tata Institute of Fundamental Research Hyderabad, Hyderabad, India. [2]School of Mathematics, University of Birmingham, Edgbaston, UK. [3]Present address: Centre for Computational Medicine, University College London, London, UK. ✉e-mail: f.spill@bham.ac.uk; tdas@tifrh.res.in

Here, we performed a cartography of intracellular structures in convex and concave curved regions of micropatterned wounds of defined geometry in cultured epithelial monolayer and natural wounds in mouse embryonic epidermis. This revealed that the endoplasmic reticulum (ER) undergoes striking curvature-dependent morphological rearrangements. The ER is a continuous and highly dynamic organelle that spans the whole cell as a single entity. It comprises rounded tubular networks and dense flat sheets, which undergo constant rearrangements between themselves[17–23]. This highly dynamic nature of the ER reflects its adaptability and responsiveness to intracellular and extracellular changes[24–26]. However, the relationship between different ER morphologies and physical or geometric cues remains largely unknown. Here, we report that the endoplasmic reticulum alters its morphology and dynamics in response to edge curvature and plays a critical role in determining the modes of epithelial migration at different edge curvatures.

## Results

### Edge curvature-dependent ER morphology in patterned gaps

First, to reveal the cell biological mechanism underlying the differential response of epithelial cells to edge curvature, we used microfabrication[8,15,16] for the controlled and reproducible generation of gaps with defined shape, size and curvature. We fabricated polydimethylsiloxane (PDMS) stencils of two shapes—notched rectangles and flowers (Extended Data Fig. 1a,b)—consisting of convex, concave and flat edges[15]. We seeded Madin–Darby canine kidney (MDCK) epithelial cells around these PDMS stencils (Fig. 1a) for 15–18 h, allowed them to form a confluent monolayer and then lifted off the stencils triggering the cells to start migrating into the gaps. MDCK cells closed the gaps within 6–8 h (Extended Data Fig. 1a,b and Supplementary Videos 1 and 2). Considering that the lamellipodial protrusions or actin bundles emerge within 15–30 min (ref. [15]) (Extended Data Fig. 1c and Supplementary Videos 3 and 4), we fixed the cells 30 min after removal of the stencil and stained them with fluorescent dye-labelled phalloidin to visualize the organization of the actin cytoskeleton. As expected[15], we observed that the fraction of cells forming lamellipodia at the positively curved convex edge was significantly higher than that at the negatively curved concave edge (Fig. 1b, top, and Extended Data Fig. 1d). We next studied the arrangement of the other major cytoskeletal element, the microtubules. Microtubule filaments appeared more discrete and aligned perpendicular to the edge at the convex edge and formed thick bundles aligned parallel to the edge at the concave edge (Fig. 1b, middle). Quantification of the microtubule directionality further confirmed this conclusion (Extended Data Fig. 1e,f). To ensure the accurate analysis of curvature-dependent behaviour, we quantified curvature at both the single-cell and tissue level, fitting circles to cell edges and splines

to wound boundaries. We included only cells with more than 20% of their perimeter in contact with the wound, as lower-exposure cells were more influenced by neighbouring constraints than the geometry itself. This analysis revealed a strong one-to-one correspondence between the two curvature scales, thus validating our approach for linking local cell behaviour to global geometric cues (Supplementary Note 1 and Extended Data Fig. 2a–h).

Next, to investigate whether other intracellular entities also respond to this large-scale geometrical cue, we performed a cartographic study of cellular organelles at differently curved edges. We used anti-GRASP65 antibody, anti-LAMP1 antibody and mitotracker green to visualize the Golgi apparatus, lysosomes and mitochondria, respectively. In addition, we transfected MDCK cells with mApple-Sec61β to observe the ER structure. The Golgi showed a ribbon-like structure at both curvatures, but it was mostly polarized in the front of the nucleus at the convex edge and next to the nucleus along its horizontal axis at the concave edge (Extended Data Fig. 3a). Lysosomes appeared closely localized near the actin bundle at the concave edge, but more dispersed in the lamellipodia at the convex edge (Extended Data Fig. 3a). Mitochondria had filamentous morphology spread throughout the cell and showed no apparent difference in structure or organization depending on the edge curvature (Extended Data Fig. 3a). Interestingly, ER morphology appeared remarkably different within the cells located at convex and concave edges. We observed the formation of a reticular network of ER tubules at the convex edge and a dense sheet-like lamellar morphology at the concave edge (Fig. 1b, bottom). We further quantified cytoskeletal and organelle distribution at different edges using a probability density-based approach to compute the mean distribution from the edge (MDE), which measures the distance between the cell edge and the centre of mass of each intracellular component[27] (Extended Data Fig. 3b). A lower value of the MDE suggests that the organelle or the cytoskeleton is polarized and densely accumulated near the migrating edge. We observed that the MDE for actin, microtubules, ER, Golgi apparatus and lysosomes at the concave edge is significantly lower than the respective MDE at the convex edge, suggesting that they are differentially polarized in response to the geometrical cue (Fig. 1c and Extended Data Fig. 3c). Taken together, of all the organelles, ER showed differences not just in the distribution but also in its morphology and structure at different edge curvatures.

We next quantified the structural changes in ER by calculating the fraction of ER tubules as a function of curvature at the front of the cell by segmenting the ER into tubules and sheet-like dense regions using Trainable Weka Segmentation[28]. This analysis revealed that at convex edges, the ER tubule fraction at the cell front increases sharply with increasing positive curvature (Fig. 1d). In contrast, at concave edges, the ER tubule fraction remains consistently low and shows

**Fig. 1 | Edge curvature-dependent ER morphologies. a**, A schematic representing the experimental set-up for creating gaps of definite geometries. Cells surround the PDMS stencils placed on glass and after their removal the cells migrate into the voids. **b**, Representative images of actin, microtubule and ER at convex (left) and concave (right) edges. MDCK cells stained with phalloidin (grey, actin marker) and DAPI (cyan) (top), anti-α-tubulin (grey, microtubule marker) and DAPI (cyan) (middle), and MDCK cells expressing mApple-Sec61β (grey, ER marker) and labelled with DRAQ5 (cyan) (bottom). The edge of the cells near the curvature is enlarged on the right of individual images. Scale bar, 10 μm. **c**, MDE quantifications for microtubules, actin and ER at convex (green) and concave edges (pink). From left to right: *n* = 62, 69, 74, 62, 84 and 76. **d**, The fraction of ER tubules at the front as a function of curvature, *n* = 50 at convex and *n* = 72 at concave. **e**, Quantification of the fraction of ER tubules present at the front of the cells at the convex (green) and concave (pink) edges. From left to right: *n* = 50 and 52. **f**, Representative super-resolution images of MDCK cells expressing mApple-Sec61β at convex (left) and concave (right) edges. Insets: tubules at the front (left) and sheets at the front (right). **g**, Representative images of cells stained with anti-Climp63 (yellow, ER sheet marker), phalloidin (pink) and DAPI (cyan) at the convex (top) and concave (bottom) edges. The blue arrowheads

show low Climp63 at the edge of the cell and the white arrowhead shows Climp63 enrichment at the edge. Scale bar, 10 μm. **h**, Quantification of the fraction of Climp63 at the front of the cell edge as a function of curvature, *n* = 74 at convex and *n* = 104 at concave. **i**, A schematic representing the ex vivo wound healing experiment. **j**, Representative images of actin and microtubule at convex (left) and concave (right) edges in mouse embryonic skin wounds. E14.5 wounded skin stained for phalloidin (grey) and DAPI (cyan) (top) and anti-α-tubulin (grey) and DAPI (cyan) (bottom). The edges of the cells near the curvature are enlarged on the right of individual images. Scale bar, 10 μm. **k**, Representative images of wounded embryonic skin stained with anti-KDEL (magenta, ER marker), anti-Climp63 (yellow) and DAPI (cyan) at convex (top) and concave (bottom) edges. The edge of the tissue is marked with the dashed lines. Scale bar, 10 μm. **l**, Super-resolution images of wounded embryonic skin stained for anti-Sec61β antibody, convex (green inset) and concave (pink inset) with the edges zoomed in and enlarged on the right. Scale bar, 5 μm. Data are mean ± s.e.m. with a one-way ANOVA test (**c**) or two-tailed *t*-test (**e**). The dashed pink lines represent a concave edge and dashed green lines represent a convex edge. All experiments were performed at least three independent times.

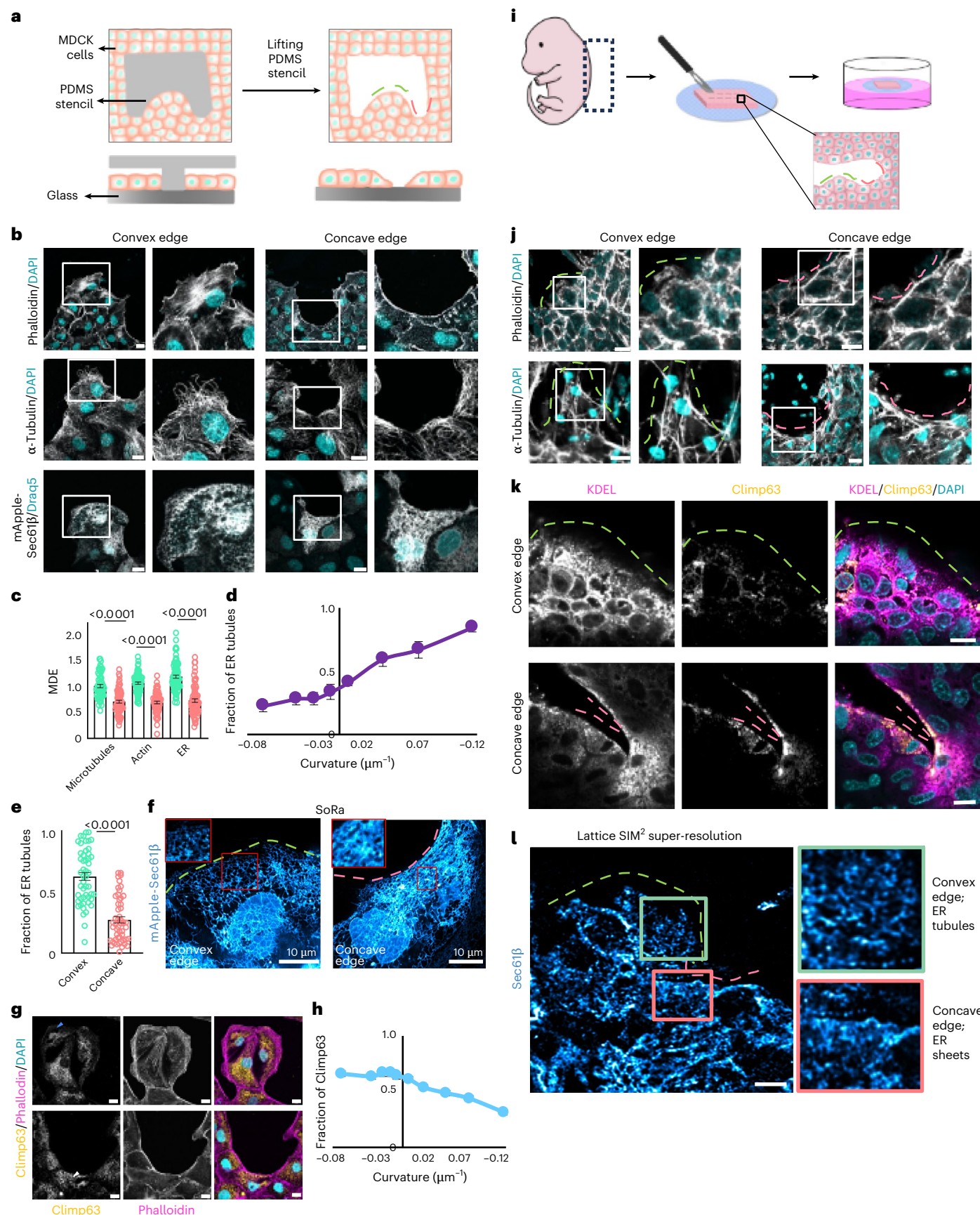

only weak variation across the range of negative curvatures. These results demonstrated that cells at convex edges have a significantly higher ER tubule fraction than those at concave edges, reflecting a clear curvature-dependent structural reorganization of ER (Fig. 1e). To visualize ER morphology at higher resolution across curvatures, we performed super-resolution imaging of mApple-Sec61β-transfected cells using the super-resolution by optical realignment (SoRa) method. We observed a highly reticulated network of ER tubules at the convex edge. In contrast, at the concave edge, we observed accumulation of dense ER, which consists of flat ER sheets as well as dense clusters of tubules[29] (Fig. 1f). Next, to check whether the dense ER at the front of migrating cells at the concave edge might be ER sheets, we labelled the cells with an antibody against cytoskeleton-linking membrane protein 63 (Climp63), which is an ER sheet marker. Interestingly, at the convex edge, Climp63 localized mostly around the nucleus at the cell centre, whereas at the concave edge, it accumulated at the cell periphery facing the edge (Fig. 1g). We also quantified the Climp63 fraction at the front, defined as the intensity of Climp63 staining at the front of the cell divided by the whole cell intensity, as a function of curvature. Climp63 fraction decreased with increasing convex curvature, but it remained comparatively high and relatively independent of the curvature magnitude at the concave edges (Fig. 1h and Extended Data Fig. 3d). This contrasting ER morphology was very clearly observed in flower-shaped patterned wounds, where concave edges showed dense sheet-like ER along with actin bundles, while neighbouring convex edges displayed ER tubules and lamellipodial protrusions (Extended Data Fig. 3e).

To test whether our findings extend beyond imposed geometries, we analysed epithelial colonies where curvature arises naturally. These spontaneous convex and concave edges closely match the scale of our micropatterns, allowing direct comparison between native and controlled settings. In this set-up, MDCK cells expressing mApple-Sec61β were allowed to form small colonies and actin organization was visualized using phalloidin staining. We observed that cells along convex regions formed lamellipodial protrusions and showed enrichment of ER tubules, whereas cells at concave zones exhibited actin bundles and dense sheet-like ER (Extended Data Fig. 3f). Quantification of ER tubule fraction as a function of curvature confirmed a significantly higher tubule content at convex edges compared with concave ones (Extended Data Fig. 3g), reinforcing the idea that ER organization is shaped by the local curvature. We also verified the generality of these observations in another epithelial cell line, EpH4-Ev, which originated from the mouse mammary gland (Extended Data Fig. 4a–d).

## Curvature-dependent ER morphology in mouse embryonic skin

Next, we examined the cellular response to spontaneously arising edge curvature in an incision wound made in a mouse embryonic skin explant (Fig. 1i). In this ex vivo model, dorsal skin explants from E14.5–E16.5 mouse embryos were wounded and cultured at an air–liquid interface to allow migration. Live imaging of SiR-actin-labelled tissue confirmed active migration, with actin accumulating at concave edges and small protrusions forming at convex edges within 30 min (Extended Data Fig. 4e and Supplementary Video 5). Hence, for subsequent experiments, we maintained wounded skin in the air–liquid interface and fixed after 30 min. We stained them for actin, microtubules and ER. Phalloidin staining showed that the epidermal cells formed lamellipodia at the convex wound edge and supracellular actin bundles at the concave wound edge (Fig. 1j and Extended Data Fig. 4f). Microtubule orientation also depended on the local wound curvatures, showing perpendicular and parallel alignments at the convex and concave wound edges, respectively (Fig. 1j and Extended Data Fig. 4g). Finally, to study ER morphology in the mouse embryonic skin wounds, we stained the skin explants with an anti-Climp63 antibody. Consistent with the results from the micropatterns (Fig. 1g,h), Climp63 was specifically localized at the cell periphery at the concave wound edge. In contrast,

staining these explants with an antibody against the KDEL sequence, a general ER marker, revealed that the ER network covered almost the entire cytoplasm (Fig. 1k). Together, these results indicated the specific enrichment of sheet-like dense ER structures at the concave wound edge. To further resolve the structural details of ER in the mouse skin tissue, we used Lattice SIM[2] super-resolution imaging and observed that ER indeed showed more tubular morphology at the spontaneously generated convex wound edge and dense sheet-like structures at the concave wound edge (Fig. 1l). Taken together, these observations suggested that epithelial cells interfacing different edge curvatures have proclivity for generating different ER morphologies.

## Characterization of dense sheet-like ER at the concave edge

Next, we wanted to determine whether the dense ER structures observed at the front of migrating cells at concave wound edges more closely resemble ER sheets or tubule clusters[29]. Although Climp63 and p180 are validated ER sheet markers[30,31], we directly benchmarked their localization in our system to confirm how faithfully they label ER sheets over ER tubules. We transfected MDCK cells with the general ER marker Sec61β, allowed them to form confluent monolayers, and performed SoRa imaging after immunostaining for Climp63 or p180. In nonmigrating cells, both Climp63 and p180 localized exclusively to dense, perinuclear ER structures, consistent with their known enrichment in ER sheets[30,31] (Extended Data Fig. 5a). These markers were absent from the peripheral ER, where tubules predominate, including potential dense tubular networks. This confirmed that Climp63 and p180 reliably distinguish ER sheets from tubules. We then examined migrating cell fronts, where Climp63 and p180 were strongly enriched in dense ER structures at concave edges (Fig. 2a,b), but not at convex edges. Quantitative analysis showed that the fraction of Climp63 and p180 signal over total ER (Sec61β) was significantly higher at concave than convex fronts (Fig. 2c,d). These results indicated that the dense ER at concave edges is largely composed of sheet-like structures. To further investigate the composition of the dense ER structures at the concave edge, we examined the localization of Rtn4b, a tubule-associated curvature-stabilizing protein also known to localize at the edges of ER sheets. Using dual labelling with Sec61β and Rtn4b followed by super-resolution imaging (Extended Data Fig. 5b), we observed that while Rtn4b localized broadly across the ER network, it was most prominently enriched at the rims of dense ER sheets (Extended Data Fig. 5b, right), with only little presence along distinct tubules and little to no signal within the sheet interiors. This preferential localization to sheet boundaries was more pronounced and supports the view that the dense ER at the concave edge is primarily composed of Climp63-positive sheets, with minimal contribution from tubular elements.

Finally, a recent study using the ER–plasma membrane contacts marker MAPPER demonstrated that ER–plasma membrane contacts polarize in single migrating cells, with Climp63-rich ER sheets at the rear showing more ER–plasma membrane contacts than Rtn4a-rich tubular ER at the front, facilitating polarized receptor tyrosine kinase signalling and phosphoinositide distribution[32]. While their focus was on the front and back of cells during single-cell migration, our study examines curvature-dependent collective epithelial migration, where mechanical forces and ER organization are regulated at the tissue scale. Despite these differences, we find an intriguing connection: curvature-driven ER morphology correlates with ER–plasma membrane contact distribution, with reduced ER–plasma membrane contacts at convex edges (ER tubule enriched) and increased contacts at concave edges (ER sheet enriched) (Extended Data Fig. 5c,d). This suggests that curvature may influence ER–plasma membrane contact site regulation through ER structural changes, affecting receptor tyrosine kinase and phosphoinositide signalling. Regulating phosphoinositide distribution can then regulate cytoskeletal dynamics, membrane trafficking and polarity, which are essential components cell migration. Altogether, multiple lines of evidence converge to support

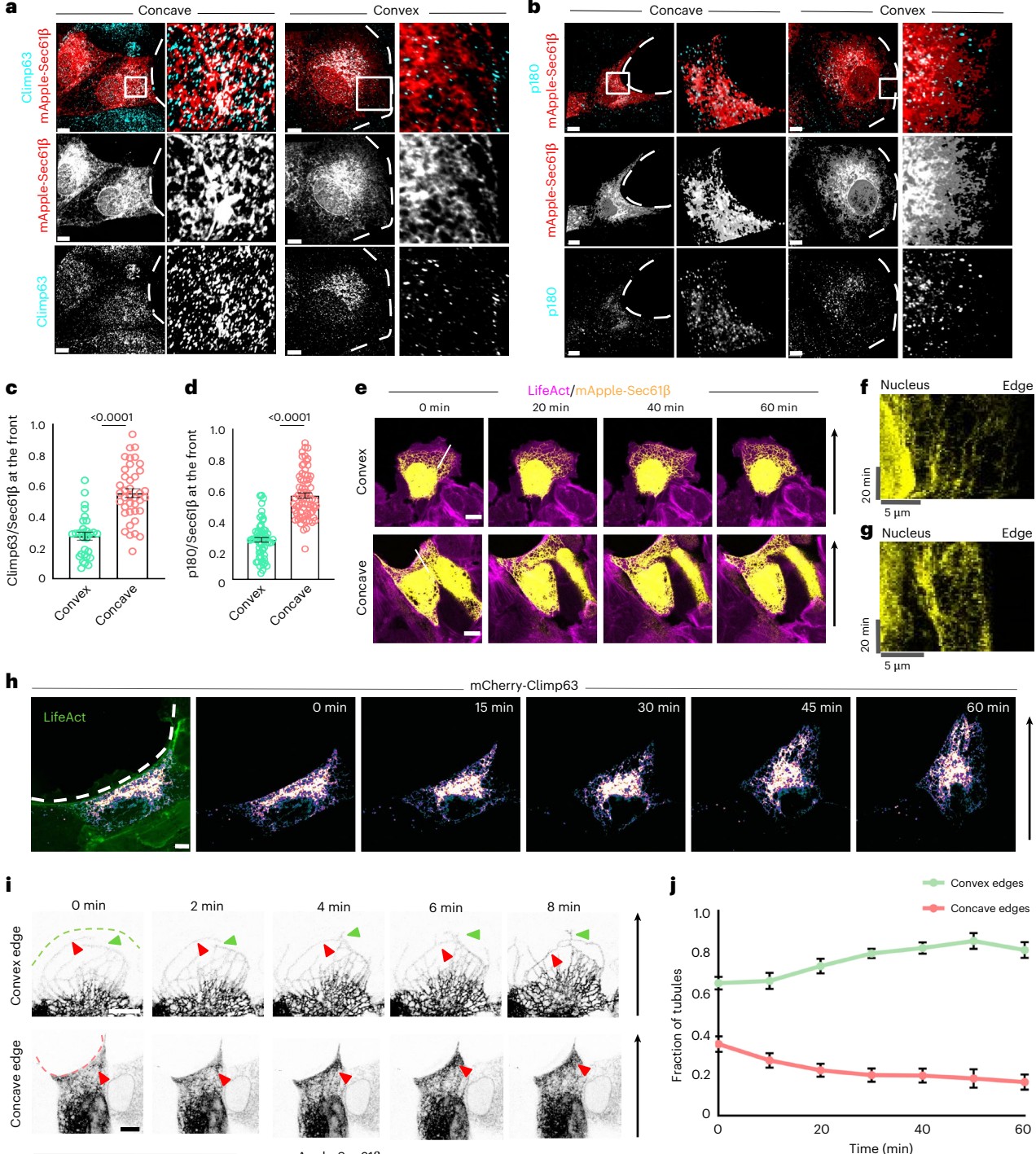

**Fig. 2 | Differential ER dynamics at convex and concave edges. a**, Representative images of MDCK cells expressing mApple-Sec61β (red) stained with anti-Climp63 antibody (cyan) migrating at concave (left) and convex (right) edges. Insets: zoomed-in enrichment of Climp63 towards the migrating front (dashed white line). Scale bar, 5 μm. **b**, Representative images of MDCK cells expressing mApple-Sec61β (red) stained with anti-p180 antibody (cyan) migrating at concave (left) and convex (right) edges. Insets: zoomed-in enrichment of p180 towards the migrating front (dashed white line). Scale bar, 5 μm. **c**, Quantification of the fraction of Climp63 on Sec61β at the front of migrating cells at convex (left, $n = 30$) and concave (right, $n = 39$) edges. **d**, Quantification of p180 on Sec61β at the front of migrating cells at convex (left, $n = 58$) and concave (right, $n = 73$) edges. **e**, Time-lapse images of LifeAct MDCK cells expressing mApple-Sec61β migrating at convex (top) and concave (bottom) edges. Scale bar, 10 μm. The white lines represent the lines for kymographs and the arrows represent the direction of migration. **f**, A kymograph of the line in **e** (top) representing the movement of the ER. **g**, A kymograph of the line in **e** (bottom) representing the movement of the ER. **h**, Time-lapse images of LifeAct (green) MDCK cells expressing mCherry-Climp63 (Fire LUT)) migrating at the concave edge. The dashed white line represents the migrating edge and the black arrow represents the direction of migration. Scale bar, 5 μm. **i**, Time-lapse images of the front of MDCK cells expressing mApple-Sec61β at convex (top) and concave (bottom) edges. The green arrowheads show the ER tubule growing in the direction of migration and the red arrowheads show the retrograde movement of the ER tubule (top) or the anterograde movements of tubules towards the edge (bottom). Scale bar, 10 μm. **j**, Quantification of the ER tubule fraction in the cells over 60 min at the two curvatures ($n = 9$ at concave and $n = 7$ at convex). Data are mean ± s.e.m. from two-tailed *t*-tests (**c** and **d**). All experiments were performed at least three independent times.

the conclusion that ER sheets are the predominant components of the dense ER structure at the concave edge.

## Differential ER dynamics at convex and concave edges

To understand how distinct ER structures might be linked to ER dynamics at different edge curvatures, we performed live-cell imaging using MDCK cells expressing LifeAct, which marks the actin cytoskeleton, and mApple-Sec61β, which marks the ER. We seeded these cells around the micropatterned PDMS, incubated for 15–18 h and subsequently lifted off the stencils, allowing the cells to start migrating into the gaps. Live imaging revealed two kinds of ER dynamics at the front of the migrating cell. At the convex edge, we observed a retrograde flow of dense ER structures towards the nucleus where major ER sheets are present (Fig. 2e and Supplementary Video 6). Plotting a kymograph confirmed this retrograde flow (Fig. 2f). In contrast, at the concave edge, we observed an anterograde movement of ER sheets towards the front of the migrating cells (Fig. 2e and Supplementary Video 7), also confirmed by the kymograph (Fig. 2g). Next, we conducted super-resolution live imaging using SoRa microscopy in LifeAct MDCK cells expressing Sec61β, which provided enhanced spatial clarity. This allowed us to clearly visualize dense ER sheet-like structures moving anterogradely towards the leading edge at concave curvatures, and retrogradely at convex curvatures, as shown in Extended Data Fig. 5e (Supplementary Videos 8 and 9). To further confirm that the dense ER structures at the concave front are indeed sheets, we expressed mCherry-Climp63, a known ER sheet marker, in LifeAct MDCK cells. To minimize the possibility of overexpression-induced structural artefacts, we selected cells with low to moderate expression. Live imaging of these cells revealed the anterograde movement of Climp63-positive ER sheets towards the leading edge at the concave wound edge (Fig. 2h and Supplementary Video 10). High-magnification imaging further revealed that at the convex edge, as the cells started migrating, newer ER tubules grew in the direction of migration (Fig. 2i, green arrowhead, and Supplementary Video 11) and existing tubules retracted back into the tubular ER network (Fig. 2i, red arrowhead, and Supplementary Video 11). In contrast, at the concave edge, the tubules grew towards the cell periphery, accumulated there and formed sheets (Fig. 2i and Supplementary Video 12). Interestingly, we also observed anterograde movements of the existing sheets towards the edge. Notably, even within the short timescales, individual ER tubules demonstrate consistent retrograde movement at convex edges and anterograde movement at concave edges, aligning with the overall patterns observed. We then quantified the fraction of ER tubules at the cellular front at different timepoints. This analysis revealed that a bias existed at the beginning, where the ER tubule fraction was determined to be 0.64 and 0.35, at the convex and concave edges, respectively (Fig. 2j). As time progressed, the fraction of tubules at the convex edge increased further, while it

decreased at the concave edge (Fig. 2j). Taken together, these results suggested that contrasting ER morphologies at the two edge curvatures are coupled with differential dynamics of ER.

## Protrusive and contractile forces regulate ER morphology

Epithelial cells exposed to the wound respond by generating protrusive forces due to branched actin polymerization at the convex edge and contractile forces due to actomyosin activity at the concave edge[15,16,33]. We investigated whether the microtubules and ER responded to these forces by altering their orientation and morphology, respectively. We first treated the cells with an Arp2/3 inhibitor, CK666, which prevented lamellipodia formation, attenuating the protrusive force. CK666-treated cells predominantly formed actin bundles at both convex and concave edges (Extended Data Fig. 6a,b). Importantly, in these cells at both edges, microtubules were arranged as parallel bundles (Fig. 3a,b). Simultaneously, the ER showed sheet morphology at the edge of cells (Fig. 3c,d), as opposed to the control cells where bundled microtubules and ER sheets were prevalent mainly at the concave edge. We next treated the cells with blebbistatin, inhibiting actomyosin contractility. Blebbistatin-treated cells formed protrusive structures at both convex and concave edges (Extended Data Fig. 6a,b). In these cells, microtubules aligned perpendicularly to the edge (Fig. 3a,b) and ER showed a tubular structure at both convex and concave edges (Fig. 3c,d). However, we wanted to confirm whether CK666 or blebbistatin treatments affected ER structure globally and therefore we examined several cells in the middle of the monolayer. The overall morphology of ER was not affected, rather the effect was confined to the edge of the monolayer (Extended Data Fig. 6c,d). These results suggested that microtubules and ER responded to different kinds of forces generated by the protrusive and contractile actin structures at different edge curvatures.

Next, to perturb the system in a more controlled manner, we used optogenetic tools to drive lamellipodia formation and followed the morphological changes in ER and microtubules using live-cell imaging. Rac1 is a small Rho GTPase that regulates actin assembly and dynamics, especially during lamellipodia formation. We used a genetically encoded photoactivatable Rac1 (PA-Rac1), which induces the formation of lamellipodial protrusions on activation with light with a wavelength of 458 nm (ref. 34). First, to check the response of microtubules to Rac1 activation, we transfected MDCK cells with mCherry-PA-Rac1 and labelled microtubules with SiR-tubulin. On photoactivation of Rac1, bundled microtubules opened up and grew perpendicularly to the cell edge (Fig. 3e and Supplementary Video 13), while we did not observe such microtubule dynamics in the neighbouring nonactivated cells (Fig. 3e and Supplementary Video 13). Subsequently, we cotransfected MDCK cells with mCherry-PA-Rac1 and GFP-Sec61β to study ER dynamics. On photoactivation of Rac1, ER matrices converted to tubules at the

**Fig. 3 | Protrusive and contractile forces regulate ER morphologies at two curvatures. a**, Representative images of MDCK cells stained with anti-α-tubulin (yellow) and DAPI (cyan) treated with dimethylsulfoxide (DMSO) (left), CK666 (middle) and blebbistatin (right) migrating at convex (top) and concave (bottom) edges. Scale bar, 10 μm. **b**, Quantification of the directionality of microtubules in cells treated with CK666 or blebbistatin (at different curvatures (*n* = 56, 59, 77, 79, 95 and 76 from the top (convex, control) to bottom (concave, CK666)). **c**, Representative images of MDCK cells expressing Sec61β (yellow) stained with phalloidin (magenta) and DAPI (cyan) treated with DMSO (left), CK666 (middle) and blebbistatin (right) at convex (top) and concave (bottoml) edges. Scale bar, 10 μm. **d**, Quantification of the fraction of ER tubules at the front of the cells treated with DMSO, CK666 or blebbistatin at convex (green, left) and concave (pink, right) edges. From left to right: *n* = 40, 42, 41, 37, 37 and 42. Ctrl, control (DMSO). **e**, Time-lapse imaging of cells expressing PA-Rac1 (magenta) labelled with SiR-tubulin (yellow), photoactivated with a 445 nm laser in the region shown with the red circle. The green circle represents a part of the nonactivated region and the zoomed-in regions show SiR-tubulin at activated and nonactivated regions. Scale bar, 10 μm. **f**, Time-lapse images of cells

expressing PA-Rac1 (magenta) and GFP-Sec61β (yellow), photoactivated with a 445 nm laser in the region marked with the red circle. The green circle represents the nonactivated region and the zoomed-in images show GFP-Sec61β at activated and nonactivated regions. Scale bar, 5 μm. **g**, Quantification of the fraction of ER tubules in photoactivated regions before and after 20 min of activation (right) and nonactivated regions before and after 20 min (left). *n* = 8 nonactivated and *n* = 9 activated. **h**, Representative images of MDCK cells expressing mApple-Sec61β treated with DMSO (left) or Calyculin A (right). Scale bar, 10 μm. **i**, Quantification of the fraction of tubules in cells treated with DMSO or Calyculin A. From left to right: *n* = 47 and 66. **j**, Representative images of mouse embryonic skin wounds stained with KDEL (magenta), Climp63 (yellow) and DAPI (cyan) and treated with DMSO (top), blebbistatin (middle) and CK666 (bottom). Green insets: Climp63 in cells zoomed in at the convex edge. Pink insets: Climp63 in cells zoomed in at the concave edge and blue arrowheads point towards the edge of the cells. Scale bars, 10 μm. Data are mean ± s.e.m. with a one-way ANOVA test (**d**), two-tailed paired *t*-test (**g**) or two-tailed *t*-test (**i**). The dashed pink lines mark the concave edges and dashed green lines mark the convex edges (**a**, **c** and **j**). All experiments were performed at least three independent times.

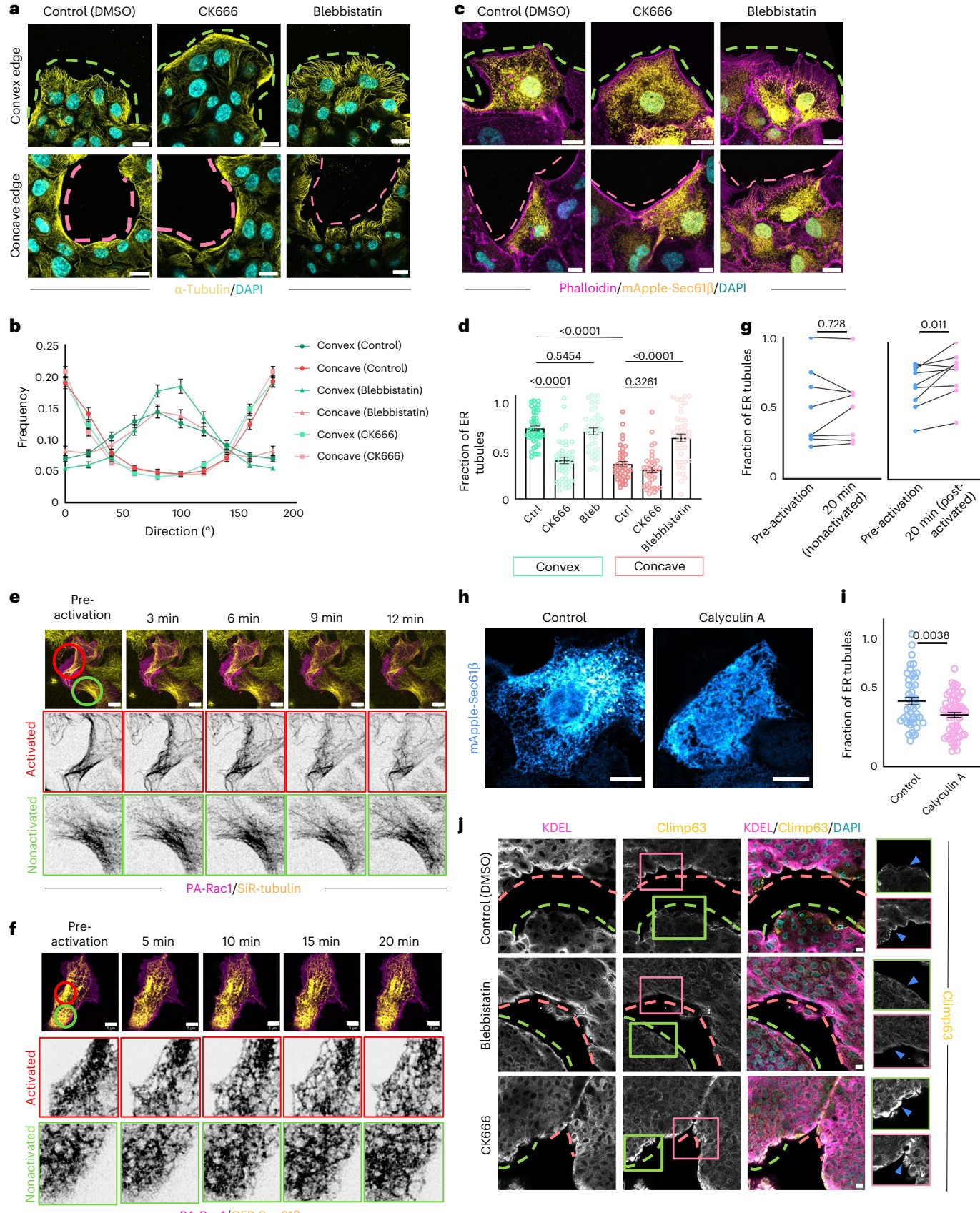

site of activation of Rac1, while ER morphology did not change considerably in the nonactivated region of the cell (Fig. 3f and Supplementary Video 14). Quantifying the fraction of ER tubule on Rac1 activation showed a significant increase in this parameter in activated regions as opposed to nonactivated regions (Fig. 3g). These results together confirmed that the protrusive forces generated due to actin polymerization and branching at the convex edge guide the organization of ER and the microtubule network. Conversely, to augment actomyosin contractility, we went on to treat MDCK mApple-Sec61β cells with Calyculin A. Calyculin A is a myosin light chain phosphatase inhibitor that leads to hyperactivated myosin[35]. We observed a considerable expansion of ER sheets and matrices in the cell on Calyculin A treatment (Fig. 3h) and a decrease in ER tubule fraction (Fig. 3i).

We then explored whether these mechanical forces influence ER structure in the ex vivo wound healing model. To test this, we pretreated the embryonic skin explants with blebbistatin (100 µM) or CK666 (100 µM) 2 h before wounding. After creating the wounds, we let the migration happen in the medium containing the inhibitors. We then fixed the tissue and stained it for KDEL and Climp63. We observed that on blebbistatin treatment, the cells at both convex and concave edges had considerably less accumulation of Climp63 as compared with control (Fig. 3j), whereas on CK666 treatment, we observed a high Climp63 signal at the edge of the cells at both convex and concave edges (Fig. 3j). Taken together, these results suggested that the formation of ER tubules and sheets at convex and concave edges, respectively, depends on the different cellular forces acting at different edge curvatures.

## Mathematical model of curvature-dependent ER morphologies

Next, to obtain a fundamental understanding of why a particular morphology of ER is preferred in response to geometrical and mechanical cues, we constructed a mathematical model. The ER forms a dynamic and intricate network within cells. While previous models have explored ER shape using Langevin dynamics[36,37] and the influence of curvature-stabilizing proteins on tubular/sheet abundance[30,38], our experiments highlight a crucial role for mechanical forces, specifically protrusion and contraction, in shaping ER morphology. To address this, we developed a computational model of the cell, comprising cytoplasm, actin cortex and ER, examining three distinct ER morphologies: perpendicular (tubular), sheet and parallel (Fig. 4a). We assume that morphology-dependent deformation of ER contributes to the strain energy of the cell. We hypothesized that the difference in the morphologies of ER observed at convex and concave edges is due to differences in the strain-energy state of the cell when the cell contracts and protrudes. That is, ER tends to be in tube-like morphology during protrusion and sheet-like during contraction because the combination of morphologies and forces acting on the cell results in lower strain energy.

We modelled cells as 3D mechanical entities with ER embedded in the cytoplasm and an actin cortex at the cell edges. The geometry considered for analysis was the front of the cell where ER morphologies were observed to change due to varying mechanical and geometric stimuli; hence, we did not take the effect of the nucleus into account. We modelled the actin cortex as a strain rate-dependent nonlinear constitutive law, as proposed previously[39], and generated forces necessary for curvature-dependent contraction or protrusion (Supplementary Note 2). For simplicity, the cytoplasm and ER were modelled as linear elastic materials. Moreover, since the aim of the model is to study the effect of curvature-dependent forces on ER morphology, we have not explicitly modelled microtubule structures, their response to curvature or their effect on ER reorganization. In vitro and ex vivo experiments showed that ER exhibited a tube-like perpendicular morphology at the convex edge during lamellipodia-based protrusion and sheet-like morphology at the concave edge during purse string movements. In silico studies were performed by considering cells with convex or concave

edges with varying magnitudes of curvatures and ER morphologies. While perpendicular and sheet ER indicated distinct ER configurations, parallel ER is assumed to represent a cluster of parallel tubules. We compared the strain energy density, as given in equation (1), between the cases to find out which morphological state of ER was preferred for the given edge curvature, where $\Sigma$ is the stress, $\epsilon$ is the strain and $V$ is the volume of the cell.

$$\text{Strain energy density} = \frac{1}{V}\frac{1}{2}\int_V \Sigma : \epsilon dV \qquad (1)$$

To begin with, we considered a cell with a flat edge denoted as having a curvature of 0 (Fig. 4a). We observed that when cells experience purse string forces resulting in contraction, the cell with ER sheet morphology exhibited lower strain energy compared with cells with perpendicular ER (Fig. 4b). Owing to the contractile forces generated by the actin cortex, the ER experiences high bending moments (Extended Data Fig. 7a). For a given volume of ER, the ability of sheet ER to resist bending moments better than perpendicular ER results in lowering the strain energy density of the cell experiencing purse string forces. In contrast, we observed that when cells experienced protrusive forces, the cell with perpendicular ER exhibited lower strain energy density compared with the cell with sheet ER (Fig. 4c). The axial forces during protrusion are dominant over the bending moments (Extended Data Fig. 7b). For a given volume of ER, since perpendicular ER resist protruding forces better than sheet ER, the strain energy density of the cell with perpendicular ER is lower during protrusion.

Next, we simulated cells with positive (convex) and negative (concave) curvatures (Fig. 4d). For a given curvature, we kept the volume of ER in the cell constant across the different ER morphologies considered. We observed that during protrusion at convex edges, the strain energy density of a cell with either sheet ER or parallel ER was higher than that of a cell with perpendicular ER, indicating that ER prefers to stay in the perpendicular tube-like morphology aligned along the direction of protrusion (Fig. 4e). Plotting the displacement profile for the cell undergoing protrusion showed that the displacement at the tip was higher for the cell with sheet ER compared with perpendicular ER (Extended Data Fig. 7c), leading to higher elastic strain energy. In addition, as the curvature increased, the difference in strain energy density between cells with sheet and perpendicular ER morphologies increased, while the strain energy density of a cell with parallel ER remained close to that of the sheet ER, irrespective of curvature (Fig. 4e). This implies that, while the probability of finding perpendicular ER is always higher than sheet ER, it increases further with increasing positive curvature. Interestingly, this result corroborated our previous experimental finding that the fraction of ER tubules increases at the convex edge with increasing magnitude of curvature (Fig. 4f). Plotting ER tubules and sheets at the convex edge as a function of curvature also revealed that as the curvature increased, the amount of tubules at the front increased while the amount of sheets decreased (Fig. 4g).

In contrast, during contraction, the strain energy density with sheet ER was lower than that of a cell with perpendicular ER, indicating that cells preferred sheet ER morphology over perpendicular tube-like morphology (Fig. 4h). We also observed that parallel ERs had a strain energy density higher than that of sheet ER but lower than that of perpendicular ER. In contrast to protrusion, under contraction, displacement profiles showed that cells with perpendicular ER had higher displacement compared with cells with sheet ER (Extended Data Fig. 7d). As the curvature increased, the model showed that the difference in strain energy density between cells with sheet and perpendicular ER morphologies decreased up to a certain curvature level and remained constant for higher curvatures (Fig. 4h). This implies that, while the probability of finding sheet ER is always higher than perpendicular ER, it exhibited curvature independence for higher levels of negative curvatures. The model displayed behaviour similar

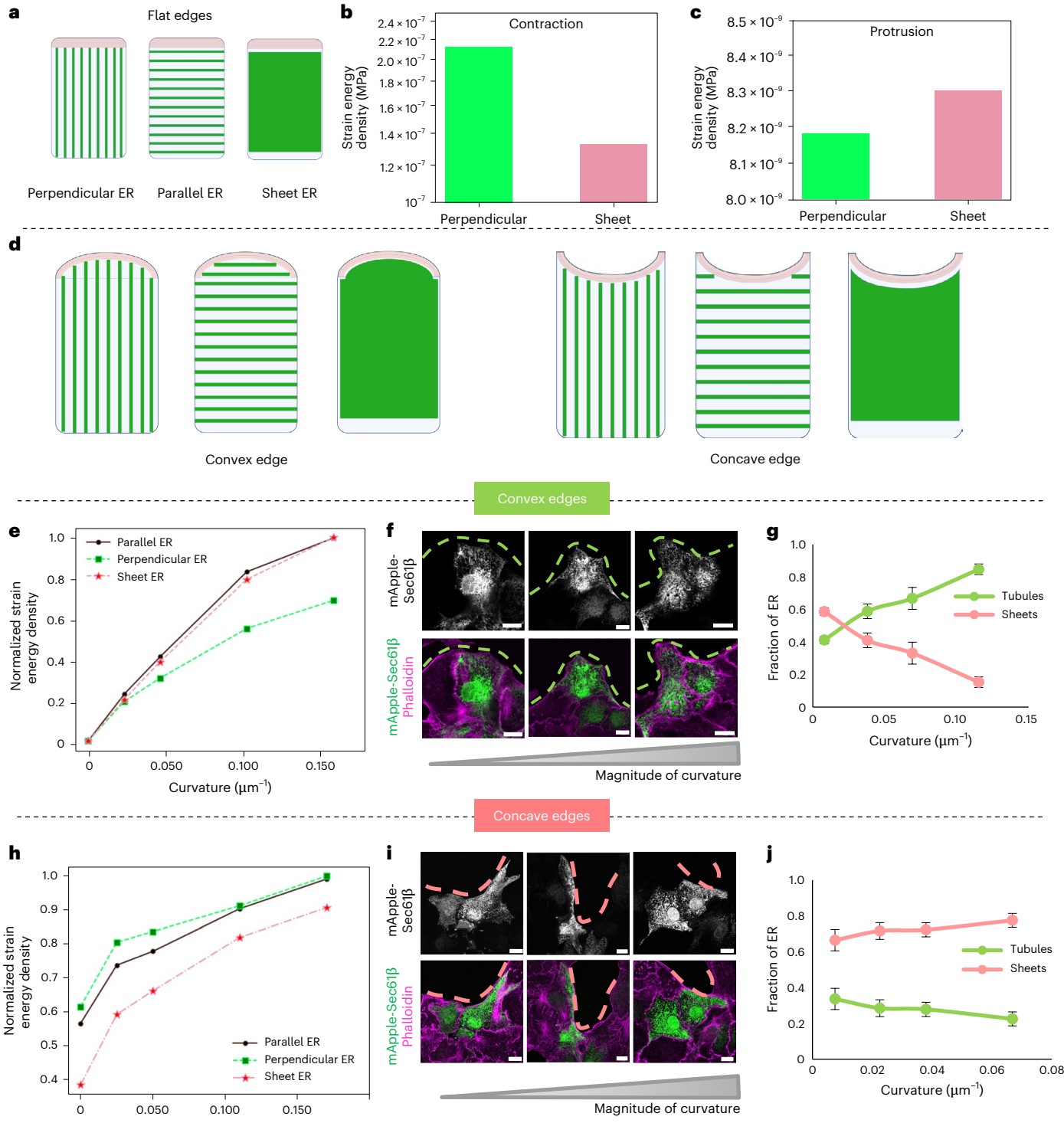

**Fig. 4 | Mathematical model of edge curvature-dependent ER morphologies.**
**a**, A schematic representing the front of the cell with a flat edge, an actin cortex (magenta) and ER (green) with perpendicular or tubular ER (left), parallel ER (middle) and sheet ER (right) morphologies. **b**, The strain energy density of a cell with a flat edge undergoing contraction with perpendicular and sheet ER morphologies. **c**, The strain energy density of a cell with a flat edge undergoing protrusion with perpendicular and sheet ER morphologies. **d**, A schematic representing a cell with convex (left) and concave (right) curvatures exhibiting different ER morphologies as in **a**. **e**, The normalized strain energy density of a cell with parallel ER, sheet ER or perpendicular ER with increasing positive curvature during protrusion. **f**, Representative images of MDCK cells expressing mApple-Sec61β (green) and stained with phalloidin (magenta) migrating at

positive curvatures of increasing magnitude (left to right). Scale bar, 10 μm.
**g**, Quantification of the fraction of tubules and sheets at the front of the cells migrating at varying magnitudes of positive curvature ($n = 52$). **h**, Normalized strain energy density of a cell with parallel ER, sheet ER or perpendicular ER with increasing negative curvature during contraction. **i**, Representative images of MDCK cells expressing mApple-Sec61β (green) and stained with phalloidin (magenta) migrating at negative curvatures of increasing magnitude (left to right). Scale bar, 10 μm. **j**, Quantification of the fraction of tubules and sheets at the front of the cells migrating at varying magnitudes of concave curvature ($n = 66$). The dashed pink lines mark the negative curvatures and dashed green lines mark the positive curvatures (**f** and **i**). Data are mean ± s.e.m. (**g** and **j**).

to our previous experimental findings, showing that the fraction of ER sheets is higher at concave edges than tubules (Fig. 4i). Plotting the ER tubules and sheets at the concave edge as a function of curvature revealed that as the curvature increased, the amount of tubules at the front decreased and the amount of sheets increased, though it did not vary as much as in the convex case (Fig. 4j). Collectively, from the simulation results shown (Extended Data Fig. 7c–i), we inferred through the strain energy minimization hypothesis that the probability of finding a particular distribution of ER depends on the curvature of the cell and whether the cell is undergoing contraction or protrusion. Therefore, the distribution of ER responds to the geometric and biomechanical stimuli present in a cell.

### ER–microtubule interactions promote ER tubule generation

ER and microtubules have remarkable structural interdependency[27,40–42]. Moreover, ER morphology and dynamics are known to be regulated by microtubules[42,43]. In our study, we observed a great correlation between the organization of ER and microtubules. The tubular ER network coincided with the perpendicularly arranged microtubules, while lamellar ER coincided with parallelly arranged microtubules (Fig. 5a). We therefore wondered whether microtubules might guide edge curvature-dependent ER morphology. To this end, we first disrupted microtubules by treating the cells with 5 μM nocodazole. As expected[44], upon nocodazole treatment, ER tubules disappeared irrespective of the edge curvature (Fig. 5b,c). We then tested whether ER tubule formation at the convex edge involved ER movements along microtubules by sliding and tip attachment complex (TAC) dynamics[45,46] (Fig. 5d). We perturbed the ER sliding dynamics by expressing KHC+ (kinectin-binding domain of kinesin) or KNT+ (kinesin-binding domain of kinectin), which led to disruption of the kinectin–kinesin interaction[47]. In the cells expressing KHC+ or KNT+, the fraction of ER tubules at the convex edge was significantly reduced as compared with the control EGFP-expressing cells (Fig. 5e,f). However, at the concave edge, the accumulation of ER sheets in KHC+ and KNT+ expressing cells did not change as compared with the control EGFP-expressing cells (Fig. 5e–g). Next, we perturbed the TAC movement of the ER by expressing a dominant negative form (EB1-c) of microtubule plus end-binding protein1, EB1, which led to its reduced binding with the microtubules[48]. In mutant EB1-c-expressing cells, the fraction of ER tubules at the front decreased at the convex edge (Fig. 5h,i), but the accumulation of ER sheets at the concave edge did not display any change as compared with the control cells (Fig. 5h–j). Taken together, these results suggested that both kinectin–kinesin-dependent sliding and TAC dynamics of ER are important for the generation of ER tubules at the convex edge but not for ER sheet accumulation at the concave edge.

### ER morphologies dictate cellular migration modes

Next, we examined whether ER morphology influences the curvature-specific mode of epithelial migration at wound edges. To this end, we manipulated the ER morphology by overexpressing either an integral ER membrane protein Rtn4a, which leads to the enrichment of ER tubules, or an ER sheet-associated protein Climp63, which leads

to the enrichment of ER sheets[30]. Indeed, MDCK cells overexpressing Rtn4a-GFP showed a higher ER tubule fraction and MDCK cells overexpressing mCherry-Climp63 showed a higher ER sheet fraction as compared with the control cells overexpressing mApple-Sec61β (Extended Data Fig. 8a). Subsequently, we seeded these cells around the PDMS micropattern and studied their migration behaviour at different edge curvatures. To normalize across experiments, we calculated the probability of forming lamellipodia by dividing the number of transfected cells forming the lamellipodia by the total number of transfected cells at a particular edge curvature. For cells overexpressing Rtn4a, the percentage of cells forming lamellipodia at the convex edge was comparable to that for wild-type (WT) cells. However, at the concave edge, where the WT cells majorly formed actin bundles, Rtn4a-overexpressing cells showed a significantly higher probability of lamellipodium formation (Extended Data Fig. 8b–d). In contrast, Climp63 overexpression, which amplified ER sheet morphology, resulted in a significant decrease in the percentage of the lamellipodium-forming cells at the convex edge (Extended Data Fig. 8c–e). Transient overexpression enabled rapid phenotype screening and dose-dependent analysis with internal controls, though variability in expression levels across cells may complicate the interpretation. To address this problem, we generated stable MDCK lines expressing Rtn4a-GFP or mCherry-Climp63 via antibiotic selection, followed by fluorescence-activated cell sorting (FACS) before each experiment to ensure uniform expression. The FACS-sorted Rtn4a-GFP cells formed prominent lamellipodia at both convex and concave edges (Fig. 6a), with MDE analysis showing increased protrusive activity at concave edges (Fig. 6b). In contrast, FACS-sorted Climp63-expressing cells displayed apical actin bundles and reduced basal lamellipodia at both curvatures, reflected in lower MDE values at convex edges (Fig. 6a,b). Time-lapse imaging revealed faster gap closure by Rtn4a-GFP cells, including individual cell migration into the gap, while Climp63-expressing cells also showed accelerated closure, probably via contractile, bundle-driven migration (Fig. 6c). These results highlight the instructive role of ER morphology in regulating migration mode and speed.

To examine whether ER morphology influences cytoskeletal organization in nonmigrating cells, we again transiently overexpressed Rtn4a and Climp63 in MDCK monolayers to induce ER tubule- and sheet-enriched states, respectively. The mosaic produced by the transient transfection allowed us to directly compare the cytoskeletal organization of transfected cells against neighbouring WT cells within the same monolayer, serving as internal controls. Immunostaining for actin and microtubules revealed no notable differences in their organization in either condition (Extended Data Fig. 8f), suggesting that ER morphological changes do not overtly disrupt cytoskeletal architecture in nonmigrating cells. To further investigate whether ER restructuring affects collective properties such as cell shape index (defined as the perimeter/sqrt(area))[49], we seeded stable MDCK populations homogeneously expressing mCherry-Climp63 or Rtn4a-GFP. ZO-1 staining was used to delineate cell boundaries, and subsequent shape index analysis revealed slightly higher values in both Climp63- and Rtn4a-expressing monolayers compared with WT.

---

**Fig. 5 | The ER–microtubule interaction is important for ER organization at the convex edge. a**, Representative images of MDCK cells expressing mApple-Sec61β (yellow), stained with anti-α-tubulin (magenta) and phalloidin (blue), migrating at convex (top) and concave (bottom) curvatures. Scale bar, 10 μm. **b**, Representative images of MDCK-expressing mApple-Sec61β treated with DMSO (left) and nocodazole (right). Scale bar, 10 μm. **c**, Quantification of fraction of tubules in the whole cell treated with DMSO (blue) or nocodazole (pink). From left to right: n = 47 and 55. **d**, A schematic representing sliding and TAC-dependent movement of ER on microtubules. **e**, Representative images of MDCK cells expressing mApple-Sec61β (yellow) and pEGFP vector control (VC) (magenta, top), KHC+ (magenta, middle) or KNT+ (magenta, bottom) migrating at convex (left) and concave (right) edges. Scale bar, 10 μm. **f,g**, Quantification of the fraction of ER tubules at the front of cells expressing pEGFP VC, KHC+ or KNT+ migrating at the convex edge (from left to right: n = 33, 39 and 30) (**f**) or the concave edge (from left to right: n = 34, 24 and 18) (**g**). **h**, Representative images of MDCK cells expressing mApple-Sec61β (yellow) and EGFP VC (magenta, top) or EB1-c (magenta, bottom) migrating at convex (left) and concave (right) edges. Scale bar, 10 μm. **i,j**, Quantification of the fraction of ER tubules at the front of the cells expressing the EGFP VC or EB1-c migrating at the convex edge (from left to right: n = 27 and 33) (**i**) or the concave edge (from left to right: n = 24 and 20) (**j**). Data are mean ± s.e.m. with one-way ANOVA tests (**f** and **g**) or two-tailed t-tests (**c**, **i** and **j**). The dashed pink lines mark the concave edges and dashed green lines mark the convex edges (**e** and **h**). All experiments were performed at least three independent times.

While statistically significant, these changes were modest, and most cells retained a cobblestone-like morphology, suggesting limited physiological relevance from a shape-transition perspective[49] (Extended Data Fig. 8g,h).

Next, to dissect the relationship between ER morphology and cytoskeletal dynamics at the wound edge, we introduced a second layer of perturbations using drug treatments in Climp63- and Rtn4a-overexpressing cells. For example, we seeded the sorted

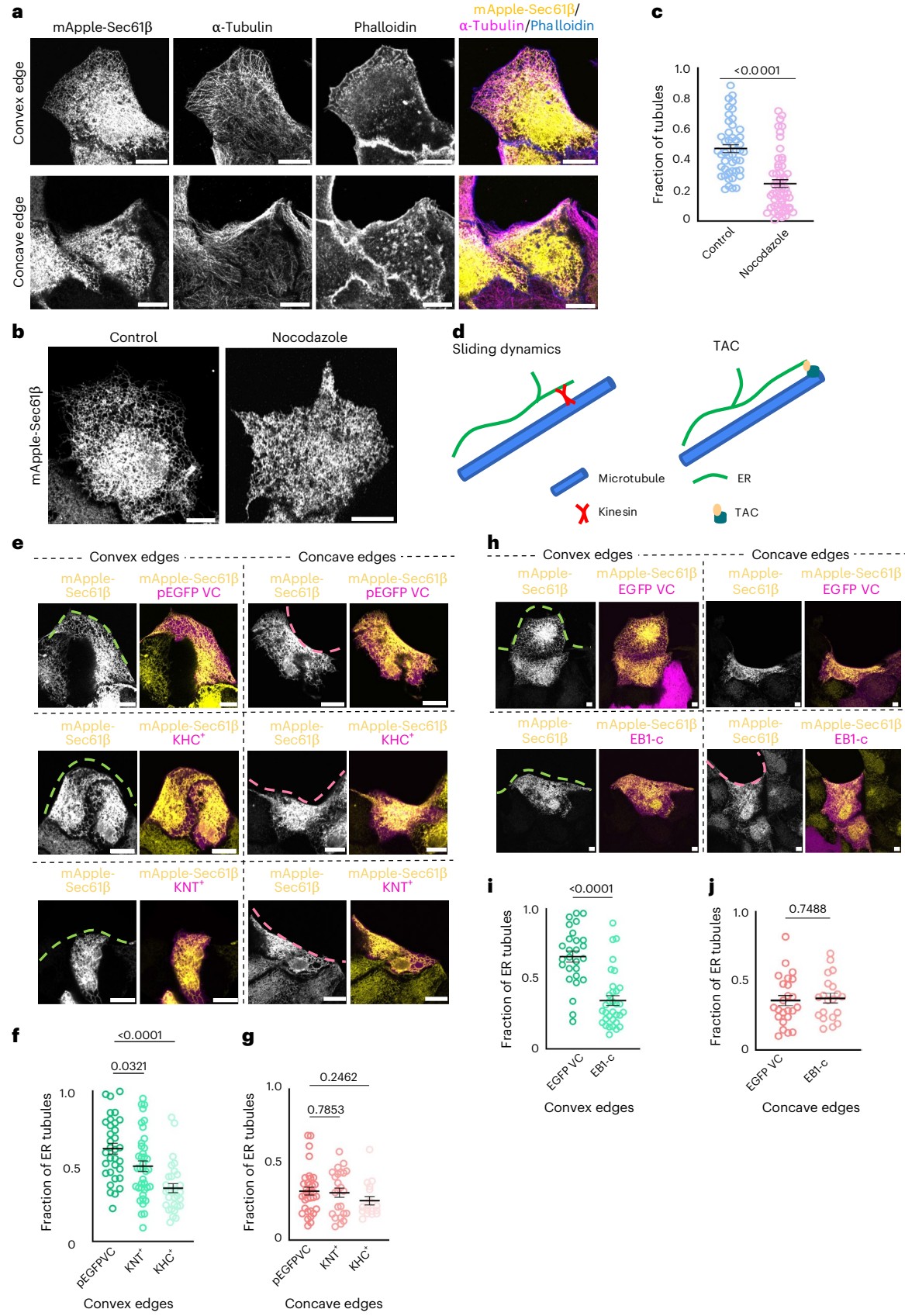

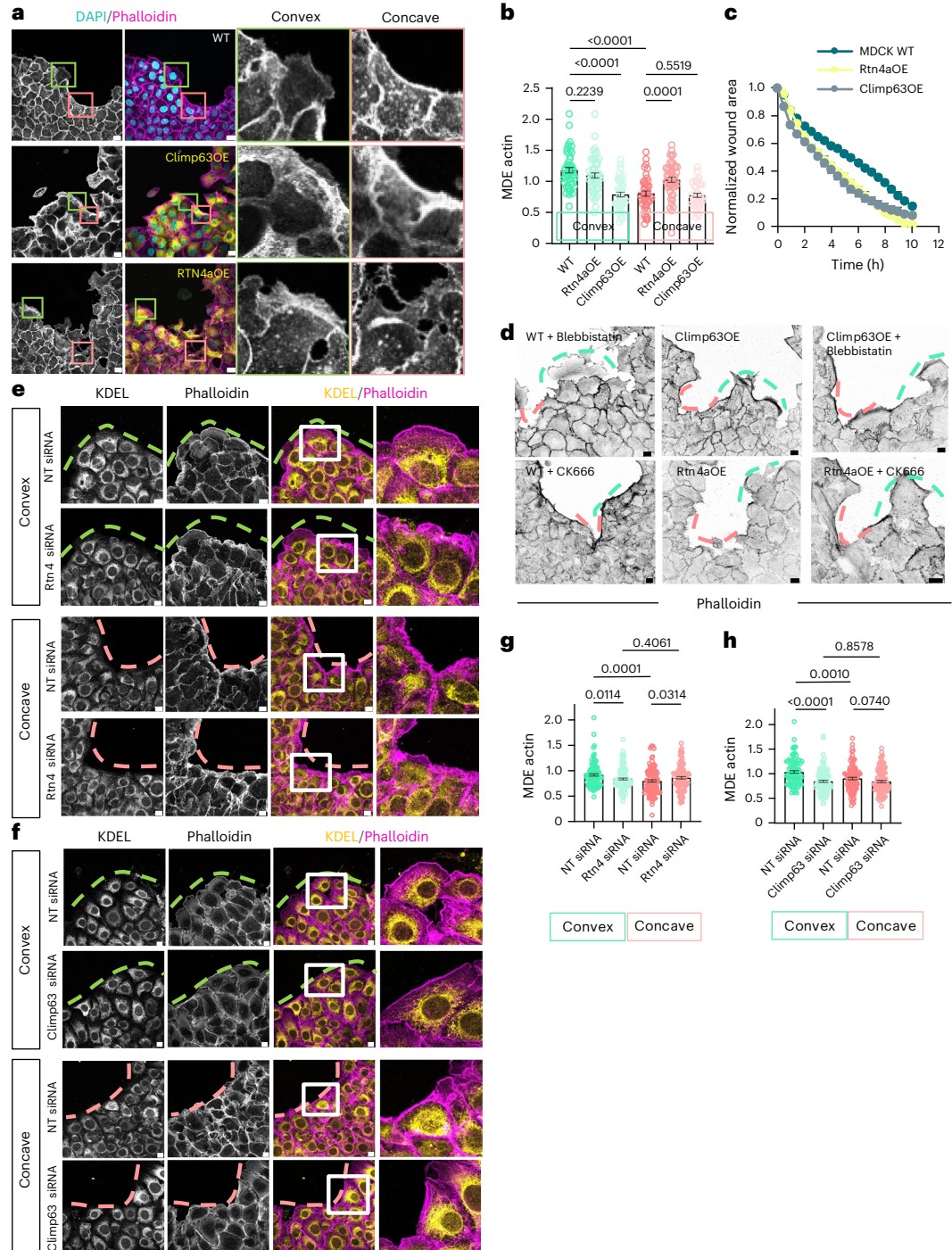

**Fig. 6 | Curvature-specific ER morphologies regulate the modes of epithelial migration. a**, Representative images of WT MDCK cells (top), overexpressing (OE) mCherry-Climp63 (yellow, middle) and overexpressing Rtn4a-GFP (yellow, bottom) migrating at micropatterned gaps stained with DAPI (cyan) and phalloidin (magenta). Zoomed insets: actin staining at convex (green) and concave (pink) edges in all three conditions. Scale bar, 10 μm. **b**, Quantification of MDE for actin at convex (green) and concave (pink) edges in WT, Rtn4aOE and Climp63OE cells. From left to right: n = 49, 50, 47, 43, 47 and 37. **c**, Quantification of a normalized wound closure area over time for MDCK WT (blue), Rtn4aOE cells (yellow) and Climp63OE cells (grey). Number of ROIs: 50 (WT), 45 (Rtn4a) and 49 (Climp63). **d**, Representative images of MDCK WT cells treated with blebbistatin (left), Climp63OE cells (middle) and Climp63OE cells treated with blebbistatin (right) seeded at micropatterned gaps and stained with phalloidin (black) (top) and MDCK WT cells treated with CK666 (left), Rtn4aOE cells (middle) and Rtn4aOE cells treated with CK666 (right) seeded at micropatterned gaps and stained with phalloidin (black) (bottom). Scale bar, 10 μm. **e**, Representative

images of EpH4 cells treated with nontargeting (NT) siRNA or Rtn4 siRNA migrating at convex (top two rows) and concave (bottom two rows) edges stained with anti-KDEL antibody (yellow) and phalloidin (magenta). Insets: zoomed-in regions of the front of migrating cells. Scale bar, 10 μm. **f**, Representative images of EpH4 cells treated with NT siRNA or Climp63 siRNA migrating at convex (top two rows) and concave (bottom two rows) edges stained with anti-KDEL antibody (yellow) and phalloidin (magenta). Insets: zoomed-in regions of the front of migrating cells. Scale bar, 10 μm. **g**, Quantification for MDE for actin in cells treated with NT siRNA or Rtn4 siRNA migrating at convex (green) and concave (pink) edges. From left to right: n = 104, 131, 118 and 104. **h**, Quantification for MDE for actin in cells treated with NT siRNA or Climp63 siRNA migrating at convex (green) and concave (pink) edges. From left to right: n = 95, 112, 99 and 110. Data are mean ± s.e.m. with one-way ANOVA tests (**b**, **g** and **h**). The dashed pink lines mark the concave edges and dashed green lines mark the convex edges (**d**–**f**). All experiments were performed at least three independent times.

Climp63-overexpressing cells around the PDMS chambers and treated them with blebbistatin to inhibit actomyosin contractility and stained them with phalloidin and anti-alpha-tubulin antibody. In WT cells, blebbistatin treatment promoted lamellipodia formation at both concave and convex edges (Extended Data Fig. 6a). However, in Climp63-overexpressing cells, blebbistatin failed to rescue lamellipodia, indicating that ER sheets override the effect of contractility inhibition, locking cells into a contractile state (Fig. 6d). These results suggests that ER tubules are necessary for branched actin polymerization and lamellipodia-driven migration. Blebbistatin treatment also failed to completely rescue microtubule rearrangements from parallel to perpendicular in Climp63-overexpressing cells (Extended Data Fig. 8i,j). In another experiment, we seeded sorted Rtn4a-overexpressing cells around PDMS chambers and treated them with CK666 to inhibit Arp2/3-mediated branched actin polymerization. In WT cells, CK666 treatment suppressed lamellipodia and promoted actin bundle formation at the convex edge (Extended Data Fig. 6a). However, in Rtn4a-overexpressing cells, CK666 failed to fully restore actin bundles, leaving a few persistent protrusive structures at the edge and disrupting the normally continuous contractile actin architecture (Fig. 6d). These results suggested that once ER tubules accumulate, they prevent full re-establishment of contractile actin structures, even when lamellipodia formation is inhibited. In contrast, CK666 treatment in Rtn4a-overexpressing cells led to a reorientation of microtubules from a perpendicular to a parallel arrangement, suggesting that microtubule organization adapts to the dominating actin architecture, and ER morphology may exert a more decisive influence on actin dynamics than on microtubules (Extended Data Fig. 8k,l). Notably, nocodazole treatment, which disrupts microtubules, consistently promoted ER sheet formation along with thick, continuous actin bundles at the edge and increased stress fibres within the cell, in both Climp63- and Rtn4a-overexpressing cells (Extended Data Fig. 9a,b). This again supports our earlier observation that microtubule–ER interactions facilitate ER tubule generation and suggests that microtubules contribute primarily to lamellipodia formation at convex edges but play a lesser role in actin bundling at concave edges (Fig. 5d–i). By combining ER structural manipulation with cytoskeletal drugs, we created a mechanistic tug-of-war that revealed ER morphology as a key regulator of cytoskeletal organization: sheets stabilize actin bundles and suppress lamellipodia, while tubules promote branched actin and weaken contractile structures. The failure of cytoskeletal drugs to fully rescue migration modes after ER perturbation underscores the instructive role of ER in guiding curvature-dependent migration.

To further validate the functional roles of ER structure, we conducted siRNA-mediated knockdown of Climp63 and Rtn4 in EpH4 epithelial cells (Fig. 6e–h). Knockdown efficiency was confirmed by western blotting (Extended Data Fig. 9c). Upon Rtn4 knockdown, we observed a loss of ER tubules (anti-KDEL staining; Extended Data Fig. 9d) and a corresponding decrease in lamellipodia formation, with increased polarized actin bundles at the wound edge. This was reflected in lower MDE values for actin (Fig. 6e,g), consistent with a shift towards contractile behaviour at convex curvatures. Conversely, Climp63 knockdown led to expansion of sheet-like ER (Extended Data Fig. 9e) and formation of actin bundles even at convex edges (Fig. 6f,h), again supporting the idea that ER sheets promote contractile migration modes and inhibit protrusive activity.

Finally, we tested whether smaller reticulons have similar effects by overexpressing Rtn2b, Rtn3S and Rtn4b in MDCK cells. Among these, only Rtn4b overexpression led to enhanced lamellipodia formation and suppression of actin bundles, including at concave edges (Extended Data Fig. 10a–f), mirroring the phenotype observed with Rtn4a overexpression. All reticulons (Rtn1–4) have been shown to localize on ER tubules and promote membrane curvature generation through a conserved reticulon homology domain[50]. Rtn4a and Rtn4b overexpression shifts ER morphology towards tubules in mammalian cells[51]

and, as shown in our study, enhances lamellipodia formation at both convex and concave edges. Rtn4b has also been linked to wound healing through its role in macrophage migration[52]. Together with previous findings, our results suggest that altering ER morphology through various methods consistently influences epithelial migration modes.

## ER morphologies regulate orientations of focal adhesions

Since ER-resident proteins interact with the focal adhesion complex[53–56], ER dynamics might be crucial for the maturation of cell-substrate adhesions during cell adhesion and migration[57]. Importantly, focal adhesions orient differently at the two edge curvatures during migration, aligning perpendicular to the edge at convex regions and parallel to the edge at concave regions[15,16]. We therefore hypothesized that different ER morphologies at the convex and concave edges might influence the orientation and stability of focal adhesions. To test this hypothesis, we first checked whether ER tubules or sheets contacted focal adhesion punctae, using paxillin as the focal adhesion marker. Co-localization images revealed that ER tubules were associated with perpendicular paxillin punctae at the convex edge, while ER sheets were associated with parallel focal adhesions at the concave edge (Fig. 7a). Next, we quantified the fraction of focal adhesions of a particular orientation associated with ER sheets or tubules. Adhesions aligned at 0–18° relative to the edge were considered as parallel and those aligned at 72–90° were considered as perpendicular focal adhesions. Quantification revealed that at both curvatures, perpendicular focal adhesions mainly contacted ER tubules, while the parallel focal adhesions were associated with the ER sheets (Fig. 7b). Then, to determine whether altering ER morphology would alter the orientation of focal adhesions, we examined the orientation of paxillin punctae in Rtn4a- or Climp63-overexpressing cells (Fig. 7c), which showed an increased propensity to form ER tubules and sheets, respectively. Subsequently, Rtn4a overexpression led to a decrease in the fraction of parallelly oriented focal adhesions and increased perpendicularly oriented adhesions at the concave edge (Fig. 7c). In contrast, Climp63 overexpression led to a decrease in the fraction of perpendicularly oriented focal adhesions and increased parallelly oriented focal adhesions at the convex edge (Fig. 7c). Taken together, these results suggested that morphological changes in the ER modulate focal adhesion, promoting different modes of migration at the two curvatures.

## Discussion

Epithelial gap closure is a fundamental process in both health and disease, involving collective migration through lamellipodia or purse string contractions, each with distinct actin and adhesion dynamics. This response is strongly influenced by edge curvature, but the underlying mechanisms remain unclear. Here, we show that among intracellular organelles, the ER undergoes striking curvature-dependent morphological changes and plays a key role in determining the mode of epithelial migration (Fig. 7d). The ER is the largest and most dynamic cell organelle, which has been hypothesized to be a sensor of multiple extracellular cues attributing to its dynamicity and elaborate structure spanning the whole cell[17,24,58,59]. The curvature-dependent shift in ER morphology, from tubules at convex edges to sheets at concave edges, suggests that ER organization senses and responds to physical cues, driven by protrusive and contractile forces from actin, microtubules and potentially intermediate filaments. Alongside these findings, our theoretical model suggests that these morphologies minimize strain energy within their corresponding geometric contexts. This provides a mechanistic basis for curvature-specific ER organization, though further work is needed to clarify how mechanical constraints and cytoskeletal inputs jointly shape ER structure during migration. Relevantly, an important consideration is whether the dense ER structures at concave edges represent true ER sheets or instead dense tubular matrices, as described in previous studies[29]. Our results indicate that the dense ER structures at concave edges are predominantly sheets,

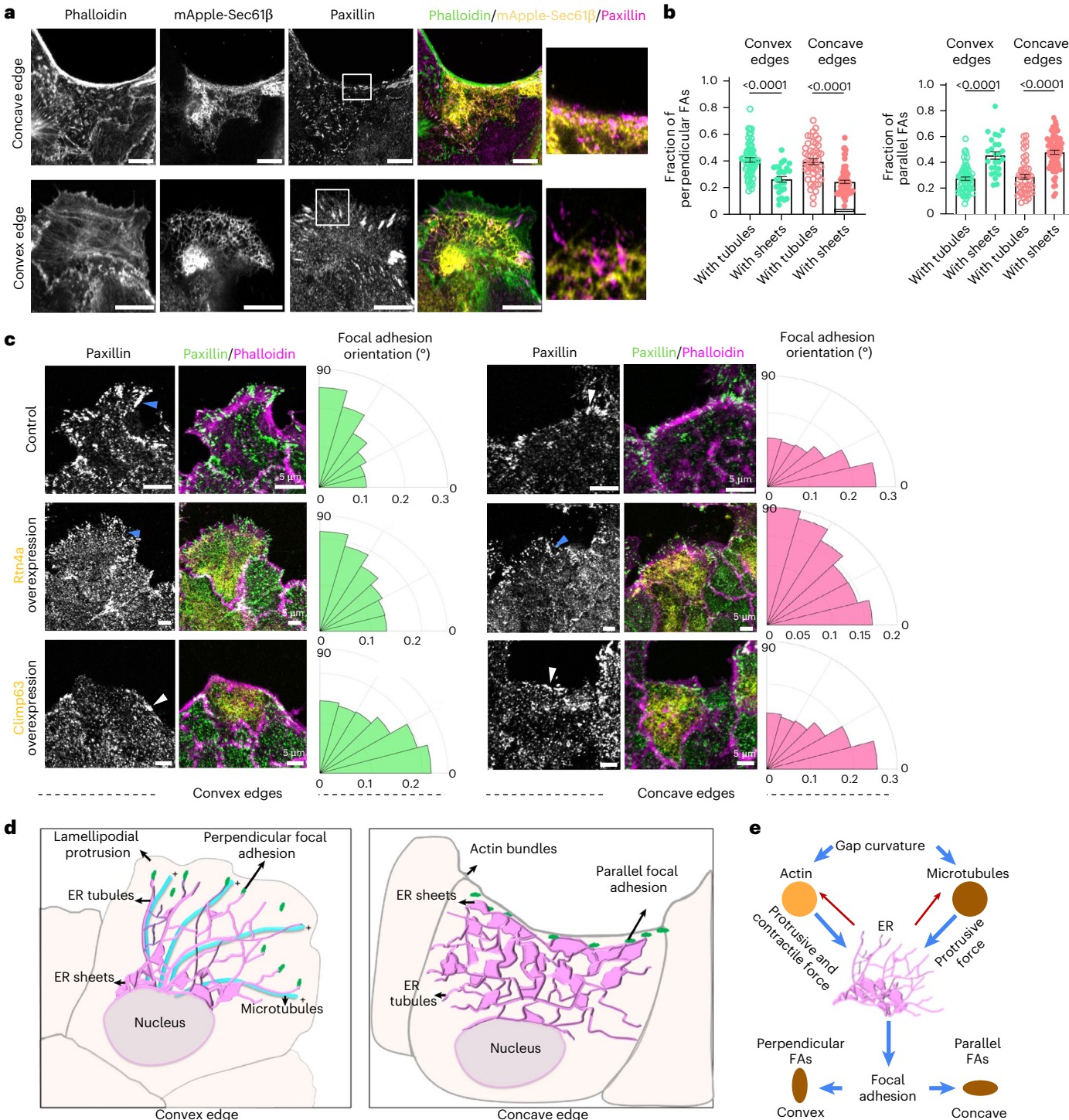

**Fig. 7 | ER morphologies affect focal adhesion orientations. a**, Representative images of MDCK cells expressing mApple-Sec61β (yellow) migrating at concave (top) and convex (bottom) edges stained with anti-paxillin (magenta) and phalloidin (green). Scale bar, 10 μm. Zoomed-in insets: the association between ER sheets and parallel adhesions (top) and ER tubules and perpendicular adhesions (bottom). **b**, Quantification of the fraction of perpendicular focal adhesions (from left to right: $n = 58, 26, 47$ and 70) and parallel focal adhesions (from left to right: $n = 58, 26, 47$ and 70). **c**, Representative images of MDCK WT (top), Rtn4a-GFP (yellow, middle) and mCherry-Climp63 (yellow, bottom) cells stained with anti-paxillin (green) and phalloidin (magenta). The blue arrowheads show perpendicular FA and white arrowheads show parallel FA. The rose plots on the right show the quantification for focal adhesion orientation with respect to

wound from 0° to 90° (at convex, WT $n = 63$, Rtn4a $n = 78$ and Climp63 $n = 103$; at concave, WT $n = 51$, Rtn4a $n = 100$ and Climp63 $n = 62$). **d**, Summary of the study: a schematic (left) depicting ER tubules at the front of cells migrating at convex edge sliding and growing along the microtubules, and associating with the perpendicularly oriented focal adhesions promoting lamellipodial crawling. A schematic (right) depicting ER sheets accumulated at the front of cells migrating at the concave edge, associating with parallel focal adhesions and promoting actin bundle-driven purse string closure. **e**, ER in mechanotransduction. Positioning ER as a key player in mechanotransduction, integrating signals from microtubules and actomyosin networks to regulate cellular responses to physical stimuli. Data are mean ± s.e.m with a one-way ANOVA test (**b**). All experiments were performed at least three independent times.

supported by Climp63 and p180 enrichment, ER–plasma membrane contact patterns and Rtn4b imaging. Consistently, modelling shows that ER sheets have the lowest strain energy at concave edges, reinforcing their role as key regulators of curvature-guided cytoskeletal organization and migration.

Interestingly, ER tubule formation at convex edges relies on microtubule-dependent dynamics, including kinesin-driven sliding and tip attachment, whereas these interactions are dispensable for ER sheet accumulation at concave edges. While it is possible that some ER sheet-associated proteins[60,61], including Climp63 and p180, may support ER sheet accumulation at the concave edge, whether actomyosin contractility can directly promote ER sheet accumulation remains unknown. In addition, ER structure and dynamics influence the distribution and post-translational modifications of microtubules[27,62]. Therefore, ER reorganization in response to curvature may influence microtubule modifications, potentially altering organelle distribution and polarity during migration.

Functionally, manipulating ER morphology alters the migration mode, with Climp63 promoting actin bundles and contractility even at convex edges, and Rtn4a enhancing protrusions at concave edges, probably through changes in focal adhesion orientation. However, ER-mediated regulation of focal adhesions could be just one of the mechanisms by which ER dictates the mode of epithelial cell migration. ER tubules are involved in lipid biogenesis, calcium signalling and forming membrane contact sites with other organelles and the plasma membrane. A reticulated network of ER tubules at the convex edge could also regulate lamellipodial crawling by providing membrane lipids via the contact sites and by releasing calcium for the activation of the actomyosin network. Conversely, ER sheets are involved in protein synthesis, and sheet accumulation at the concave edge might promote directed protein synthesis and delivery, facilitating purse string-like movements. How the conventional roles of ER subdomains differ at the two curvatures and guide cell migration remains unknown. Despite these remaining questions, our findings highlight the ER as a central mechanotransducer that integrates cytoskeletal signals (Fig. 7e) to coordinate curvature-dependent collective cell migration, underscoring the broader role of organelles in tissue homeostasis, development, and regeneration.

## Online content

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

## Methods

### Cell culture

The tetracycline-resistant WT MDCK cells utilized in this investigation were generously provided by Yasuyuki Fujita. These cells were cultured in Dulbecco's modified Eagle medium (DMEM) with GlutaMAX (Gibco), supplemented with 5% fetal bovine serum (Tet-free FBS, Takara Bio) and 10 U ml$^{-1}$ penicillin and 10 µg ml$^{-1}$ streptomycin (pen−strep, Invitrogen). Cultures were maintained in a $CO_2$ incubator at 37 °C. EPH4-Ev epithelial cells (ATCC, CRL-3063), used in the study, were maintained in DMEM with GlutaMax (Gibco), supplemented with 10% FBS (Tet-free FBS, Takara Bio) and 1.2 µg ml$^{-1}$ puromycin (Gibco). These cells were also maintained in a 37 °C, 5% $CO_2$ environment. Subculturing of both cell lines occurred every 3−4 days utilizing Trypsin/EDTA (Invitrogen).

### Mice

C57/6J strain mice, 8−12-week-old adult females were used in the study for all experiments involving mice. No statistical methods were used to predetermine sample size. For each independent experiment, embryos were dissected from different pregnant females. E14.5−15.5 stage embryos were dissected and used for all wound healing assays. The results are independent of the sex of the animals. The development and physiology of the embryonic epidermal tissue is not known to be different in different sexes, therefore sex of the animal was not considered a parameter in the study. The protocol was approved by the Institutional Animal Ethics committee, TIFR Hyderabad on behalf of Committee for the Purpose of Control and Supervision of experiments on Animals, India.

### siRNA and plasmid transfection

siRNAs from Dharmacon (Supplementary Table 2) were dissolved in 1× siRNA buffer (B-002000-UB-100) to prepare 5 µM siRNA stock. Stock solutions were aliquoted into multiple tubes to avoid repeated thawing. For siRNA transfection experiments in 12-well plates, Eph4 cells were seeded 16 h before transfection and transfected with siRNA using DharmaFECT 1 (Dharmacon, T-2001-02) according to the manufacturer's protocol. In brief, 75 nM siRNA and 2 µl DharmaFECT 1 were diluted with Opti-MEM medium (Gibco, 31-985-070) to prepare a 100 µl volume system in separate tubes. After 5 min of incubation at room temperature, the two tubes were mixed thoroughly and gently for another 20 min incubation. Then the 200 µl of transfection mix was transferred to each well for a 9−10 h transfection before the medium was replaced with the complete growth medium. The transfected cells were collected from the 12 wells 48 h after transfection and were seeded for migration experiments. For plasmid transfection, cells were treated with respective plasmids and Lipofectamine 2000 (Invitrogen), following the instructions provided by the manufacturer. At approximately 8−10 h post-transfection, cells were seeded for the migration assays.

### Stable cell lines

MDCK cells stably expressing Rtn4a-GFP or mCherry-Climp63 cells were transfected with the respective plasmids using Lipofectamine 2000 (Invitrogen) following the instructions provided by the manufacturer. At 16−18 h post-transfection, cells were seeded via serial dilution in a 96-well plate. At 48 h post-cell seeding, the cells were subjected to selection media (DMEM−GlutaMax plus 5% FBS) containing 600 µg ml$^{-1}$ of Geneticin (Invitrogen). The growth of single cell-derived colonies was monitored over 2 weeks following fluorescence confirmation. Colonies with a homogeneous expression of the fluorescence-tagged protein of interest were expanded and maintained in media containing 400 µg ml$^{-1}$. To ensure further homogeneous expression, these cells were sorted using FACS before each experiment. Briefly, the cells were trypsinized and the pellet was resuspended in Fluorobrite media with 20 mM HEPES. The suspension containing approximately 1 million cells were sorted and only cells showing high expression as compared with WT were selected and seeded for migration experiments.

### Western blotting

EpH4 cells treated with nontargeted siRNA, CLimp63 siRNA and Rtn4 siRNA were trypsinized and the pellet was resuspended in RIPA lysis buffer (50 mM Tris, 150 mM NaCl, 0.2% Triton X-100 and 1× protease and phosphatase inhibitor cocktail), followed by centrifugation at 16,000g for 10 min at 4 °C. The supernatant was used for total protein estimation by a Pierce 660 nm assay. The lysate was mixed with equal volume of 1× Laemmli buffer (0.004% bromophenol blue, 20% glycerol, 4% SDS and 0.125 M Tris−HCl pH 6.8) and 5% beta-mercaptoethanol and heated at 98 °C for 5 min, centrifuged and then 20 µg of protein was loaded in each well along with the ladder. Wet transfer was done onto 0.2 µm nitrocellulose membrane as per the Trans-Blot Turbo kit (Bio-Rad, 1704270) for 30 min. Blocking was in or with 5% BSA made in 1× TBS with 0.1% Tween-20 (1× TBSTw20) for 1 h at room temperature followed by primary antibody incubation overnight at 4 °C. After washing three times in 1× TBSTw20, the membranes were incubated in HRP-conjugated secondary antibodies for 2 h at room temperature. After washing, they were developed using Clarity Western ECL Substrate (Pierce ECL, Thermo Fisher Scientific) and visualized using an iBright gel doc.

### Micropatterning

PDMS6 stencils with pillars of defined shapes were created using soft photolithography. Briefly, 150-µm-thick SU-8 2075 (Kayaku advanced materials, Y111074) was coated on a silicon wafer followed by soft baking at 65 °C for 5 min and 95 °C for 20 min. The SU-8 was exposed in the desired pattern using the MicroWriter ML 3 Pro (Durham Optics Magneto Ltd.) followed by post-exposure baking at 65 °C for 7 min and 95 °C for 15 min. The unexposed photoresist was removed by immersing the wafers in the SU-8 developer (Kayaku Advanced Materials, Y020100) for 8−10 min. Patterned SU-8 wafers were then hard baked at 170 °C for 45 min. PDMS (Sylgard 184, Dow Corning) was mixed in a ratio of 1:10, degassed and poured over the patterned wafers. After curing at 85 °C for 2 h, stencils with pillars were peeled off. The stencils were then treated with 2% pluronic acid for 2 h followed by 15 min of a 1× PBS wash. The PDMS stencil and a Petri dish were plasma cleaned for 10 s, and the stencil was then placed on the dish. Cells were seeded around the pillars and, after confluence, the stencils were removed allowing the cells to migrate into the gaps.

### Cell migration assay

A solution of 20 µg ml$^{-1}$ fibronectin was applied to a glass-bottom Petri dish (Ibidi). After three washes to remove excess protein, the dish was air dried. The PDMS stencil underwent plasma cleaning for 20 s before being positioned onto the dish. The Petri dish with the stencil was then subjected to UV sterilization for 1 h. A concentrated suspension of MDCK or other cells was seeded around the pillars, followed by the addition of excess media around the stencil. Cells were allowed to settle around the pillars for 15−18 h. Subsequently, the stencils were lifted, enabling cell migration at various curvatures. Live-cell imaging or fixation occurred either immediately or after 30 min of migration.

### Mouse embryonic skin wounding

E14.5−E16.5 mouse embryos were dissected out. The dorsal skin from these embryos was then excised. The skin was placed on a filter paper (Whatman filter, 8 µm pore size) with the epidermal side up. Using a scalpel/fine scissor, incision wounds were made on the epidermis. The filter paper along with the tissue was kept floating in a 6-well plate containing 2 ml of DMEM with 5% FBS and 1% pen−strep. After the desired incubation period, the tissue was fixed in 4% paraformaldehyde for 2 h at room temperature. The tissue was washed in PBS and transferred to 90% ethanol in PBST (0.2% Triton X-100 in 1× PBS) overnight at −20 °C. The tissue was then rehydrated using graded ethanol (75%, 50% and 25% in PBST). After two more washes in PBST, the tissue was blocked using 10% normal goat serum in PBST for 2 h. Primary antibodies were added

and incubated overnight at 4 °C. This was followed by subsequent PBST washes, after which secondary antibodies were added and incubated for 2 h at room temperature. After washing, the tissue was mounted on a slide using Fluoroshield (Sigma-Aldrich) mounting media. The slide was cured overnight and imaged using an upright confocal (Leica Stellaris). For inhibitor experiments, the tissue was incubated with the inhibitor, blebbistatin (100 μM) or CK666 (100 μM) in the complete medium for 2 h before making the incision wounds. After 2 h, the wounds were made and the tissue was incubated in the same media for the next 30 min followed by fixation and antibody staining.

### Live imaging of skin sample
After the skin was excised from the embryo, it was placed on a filter paper with the epidermis side up. The explant was incubated in a well containing the culture media (DMEM with 5% FBS and 1% pen–strep) and staining solution (SiR-actin and verapamil) overnight. Then, using a scalpel, incision wounds were made, and it was immediately transferred to a glass-bottom dish, such that the epidermis side was now facing the glass coverslip. Next, 20 μl of Matrigel was added and incubated for 10 min to help immobilize the explant. A confined chamber was placed on top of the filter and culture medium was added. The sample was then imaged using an inverted confocal.

### Immunofluorescence
Cells were fixed using a solution containing 3% formaldehyde and 0.1% glutaraldehyde diluted in 1× PBS for 10 min at room temperature. Following fixation, cells underwent three washes with 1× PBS. Subsequently, permeabilization was carried out using 0.25% Triton X-100 in PBS for 10 min at room temperature, followed by three quick washes with 1× PBS. Cells were then incubated with a blocking solution (consisting of 0.1% Triton X-100 in 1× PBS + 2% BSA) for 1 h at room temperature. Afterwards, cells were incubated with a primary antibody prepared in the blocking solution for 2 h at room temperature. Following the primary antibody incubation, cells were washed three times with 1× PBS and then incubated with the secondary antibody, 4,6-diamidino-2-phenylindole (DAPI) (1:1,000) and phalloidin (1:40) dilutions, all prepared in the blocking/staining solution, for 1 h at room temperature. Finally, the samples underwent two quick washes and a 5 min wash with 1× PBS before proceeding to imaging.

### Confocal microscopy
Fluorescence images were captured using a 60× oil objective (PlanApo N 60× oil, NA of 1.42, Olympus) and a 100× oil objective (UPlanSApo, 100×/1.40 oil), both mounted on an Olympus IX83 inverted microscope equipped with a scanning laser confocal head (Olympus FV3000). Time-lapse imaging of live samples was conducted in the live-cell chamber provided with the microscopy set-up. For the photoactivation experiment, a 60× oil objective (PlanApo N 60× oil, NA of 1.42, Olympus) mounted on an Olympus IX83 inverted microscope equipped with a scanning laser confocal head (Olympus FV3000) was utilized, supported by a live-cell imaging set-up. MDCK cells, either co-expressing photoactivable Rac1 and GFP-Sec61β or expressing photoactivable Rac1 and labelled with SiR-tubulin, were seeded on a glass-bottom dish in Opti-MEM medium with HEPES (20 mM). A region of interest (ROI) approximately 5 μm in diameter was irradiated with a 1% intensity 445 nm laser at 1,000 μs pixel$^{-1}$, followed by laser scanning microscopy imaging of the two channels (561 nm and 488 nm or 561 nm and 647 nm) at 0.1–0.5% intensity. This cycle of activation and imaging was repeated continuously for 15–20 min.

### SoRa
Fluorescence images were acquired using 100× oil objective (Apo TIRF, NA of 1.49, WD of 0.12) mounted on a Nikon inverted research microscope Eclipse Ti2-E with Yokogawa CSU-W1 SoRa unit. A further 2.8× magnification was used to acquire super-resolution images with

the SoRa unit. After the acquisition, images were subjected to Richardson–Lucy 3D deconvolution.

### Inhibition studies
For all inhibition studies, cells underwent pretreatment with the desired inhibitor concentration in complete DMEM for 1 h at 37 °C in a 5% $CO_2$ humidified incubator before lifting the PDMS stamps. Throughout the migration process, cells remained in complete media containing the inhibitor at the specified concentration for the required migration duration. To modulate actomyosin contractility, MDCK cells were treated with blebbistatin (Sigma), a myosin inhibitor, at a concentration of 50 μM. Arp2/3-inhibition was achieved using CK666 (Sigma) at a concentration of 50 μM. Calyculin A (Sigma), a phosphatase inhibitor, was used to enhance actomyosin contractility at a concentration of 20 nM. Nocodazole (Sigma) was utilized to disrupt microtubules in MDCK cells at a concentration of 10 μM. Following inhibitor treatment, cells were fixed and immunostained with the desired antibodies.

### Antibodies and plasmids
Source and dilution information for all primary and secondary antibodies used in immunofluorescence staining are given in Supplementary Table 1. Details of plasmids used in this study are listed in Supplementary Table 2 with their source.

### Image analysis
Image analysis was conducted using Fiji. To determine the fraction of ER tubules, Trainable Weka Segmentation was utilized to distinguish between tubules and sheets within the ER structures. To train the model, we selected key image processing features, including Gaussian blur, Sobel filter, Hessian, difference of Gaussians and membrane projections, which help distinguish ER tubules from sheets based on intensity, texture and edge detection. A sample image was manually annotated to define three structural categories or classes as follows: (1) thin peripheral ER tubules; (2) dense, bright ER sheets; and (3) background regions that included cytoplasmic spaces between ER structures and the extracellular area. The tool was iteratively trained by manually marking multiple areas in each category until the segmented output closely matched the original image. Once trained, the model was saved and applied to multiple images to ensure consistency and minimize user bias in segmentation. Following classification, the segmented image was converted to an 8-bit greyscale format for further quantification. Using thresholding, we extracted binary masks for tubules, sheets and total ER. The pixel area corresponding to each category was measured, providing quantitative data on the fraction of tubules per cell. This automated approach ensured an objective assessment of ER morphology across different experimental conditions. To assess the reliability of our Weka-based segmentation, we performed a validation by comparing automated segmentation outputs with manual annotations. A subset of images ($n = 30$) was manually segmented using Fiji's polygon selection tool to trace ER tubule and sheet regions. We then applied our trained Weka model to the same images and quantified the corresponding pixel areas. The fraction of ER tubules obtained by both methods showed a strong linear correlation (Pearson $r = 0.99$, $P < 0.001$), indicating that the automated classification reliably replicates manual assessment, demonstrating the robustness and accuracy of our segmentation pipeline.

The ER tubule fraction at the leading edge of the cell was computed by dividing the tubular ER area by the total ER area manually marked in the region at the migrating front of the cell. The same classifier was applied for ER segmentation in all inhibition and mutant studies. In the photoactivation experiment of Rac1, the ROI for Rac1 activation was selected for segmentation, while a similarly sized ROI at the nonactivated region within the cell served as a control.

The Directionality plugin in Fiji was used to quantify the orientation of focal adhesions. This plugin is used to quantify the preferred

orientation of the structures in the input image. The image of the front of the cell, labelled with anti-paxillin antibody, clearly marking focal adhesions, was cropped and provided to the plugin. The Fourier components method, 6 Nbins from 0° to 90°, was used. This computes and generates a histogram representing the frequency of structures (focal adhesions) at a particular orientation. Focal adhesions (FAs) aligned between 0° and 18° were considered parallel to the wound, while those aligned between 72° and 90° were deemed perpendicular. FAs not falling within these ranges were excluded from the analysis. The data over many cells at different curvatures were collected and averaged to finally plot the focal adhesion orientation. Microtubule orientation was also determined using the directionality plugin in Fiji, with 10 Nbins and angles ranging from 0° to 180°. An ROI at the front of the migrating cell in the α-tubulin channel was chosen, and directionality frequency was calculated. Curvature analysis was performed using the Fiji plugin kappa-curvature analysis. B-spline was given as the curvature input and the scale of individual images in micron/pixel was added to the plugin. Arcs were drawn manually over images and average curvature values were computed.

### Statistical analysis

Statistical analyses were performed using GraphPad Prism 9. Statistical significance was determined using an unpaired $t$-test with Welch's correction or a one-way analysis of variance (ANOVA), as specified in the corresponding figure legend. Scatter-bar plots were presented as mean ± s.e.m. $P$ values above 0.05 were considered not statistically significant. Quantification was based on data from a minimum of three independent biological replicates. Data distribution was assumed to be normal, but this was not formally tested. No statistical method was used to predetermine sample size, but our sample sizes are consistent with the standard of the field and are appropriate for the types of analyses performed. Samples were not allocated using randomization, as this study involved observational analysis of epithelial cells under defined experimental conditions. All groups were subjected to identical culture, treatment and imaging protocols to control for potential covariates, ensuring comparability across conditions. Data collection and analysis were not performed blind to the conditions of the experiments. No data points were excluded from the analyses.

### Reporting summary

Further information on research design is available in the Nature Portfolio Reporting Summary linked to this article.

### Data availability

Data supporting the findings of this work are available in this Article and its extended data figures and Supplementary Information. All other data supporting the findings of this study are available from the corresponding author on reasonable request. The time-lapse imaging data (otherwise available as supplementary video files) can be made available only upon request owing to the large file sizes and associated storage limitations. Source data are provided with this paper.

### Code availability

The code for mathematical modelling is available as an open-source download via GitHub at https://github.com/bkprdp/Curvature_Dependent_ER_Morphology.

### Acknowledgements

We thank R. Sharma (TIFR Hyderabad) for his assistance related to micropatterning and P. Gupta (TIFR Hyderabad) for western blot experiments. We thank T. N. Narayanan (TIFR Hyderabad) for providing the facility of plasma cleaner. We acknowledge H. Yu (National University of Singapore) and A. Akhmanova (Utrecht University) for generously providing plasmids used in this study and Y. Fujita (Kyoto University) for the WT MDCK cell line. This work was funded by Science and Engineering Research Board, India (now ANRF) (core research grant no. CRG/2021/003907 to T.D.) and by the BBSRC grant BB/V002708/1 (to F.S. and P. Keshavanarayana) and by a UKRI Future Leaders Fellowship (grant no. MR/T043571/1) (to F.S.). In addition, we acknowledge generous support from the Human Frontier Science Program research grant (no. RGP0007/2022) to T.D., and intramural funds at TIFR Hyderabad from the Department of Atomic Energy, India, under the project identification number RTI 4007 to S.R., D.M. and T.D.

### Author contributions

S.R. and T.D. formulated the project. S.R. performed most of the experiments and analyses. P. Keshavanarayana and F.S. performed the theoretical modelling. D.M. and S.B. performed mouse embryonic skin wound healing experiments. D.M. exclusively performed mouse embryonic skin live imaging. S.B. helped in the shape index analysis. P. Khuntia helped in replicating the main finding in EpH4 cells. B.T. modified the MATLAB code for MDE calculations. R.M. performed Imaris analysis on a few images. S.R., T.D., P. Keshavanarayana and F.S. wrote the paper. All authors approved the paper in its submitted form.

### Competing interests

The authors declare no competing interests.

### Additional information

**Extended data** is available for this paper at https://doi.org/10.1038/s41556-025-01729-3.

**Correspondence and requests for materials** should be addressed to Fabian Spill or Tamal Das.

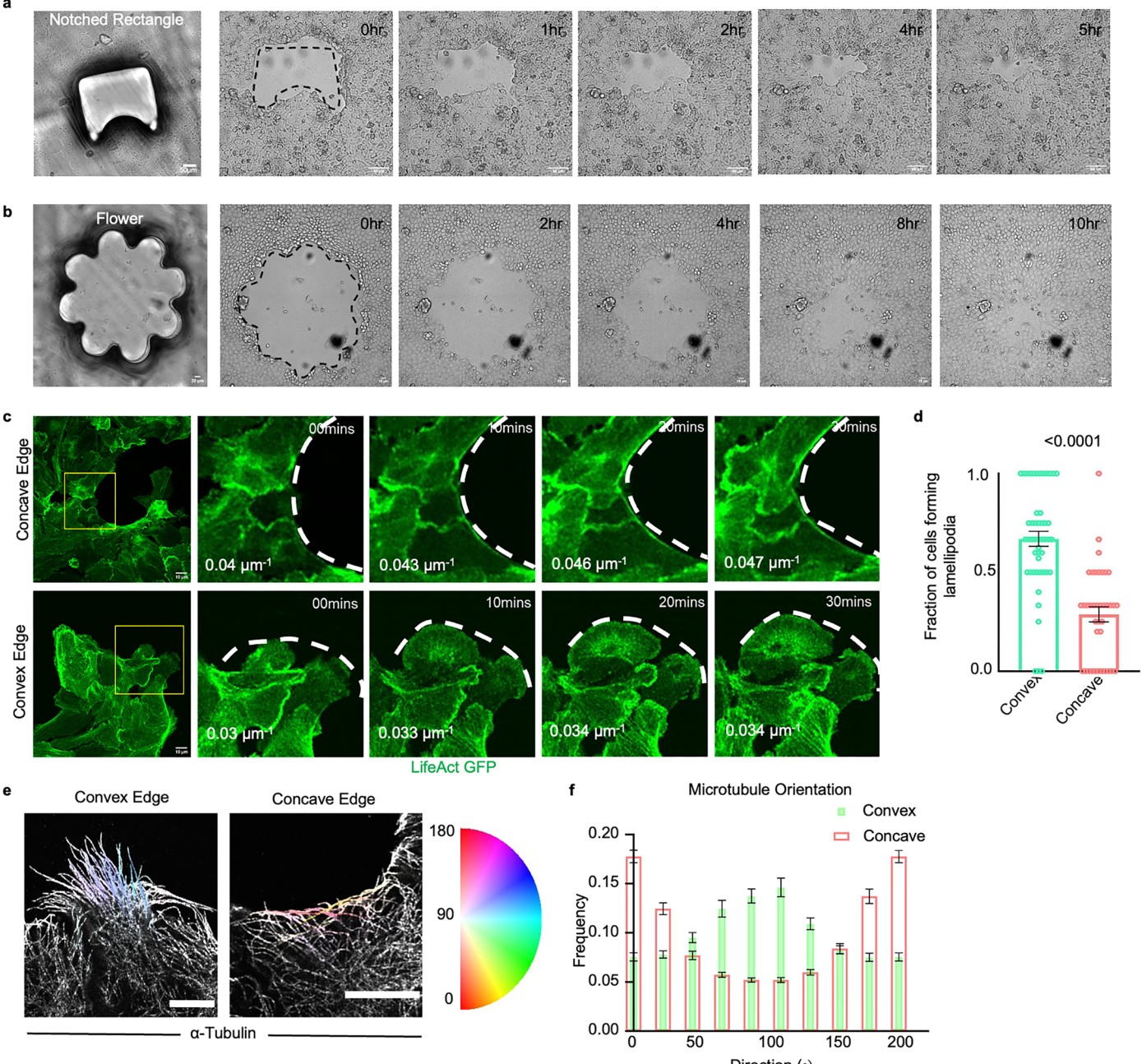

**Extended Data Fig. 1 | Migration of epithelial cells at different curvatures.**
**a**, Sample image of a notched rectangle pattern, and bright-field time lapse images of MDCK cells closing the micro-patterned gap; scale bar: 50 μm.
**b**, Sample image of a flower pattern, and bright field time lapse images of MDCK cells closing the micro patterned-flower gap; scale bar: 20 μm. **c**, Time-lapse images of LifeAct MDCK cells migrating at concave (upper panel) and convex (Lower panel), Zoomed insets showing actin remodeling and evolution of curvature in time, dashed white lines represent the edges; scale bar: 10 μm.

**d**, Quantification of the fraction of cells forming lamellipodia out of the total cells at the edge at two curvatures no. of RoIs= 49, 38 from left to right. **e**, Color wheel representation of the directionality of microtubules at convex (left) and concave (right) curvatures, color wheel ranging from 0 to 180 represents the angle of microtubules relative to the edge of the wound. Scale bar: 10 μm.
**f**, Quantification for directionality of microtubules at convex (green) and concave (pink) curvature n = 77 (convex), n = 74 (concave). Each experiment was repeated at least three independent times.

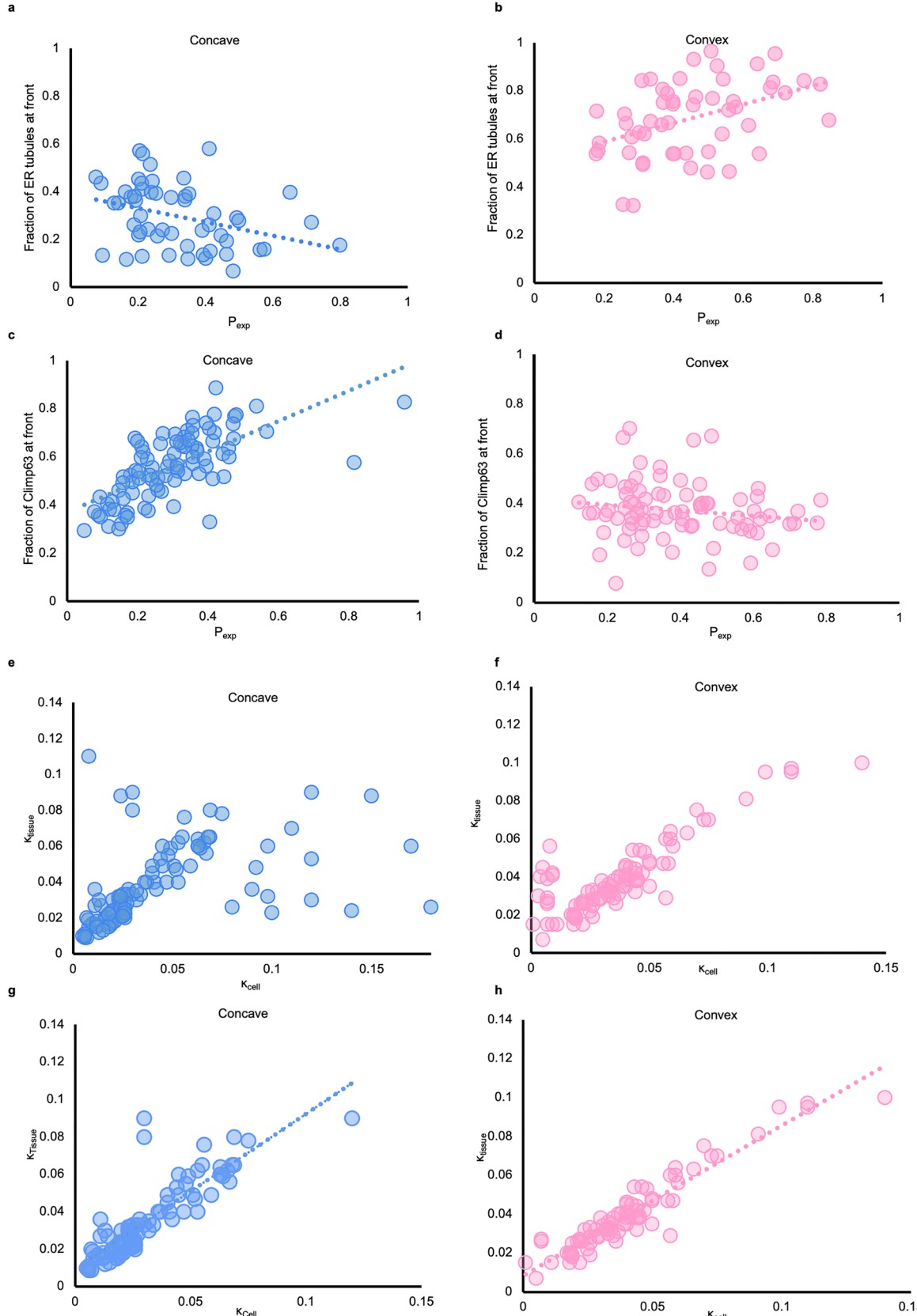

**Extended Data Fig. 2 | See next page for caption.**

**Extended Data Fig. 2 | Curvature analysis. a**, Quantification of Fraction of ER tubules at the front of cells at concave edge as a function of $P_{exp}$; dotted line-linear trend line, n = 55. **b**, Quantification of Fraction of ER tubules at the front of cells at convex edge as a function of $P_{exp}$; dotted line-linear trend line, n = 54. **c**, Quantification of Fraction of Climp63 at the front of cells at concave edge as a function of $P_{exp}$; dotted line-linear trend line, n = 92. **d**, Quantification of Fraction of Climp63 at the front of cells at convex edge as a function of $P_{exp}$; dotted line-linear trend line, n = 81. **e**, Quantification of $\kappa_{tissue}$ as a function of $\kappa_{cell}$ at concave edge n = 105 **f**, Quantification of $\kappa_{tissue}$ as a function of $\kappa_{cell}$ at convex edge n = 87. **g**, Quantification of $\kappa_{tissue}$ as a function of $\kappa_{cell}$ at concave edge considering only $P_{exp} > 20\%$ n = 85 dotted line-linear trend line. **h**, Quantification of $\kappa_{tissue}$ as a function of $\kappa_{cell}$ at convex edge considering only $P_{exp} > 20\%$, n = 77, dotted line-linear trend line. Each experiment was repeated at least three independent times.

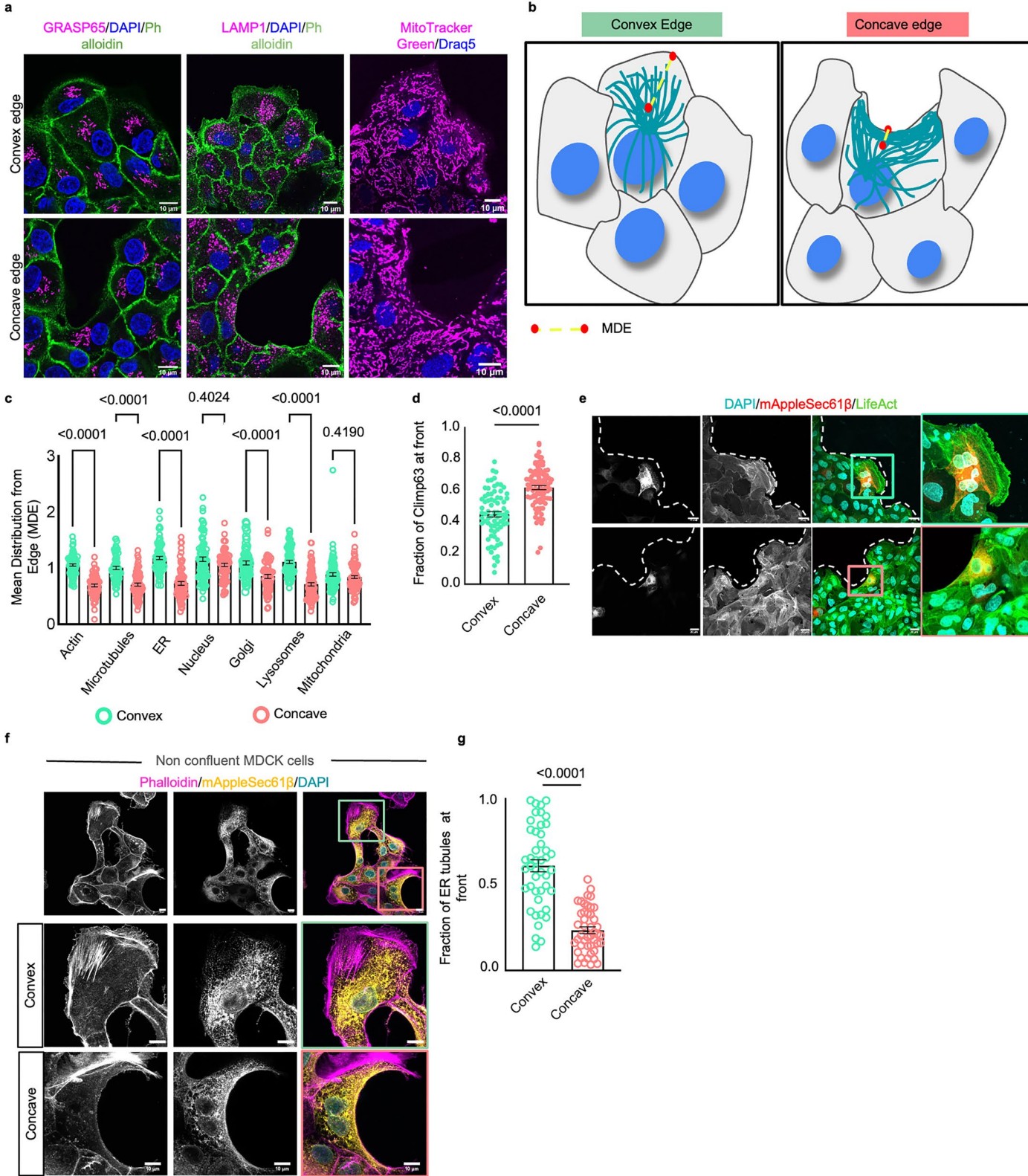

**Extended Data Fig. 3 | See next page for caption.**

**Extended Data Fig. 3 | Organelle morphology and polarity in response to wound geometry in MDCK. a**, Representative images of MDCK cells stained for Golgi, lysosome, and mitochondria at convex (upper panel) and concave (lower panel) curvature. MDCK cells stained for left panel- anti-GRASP65 (magenta, Golgi marker), phalloidin (green) and DAPI (blue); middle panel- anti-LAMP1 (magenta, lysosome marker), phalloidin (green) and DAPI (blue); right panel-cells labeled with Mitotracker green (magenta, mitochondria marker) and DRAQ5 (blue), Scale bar: 10 μm. **b**, Schematic representing MDE (mean distribution from the edge), the yellow dotted line represents the distance between the center of mass of the cellular entity and the edge of the cell; this distance normalized by the radius of the cell and is referred as MDE. **c**, Quantification of MDE for actin, microtubules, ER, Nucleus, Golgi, lysosomes, and mitochondria at convex (green) and concave (pink) curvatures. n = 74, 62, 62 69, 83, 75, 86, 71, 63, 61, 80, 78, 86, and 88 from left to right. **d**, Quantification of fraction of Climp63 at the front of cells at convex (green) and concave (pink) edge, n = 74, 97. **e**, Representative images of LifeAct MDCK cells transfected with mAppleSec61β seeded in flower shape wound, dotted lines represent migrating edge, scale bar: 20 μm. **f**, Representative images of non-confluent MDCK-mApple sec61β colonies stained with phalloidin(magenta) and DAPI (Cyan), zoomed insets show cells at convex (green) and concave (pink) edges, scale bar: 10 μm. **g**, Quantification of fraction of ER tubules at the front of cells at convex (green) and concave (pink) edges in non-confluent monolayers n = 45, 45. Data are mean ± s.e.m. One way ANOVA test (**c**), two-tailed t test (**d**, **g**). Each experiment was repeated at least three independent times.

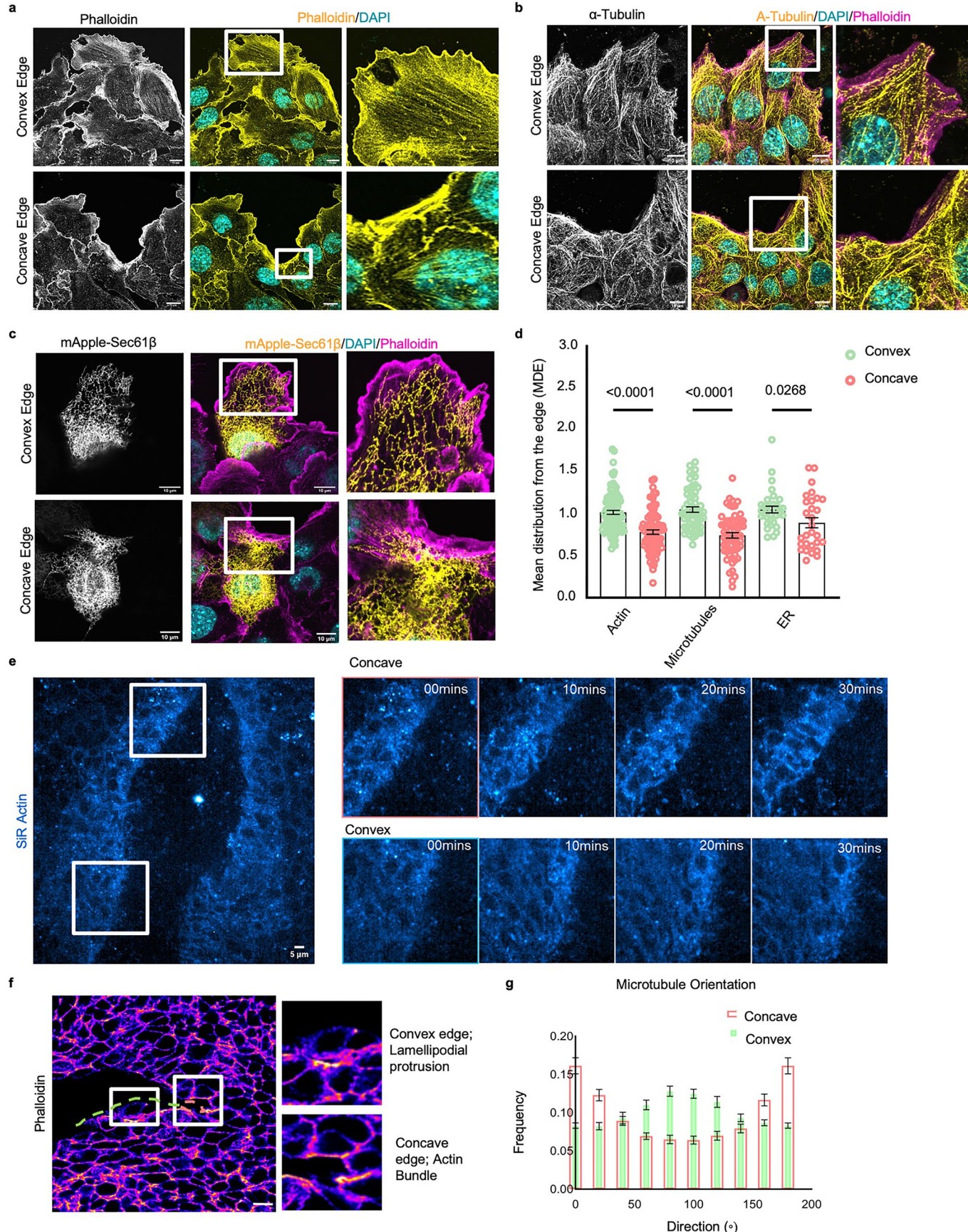

**Extended Data Fig. 4 | See next page for caption.**

**Extended Data Fig. 4 | Organelle polarity in response to wound geometry in EpH4 cells and mouse embryonic skin. a**–**c** Representative images of EpH4 cells stained with **a**, phalloidin (yellow) and DAPI (cyan) **b**, anti-α-tubulin (yellow), Phalloidin (magenta) and DAPI (cyan) **c**, DAPI (cyan), Phalloidin (magenta), and expressing mAppple-Sec61β (yellow) at convex (upper panel) and concave curvature (lower panel), front of the cells at the edge is enlarged and zoomed, scale bar: 10µm. **d**, Quantification of MDE for actin, microtubules, and ER at convex (green) and concave (pink) curvatures. n = 100, 90, 58, 61, 59, and 49 from left to right. **e**, Time lapse zoomed images of mouse embryonic skin wound, labelled with SiR actin (blue) at concave (upper panel) and convex (Lower panel) edges. **f**, Mouse embryonic wounded skin tissue stained with phalloidin showing spontaneous convex (zoomed upper panel) concave (zoomed lower panel) curvature. **g**, Quantification for the directionality of microtubules at convex and concave curvature in mouse embryonic skin wounds (n = 24 at convex and n = 29 at concave curvature). Data are mean ± s.e.m. One way ANOVA test (**d**). Each experiment was repeated at least three independent times.

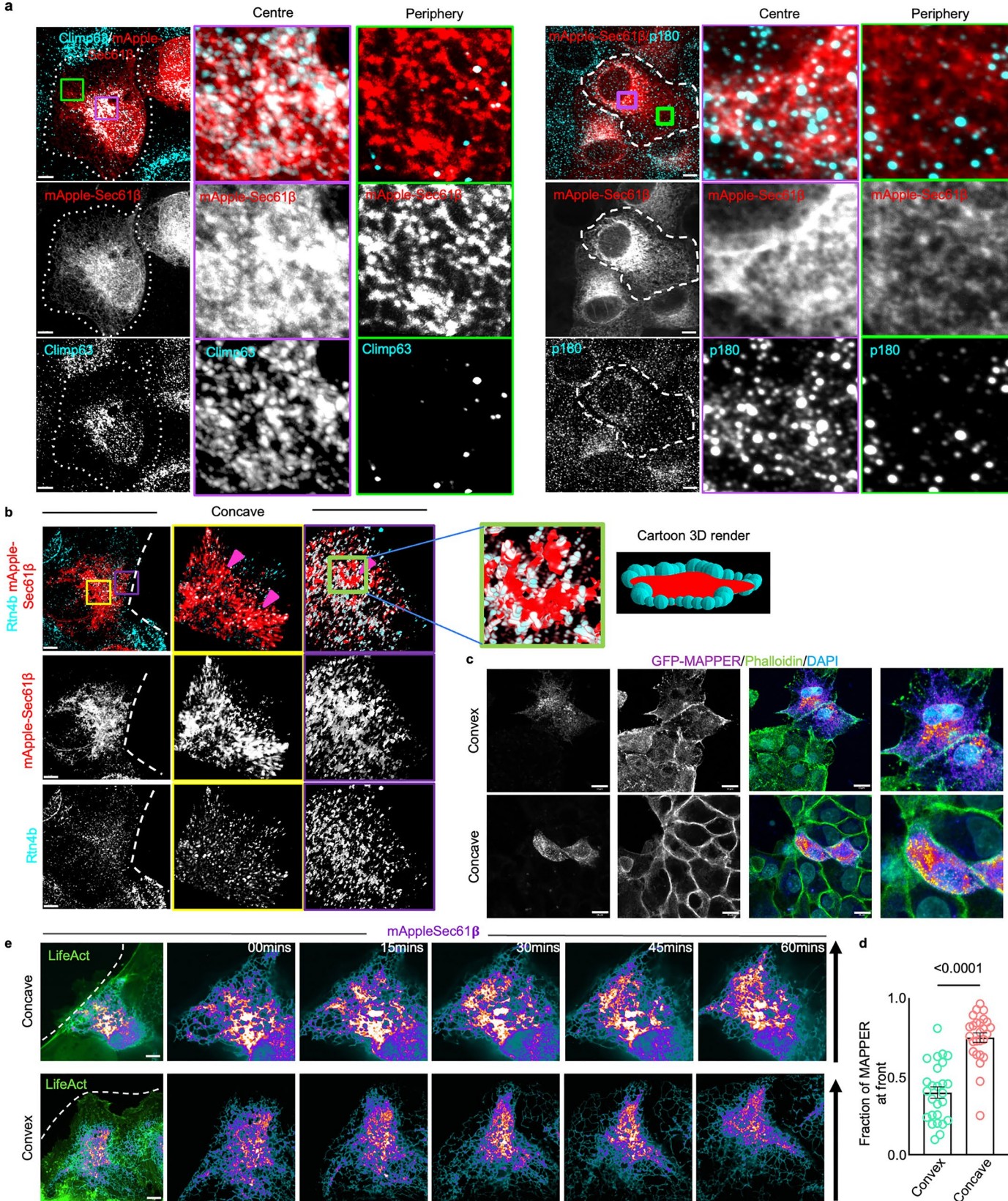

**Extended Data Fig. 5 | See next page for caption.**

**Extended Data Fig. 5 | Characterization of ER sheets and ER dynamics. a**, Representative images of MDCK cells transfected with mApple-Sec61β (red) in confluent monolayer stained with Left panel- Climp63 (cyan), zoomed insets represent enrichment of Climp63 at the cell center (purple) and at cell periphery (green); Right panel- Representative images of MDCK cells transfected with mApple-Sec61β (red) in confluent monolayer stained with p180 (cyan), zoomed insets represent enrichment of Climp63 at the cell center (purple) and at cell periphery (green) dotted boundary represents cell boundary, scale bar: 5 μm. **b**, Representative images of MDCK cells transfected with mApple-Sec61β (red) migrating at concave edge, stained with anti-Rtn4b antibody (cyan), inset (yellow and purple) represent zoomed region from the front of the cell, pink arrowhead represent Rtn4b (cyan) present around the ER sheet (red), dashed white line represent edge of the cell, scale bar: 3 μm. *Right most panel:* magnified super-resolution image of a flat ER sheet, surrounded by Rtn4b proteins, and a representative cartoon 3D render. **c**, Representative images of MDCK cells expressing GFP-MAPPER migrating at convex (upper panel) and concave (lower panel) curvature stained with phalloidin (green) and DAPI (cyan), scale bar: 10 μm. **d**, Quantification for GFP mapper intensity at the front at convex and concave curvatures n = 26 at each curve **e**, Time lapse super-resolution SoRa images of LifeAct (green) MDCK cells expressing mApple-Sec61β migrating at concave (upper panel) and Convex (lower panel) edges, black arrow represents direction of migration, dashed line represents migrating edge scale bar: 5 μm., Data are mean ± s.e.m. two-tailed t test (**d**). Each experiment was repeated at least three independent times.

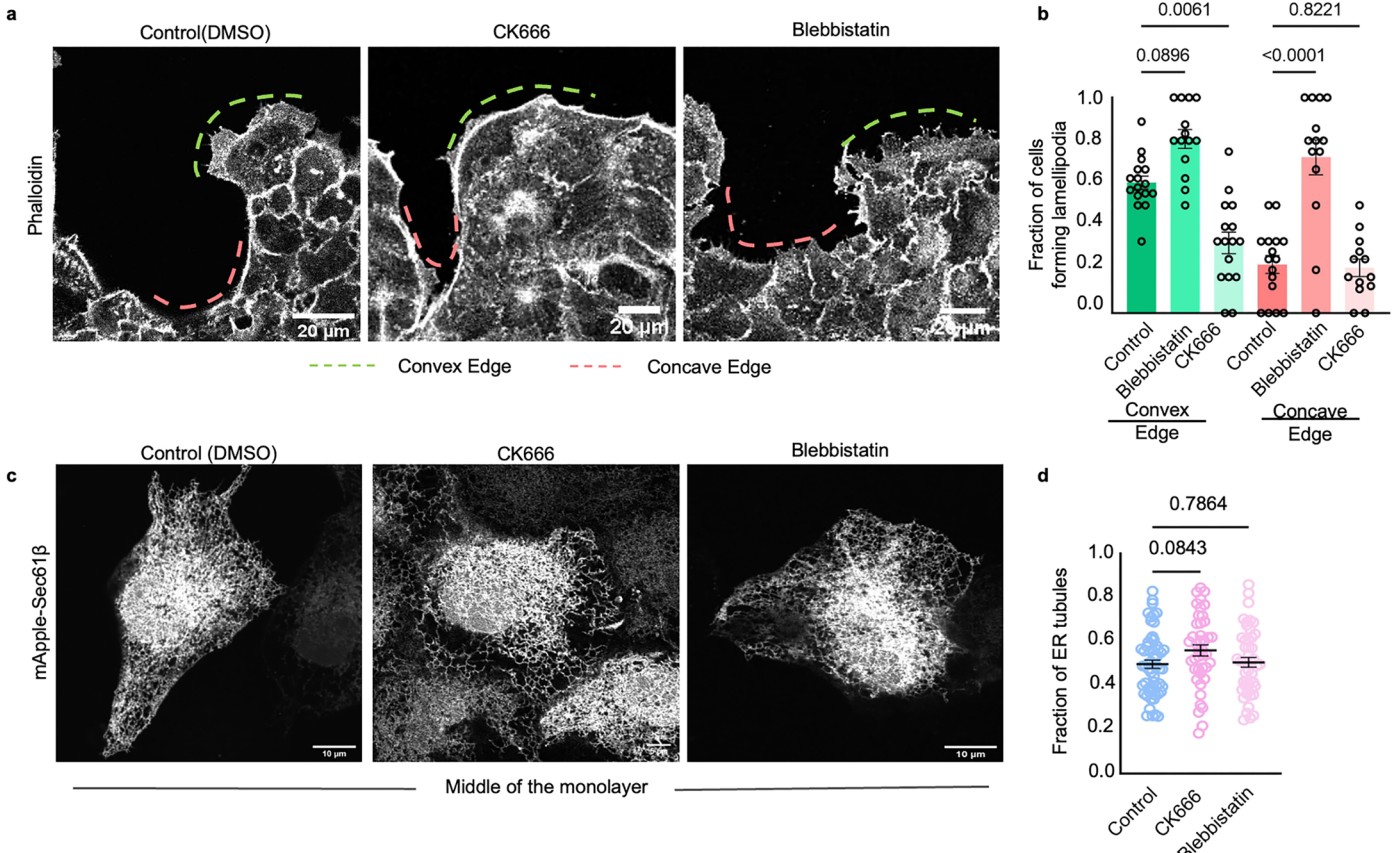

**Extended Data Fig. 6 | Effect of Blebbistatin and CK666. a**, Representative images of MDCK cells treated with DMSO (control, left), CK666 (middle), and Blebbistatin (right) stained with phalloidin (grey), pink dashed lines mark concave curvatures and green dashed lines mark convex curvatures, scale bar: 20 μm. **b**, Quantification for fraction of cells forming lamellipodia at convex or concave curvatures in cells treated with DMSO, CK666, or Blebbistatin (no of ROIs=16, 14, 16, 16, 14, and 13 from left to right, each ROI). **c**, Representative images of MDCK cells expressing mApple-Sec61β (grey) treated with DMSO (left), CK666 (middle), or Blebbistatin (right) in the middle of confluent monolayers, scale bar: 10 μm. **d**, Quantification of fraction of ER tubules in the cells in the middle of the monolayer treated with DMSO, CK666, or Blebbistatin; n = 59, 42, and 47 from left to right. Data are mean ± s.e.m, One way ANOVA test. Each experiment was repeated at least three independent times.

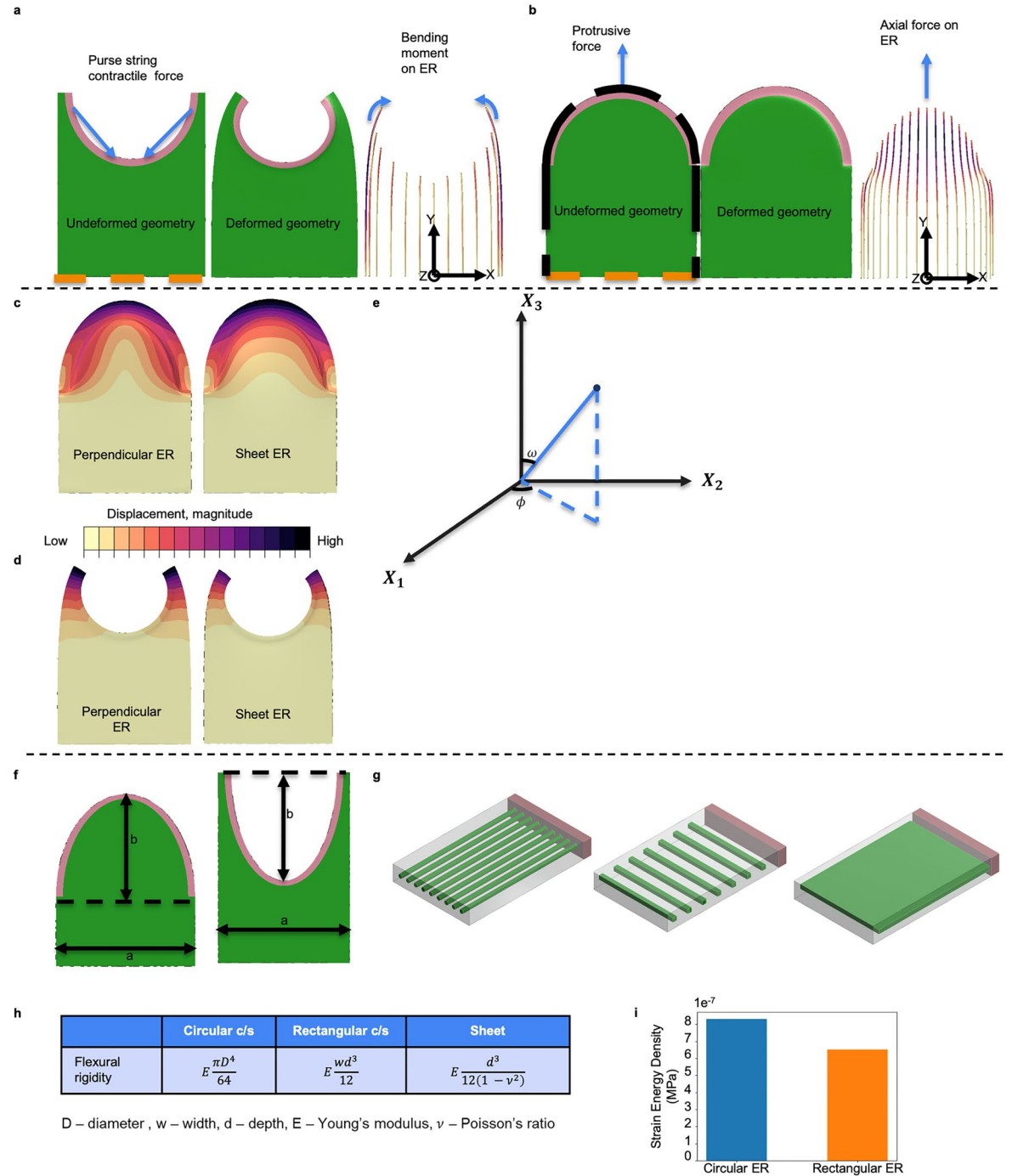

**Extended Data Fig. 7 | Mathematical model for ER morphologies at different curvature. a**, Schematic representing the purse-string contractile force acting on the cell with negative curvature on undeformed geometry (left). Orange dashed line indicate where boundary conditions are applied. Resulting deformation of cell (centre) and perpendicular ER (right). **b**, Schematic representing the protrusive force acting on the cell with positive curvature on undeformed geometry (left). Orange and black dashed lines indicate where boundary conditions are applied. Resulting deformation of cell (centre) and the perpendicular ER (right). **c**, Displacement map of cell with positive curvature during protrusion with perpendicular ER (left) and sheet ER (right). **d**, Displacement map of cell with negative curvature during contraction with perpendicular ER (left) and sheet ER (right). **e**, Representation of a scalar in (ω, φ) coordinates superimposed on Cartesian coordinate system. **f**, Schematic representing the notations used for evaluating positive (left) and negative (right) curvatures. **g**, 3D cad models of cells with perpendicular ER (left), parallel ER (centre) and sheet ER (right). **h**, Table elucidating formula to evaluate flexural rigidity of different cross sections. **i**, Strain energy density of cell with negative curvature during contraction with parallel ER and varying cross sections.

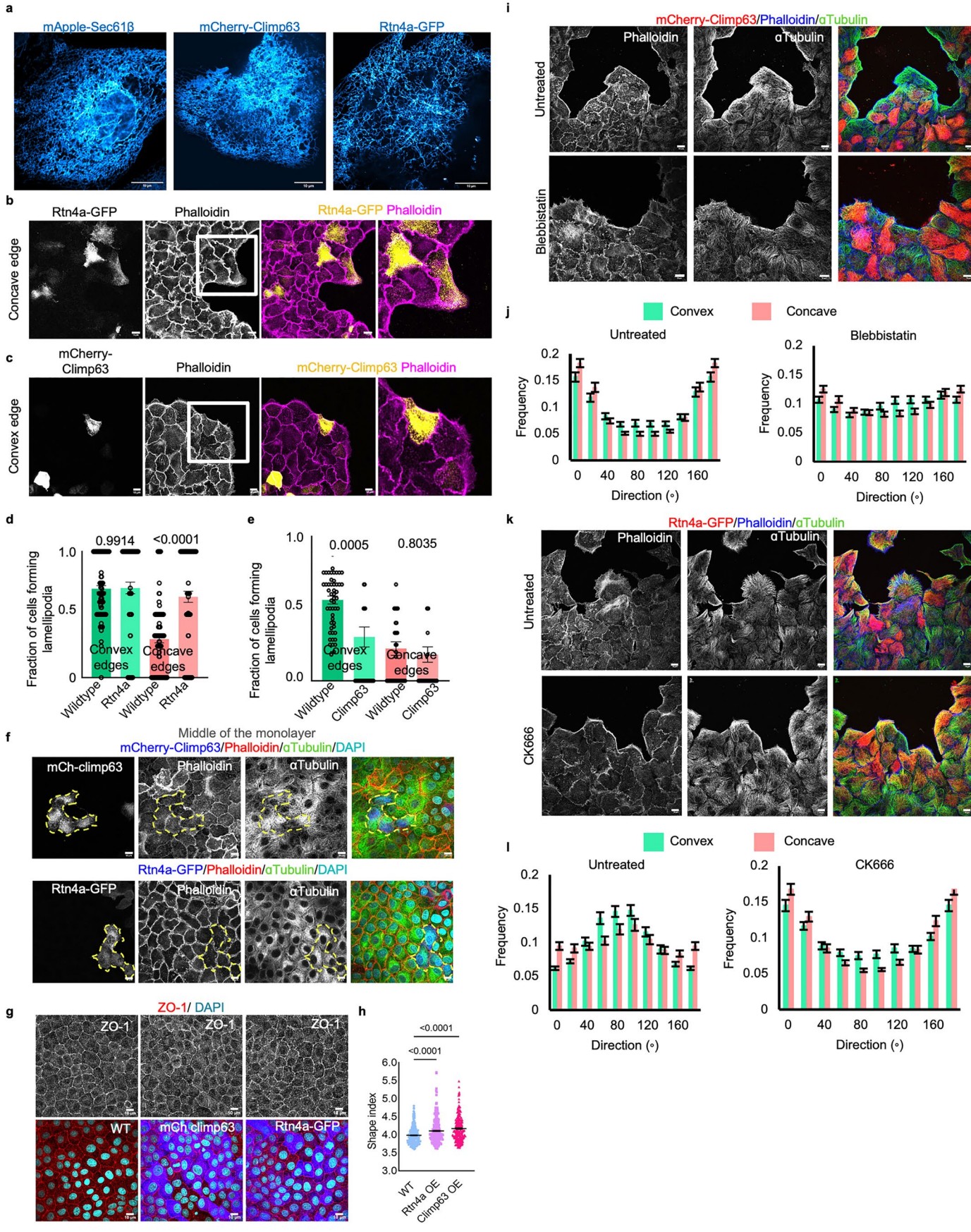

**Extended Data Fig. 8 | See next page for caption.**

**Extended Data Fig. 8 | ER structural changes affect the mode of migration.**
**a**, Representative super-resolution (SoRa) images of cells expressing mApple-Sec61β (left), mCherry-Climp63 (middle) and Rtn4a-GFP (right); scale bar: 10μm. **b**, Representative MDCK cells migrating at concave edge heterogeneously expressing Rtn4a-GFP (yellow), stained with phalloidin (magenta), zoomed inset shows Rtn4a-GFP expressing cell forming lamellipodia at concave edge, scale bar: 10μm. **c**, Representative MDCK cells migrating at convex edge heterogeneously expressing mCherry-Climp63 (yellow), stained with phalloidin (magenta) zoomed inset shows mCherry-Climp63 expressing cell forming actin bundle at convex edge, scale bar: 10μm. **d**, Quantification of fraction of MDCK WT and Rtn4a over-expressing cells forming lamellipodia at convex and concave edges no. of ROIs= 90, 76, 96, and 89. **e**, Quantification of fraction of MDCK WT and Climp63 over-expressing cells forming lamellipodia at convex and concave edges no. of ROIs= 53, 39, 41, and 44. **f**, Representative images of MDCK cells expressing mCherry-Climp63 (blue, upper panel), Rtn4a-GFP (blue, lower panel) in confluent monolayer, stained with anti-αtubulin antibody (green), Phalloidin (red) and DAPI(cyan), yellow dashed line marks the transfected colonies, scale bar: 10 μm. **g**, Representative images of MDCK WT (left), MDCK overexpressing

Climp63 (blue, middle), MDCK overexpressing Rtn4a (blue, right) in confluent monolayer, stained with anti ZO-1 antibody (red) and DAPI (Cyan), scale bar: 10 μm. **h**, Quantification of shape index of MDCK WT, Climp63 OE and Rtn4a OE, n = 268, 281, and 249. **i**, Representative images of MDCK overexpressing mCherry-Climp63 (red) seeded in micro patterned gaps untreated (upper panel) and Blebbistatin-treated (lower panel) stained for anti-αtubulin antibody (green) and Phalloidin (blue), scale bar: 10 μm. **j**, Quantification for directionality of microtubules, in Climp63 OE cells in untreated (left Convex n = 89, Concave n = 86) vs Blebbistatin-treated (right convex n = 85, concave n = 86) conditions at convex (green) and concave (pink) curvatures. **k**, Representative images of MDCK overexpressing Rtn4a-GFP (red) seeded in micro patterned gaps and untreated (upper panel) and CK666 treated (lower panel) stained for anti-αtubulin antibody (green) and Phalloidin (blue), scale bar: 10 μm. **l**, Quantification for directionality of microtubules, in Rtn4a-GFP OE cells in untreated (left Convex n = 77, Concave n = 86) vs CK666 (right convex n = 79, concave n = 78) treated conditions at convex (green) and concave(pink) curvatures. Data are mean ± s.e.m. One way ANOVA test (**d**, **e**, **h**). Each experiment was repeated at least three independent times.

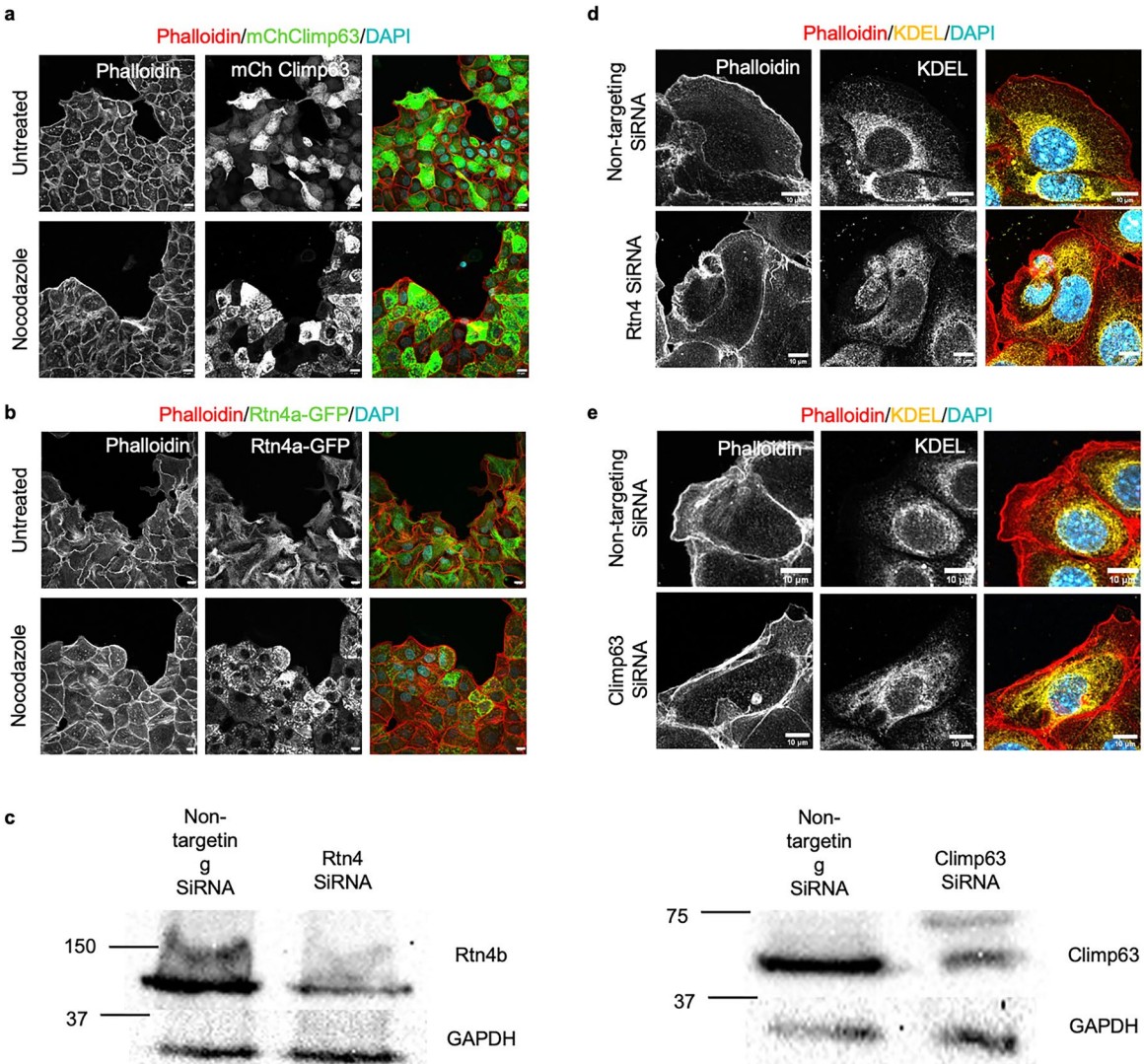

**Extended Data Fig. 9 | ER morphology alteration by ER structural proteins knockdown. a**, Representative images of MDCK cells overexpressing mCherry-Climp63 (green) untreated (upper panel) and treated with Nocodazole (lower panel) stained with phalloidin (red) and DAPI (cyan), scale bar: 10 μm. **b**, Representative images of MDCK cells overexpressing Rtn4a-GFP (green) untreated (upper panel) and treated with Nocodazole (lower panel) stained with phalloidin (red) and DAPI (cyan), scale bar: 10 μm. **c**, Representative western blot for Left- EpH4 cells treated with either non targeting siRNA or Rtn4 siRNA probed

for Rtn4b and GAPDH; Right- EpH4 cells treated for either non targeting SiRNA or Climp63 siRNA probed for Climp63 and GAPDH. **d**, Representative images of EpH4 cells treated with non-targeting siRNA (upper panel) and Rtn4 SiRNA (lower panel) stained with phalloidin (red) and DAPI (cyan) and anti-KDEL antibody (yellow), scale bar: 10 μm. **e**, Representative images of EpH4 cells treated with non-targeting siRNA (upper panel) and Climp63 siRNA (lower panel) stained with phalloidin (red) and DAPI (cyan) and anti-KDEL antibody (yellow), scale bar: 10 μm. Imaging experiments were repeated at least three independent times.

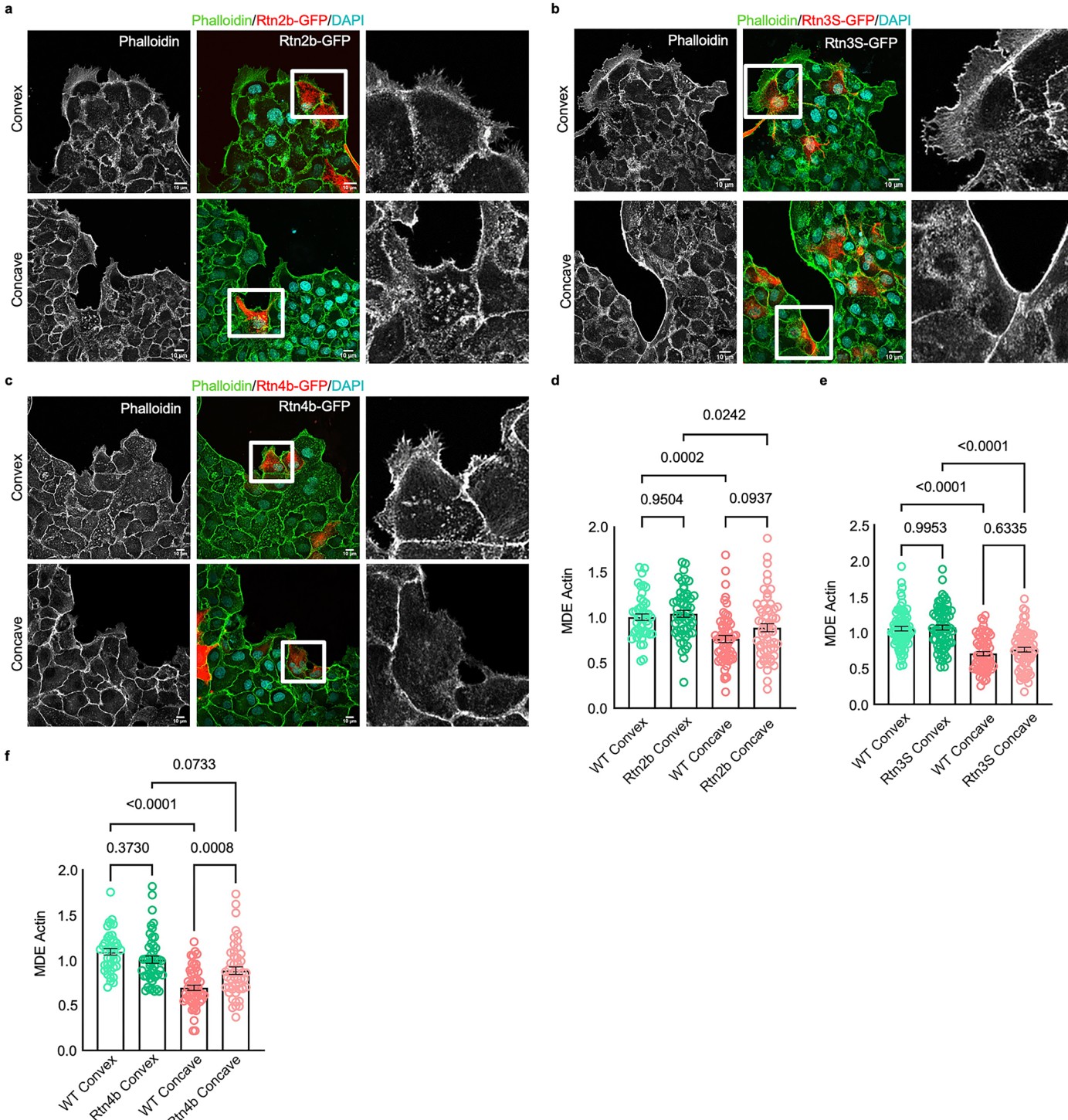

**Extended Data Fig. 10 | Overexpression of reticulons. a**, Representative images of MDCK cells transiently overexpressing Rtn2b-GFP (red) and migrating at convex (upper) and concave (lower) curvature stained with phalloidin (green) and DAPI (cyan), scale bar: 10 μm. **b**, Representative images of MDCK cells transiently overexpressing Rtn3S-GFP (red) and migrating at convex (upper) and concave (lower) curvature stained with phalloidin (green) and DAPI (cyan), scale bar: 10 μm. **c**, Representative images of MDCK cells transiently overexpressing Rtn4b-GFP (red) and migrating at convex (upper) and concave (lower) curvature stained with phalloidin (green) and DAPI (cyan), scale bar: 10 μm. **d**, Quantification of MDE actin for WT and Rtn2b overexpressing cells at convex (green) and concave (pink) curvature n = 46, 52, 56, and 59 from left to right. **e**, Quantification of MDE actin for WT and Rtn3S overexpressing cells at convex (green) and concave (pink) curvature n = 70, 71, 64, and 70 from left to right. **f**, Quantification of MDE actin for WT and Rtn4b overexpressing cells at convex (green) and concave (pink) curvature n = 40, 47, 53, and 50 from left to right. One way ANOVA test (**d**–**f**). Each experiment was repeated at least three independent times.

# Reporting Summary

## Statistics

For all statistical analyses, confirm that the following items are present in the figure legend, table legend, main text, or Methods section.

| n/a | Confirmed | |
|---|---|---|
| ☐ | ☒ | The exact sample size (*n*) for each experimental group/condition, given as a discrete number and unit of measurement |
| ☐ | ☒ | A statement on whether measurements were taken from distinct samples or whether the same sample was measured repeatedly |
| ☐ | ☒ | The statistical test(s) used AND whether they are one- or two-sided<br>*Only common tests should be described solely by name; describe more complex techniques in the Methods section.* |
| ☒ | ☐ | A description of all covariates tested |
| ☐ | ☒ | A description of any assumptions or corrections, such as tests of normality and adjustment for multiple comparisons |
| ☐ | ☒ | A full description of the statistical parameters including central tendency (e.g. means) or other basic estimates (e.g. regression coefficient) AND variation (e.g. standard deviation) or associated estimates of uncertainty (e.g. confidence intervals) |
| ☐ | ☒ | For null hypothesis testing, the test statistic (e.g. *F*, *t*, *r*) with confidence intervals, effect sizes, degrees of freedom and *P* value noted<br>*Give P values as exact values whenever suitable.* |
| ☒ | ☐ | For Bayesian analysis, information on the choice of priors and Markov chain Monte Carlo settings |
| ☒ | ☐ | For hierarchical and complex designs, identification of the appropriate level for tests and full reporting of outcomes |
| ☒ | ☐ | Estimates of effect sizes (e.g. Cohen's *d*, Pearson's *r*), indicating how they were calculated |

*Our web collection on statistics for biologists contains articles on many of the points above.*

## Software and code

Policy information about availability of computer code

| Data collection | Fluorescence images were captured using a 60X oil objective (PlanApo N 60X oil NA=1.42, Olympus) and 100X oil objective (UPlanSApo, 100X/1.4 oil) both mounted on Olympus IX83 inverted microscope equipped with scanning laser confocal head (Olympus FV3000). Super-resolution images were acquired using 100X oil objective ( Apo TIRF, NA=1.49, WD= 0.12) mounted on a nikon inverted research microscope eclipse Ti2 E Yokogawa with CSU - W1 SoRa unit . |
|---|---|
| Data analysis | Image brightness, contrast adjustments, overlay, directionality and segmentation was performed in FIJI (ImageJ) FIJI 1.53q Java 1.8.0_172 (64-bit) and is described in the methods section. MDE analysis was done using a code written in MATLAB (Mathworks, v R2022b). Shape index was calculated using CellPose 3.0 for segmentation and MATLAB for quantification. Statistical analysis was performed using GaphPadPrism v.9.5.0 and Microsoft Excel. The code for mathematical modeling is available as an open-source download from the GitHub page: https://github.com/bkprdp/Curvature_Dependent_ER_Morphology |

For manuscripts utilizing custom algorithms or software that are central to the research but not yet described in published literature, software must be made available to editors and reviewers. We strongly encourage code deposition in a community repository (e.g. GitHub). See the Nature Portfolio guidelines for submitting code & software for further information.

# Data

Policy information about availability of data

All manuscripts must include a data availability statement. This statement should provide the following information, where applicable:
- Accession codes, unique identifiers, or web links for publicly available datasets
- A description of any restrictions on data availability
- For clinical datasets or third party data, please ensure that the statement adheres to our policy

Data supporting the findings of this work are available in this study and its Extended data figures and the Supplementary Information. Source data are provided with this paper. All other data supporting the findings of this study are available from the corresponding author on reasonable request. The time lapse imaging data (otherwise available as supplementary video files) can only be made available upon request due to the large file sizes and associated storage limitations.

# Research involving human participants, their data, or biological material

Policy information about studies with human participants or human data. See also policy information about sex, gender (identity/presentation), and sexual orientation and race, ethnicity and racism.

| | |
|---|---|
| Reporting on sex and gender | Our research foes not cover this field of work. |
| Reporting on race, ethnicity, or other socially relevant groupings | Our research does not cover this field of work. |
| Population characteristics | Our research does not cover this field of work. |
| Recruitment | Our research does not cover this field of work. |
| Ethics oversight | Our research does not cover this field of work. |

Note that full information on the approval of the study protocol must also be provided in the manuscript.

# Field-specific reporting

Please select the one below that is the best fit for your research. If you are not sure, read the appropriate sections before making your selection.

☒ Life sciences  ☐ Behavioural & social sciences  ☐ Ecological, evolutionary & environmental sciences

For a reference copy of the document with all sections, see nature.com/documents/nr-reporting-summary-flat.pdf

# Life sciences study design

All studies must disclose on these points even when the disclosure is negative.

| | |
|---|---|
| Sample size | No statistical methods were used to pre-determine sample size. However, all experiments were designed to ensure robust and reproducible results, with a minimum of 25 cells analyzed per sample in nearly all cases. A slight exception is the photoactivation experiment in Fig. 3g, where the number of cells analyzed was lower due to the technically demanding nature of the assay—each cell required prolonged high-resolution imaging, limiting the throughput. Despite this, the observed effects were consistent across replicates. Overall, our sampling strategy is appropriate for single-cell resolution imaging and subcellular structure quantification, where each cell provides a rich and independent dataset. The sample sizes are in line with previous studies in the field (for example, PMIDs: 34912111, 38867038). Exact n values are reported in the figure legends. |
| Data exclusions | No data was excluded for analysis. |
| Replication | All experiments with quantifications were repeated atleast thrice. All experiments with display representations were repeated at least thrice. All attempts at replication were successful. |
| Randomization | Culture dishes consisted of 80-100 stamp wound with convex and concave curvatures. Images were captured randomly amongst these at different curvature margins. No sub-sampling was done. Cells were measured randomly under different treatment conditions. Samples were not allocated using randomization, as this study involved observational analysis of epithelial cells under defined experimental conditions. All groups were subjected to identical culture, treatment, and imaging protocols to control for potential covariates, ensuring comparability across conditions. |
| Blinding | Blinding was considered not to be necessary as the effects observed were clear. The module to segment ER on the trainable Weka segmentation was kept the same for all images. Data collection and analysis were not performed blind to the conditions of the experiments. |

# Reporting for specific materials, systems and methods

We require information from authors about some types of materials, experimental systems and methods used in many studies. Here, indicate whether each material, system or method listed is relevant to your study. If you are not sure if a list item applies to your research, read the appropriate section before selecting a response.

## Materials & experimental systems

| n/a | Involved in the study |
|-----|----------------------|
| ☐ | ☒ Antibodies |
| ☐ | ☒ Eukaryotic cell lines |
| ☒ | ☐ Palaeontology and archaeology |
| ☐ | ☒ Animals and other organisms |
| ☒ | ☐ Clinical data |
| ☒ | ☐ Dual use research of concern |
| ☒ | ☐ Plants |

## Methods

| n/a | Involved in the study |
|-----|----------------------|
| ☒ | ☐ ChIP-seq |
| ☒ | ☐ Flow cytometry |
| ☒ | ☐ MRI-based neuroimaging |

## Antibodies

| | |
|---|---|
| Antibodies used | All the antibodies used in the study have been described in supplementary table 1. |
| Validation | All primary antibodies and fluorophores used in this study were either commercially validated for immunofluorescence (IF) or western blotting (WB), and further internally validated in our laboratory based on expected subcellular localization patterns or band sizes. The commercial information about antibody validation is available on the suppliers homepage. Anti-α-tubulin (CST, 3873S) and anti-LAMP1 (Abcam, Ab24170) were validated for IF and showed characteristic microtubule and lysosomal localization, respectively. Anti-GRASP65 (Thermo Fisher Scientific, MA5-25148) was validated for IF and showed expected Golgi localization. Anti-CKAP4 (Climp63) (Proteintech, 16686-1-AP) and anti-Nogo B (Rtn4b) (Thermo Fisher Scientific, MA5-32763) were validated for both IF and WB; both showed appropriate ER-like distribution and single bands at expected sizes. Anti-RRBP1 (p180) (Allied Scientific Products, A80974), anti-KDEL (Merck, 420400), and anti-Sec61β (Thermo Fisher Scientific, PA3-015) were all validated for IF and showed consistent ER localization. Anti-ZO1 (CST, 8193S) and anti-Paxillin (Abcam, Ab32084) were validated for IF and localized specifically to tight junctions and focal adhesions, respectively. Anti-GAPDH (CST, 97166S), used for WB, showed a single band at the expected molecular weight. All antibody dilutions and experimental conditions are detailed in the supplementary table and respective figure legends. |

## Eukaryotic cell lines

Policy information about cell lines and Sex and Gender in Research

| | |
|---|---|
| Cell line source(s) | MDCK-wild type cell line was a gift from Yasuyuki Fujita. These cells were originally sourced from The European Collection of Authenticated Cell Cultures (ECACC, 85011435). EpH4-Ev was obtained from ATCC (CRL-3063). |
| Authentication | Original cell lines MDCK-Wild type and EpH4-Ev were authenticated at the source. |
| Mycoplasma contamination | Cell lines used in the study was free of mycoplasma contamination |
| Commonly misidentified lines (See ICLAC register) | No commonly misidentified cell lines were used. |

## Animals and other research organisms

Policy information about studies involving animals; ARRIVE guidelines recommended for reporting animal research, and Sex and Gender in Research

| | |
|---|---|
| Laboratory animals | Species- Mus musculus, Strain- C57/6J, age- Adult females- 8-12 weeks old, Embryos - E13.5-E16.5. All mice were housed in specific and opportunistic pathogen free conditions at an ambient temperature of 19-23 degree C and humidity of 40-60% with a 12:12 hour light dark cycle prior to use. |
| Wild animals | This study did not involve wild animals |
| Reporting on sex | The results are independent of the sex of the animals. The development and physiology of the embryonic epidermal tissue is not known to be different in different sexes, therefore sex of the animal was considered a parameter in the study. |
| Field-collected samples | Study did not involve samples collected from the field. |
| Ethics oversight | Protocol was approved by the Institutional Animal Ethics committee (IAEC), TIFR Hyderabad on behalf of Committee for the Purpose of Control and Supervision of experiments on Animals (CPCSEA), India |

Note that full information on the approval of the study protocol must also be provided in the manuscript.

## Plants

| | |
|---|---|
| Seed stocks | Our research does not cover this field of work |
| Novel plant genotypes | Our Research does not cover this field of work |
| Authentication | Our research does not cover this field of work. |

