## [Peer Review File · Nature Cell Biology]

Edge curvature drives endoplasmic reticulum reorganization and dictates epithelial migration mode

Corresponding Author: Professor Tamal Das

Version 0:

Decision Letter:

Dear Tamal,

Thank you for your interest in submitting your work to Nature Cell Biology.

I have discussed the information you provided with my colleagues, and we think that the study sounds interesting and could be appropriate for our journal. However, given the limited information provided, we would like to evaluate the complete manuscript before deciding whether to formally review it.

Please use this link to submit the complete manuscript:

Link Redacted

Please feel free to contact me if you have any questions and thank you very much for thinking of NCB for your work.

Kind regards,

Melina

Melina Casadio, PhD
Senior Editor, Nature Cell Biology
ORCID ID: <https://orcid.org/0000-0003-2389-2243>

Version 1:

Decision Letter:

*Please delete the link to your author homepage if you wish to forward this email to co-authors.

Dear Professor Das,

Thank you again for submitting your manuscript, "Curvature-dependent morphological reorganization of the endoplasmic reticulum determines the mode of epithelial migration", to Nature Cell Biology. I sincerely apologize for the delay in communicating our decision to you. Your manuscript has now been seen by 3 referees, who are experts in mechanobiology, mathematical modelling (Referee #1); ER (Referee #2); and biophysics, cell migration (Referee #3). As you will see from their comments (attached below), they found this work of potential interest but have raised substantial concerns, which in our view would need to be addressed with considerable revisions before we can consider publication in Nature Cell Biology.

Nature Cell Biology editors discuss the referee reports in detail within the editorial team, including the chief editor, to identify key referee points that should be addressed with priority, and requests that are overruled as being beyond the scope of the current study. To guide the scope of the revisions, I have listed these points below. Our standard revision period is six months and we are committed to providing a fair and constructive peer-review process, so please feel free to contact me if you would like to discuss any of the referee comments further or if you anticipate issues addressing the reviews.

In particular, we think it would be essential to strengthen the links between curvature, ER morphology, and the mode of epithelial migration, even without more insights into the underlying mechanisms (which Rev#1 requested). We stress that, beyond the request for mechanistic insight, all other reviewers' concerns are significant and would need to be addressed

thoroughly experimentally and reconsideration of the study for this journal and re-engagement of referees would depend on strength of these revisions. We recommend that efforts be dedicated in revision to address the following points:

1-- The reviewers all felt that additional methods and experimental tests were needed to show changes in ER morphology and curvature drive changes in cell migration and to clarify the curvature analyses:

Rev#1 points #1, 2, 3, 3, 4

Rev#2 points #1-2

Rev#3 points #1-2-3-4-5-6-7-12

2-- Please also address their concerns about the modelling studies to strengthen the interpretation and conclusions:

Rev#1 point #5

Rev#3 points #8, 10

3-- All other referee concerns pertaining to strengthening existing data, providing controls, methodological details, clarifications and textual changes, should also be addressed.

4-- Finally, please pay close attention to our guidelines on statistical and methodological reporting (listed below), as failure to do so may delay the reconsideration of the revised manuscript. In particular, please provide:

We would be happy to consider a revised manuscript that would satisfactorily address these points, unless a similar paper is published elsewhere or is accepted for publication in Nature Cell Biology in the meantime.

- ensure that it conforms to our format instructions and publication policies (see below and <https://www.nature.com/nature/for-authors>).

- provide a point-by-point rebuttal to the full referee reports verbatim, as provided at the end of this letter.

- provide the completed Reporting Summary (found here <https://www.nature.com/documents/nr-reporting-summary.pdf>). This is essential for reconsideration of the manuscript will be available to editors and referees in the event of peer review. For more information see <http://www.nature.com/authors/policies/availability.html> or contact me.

-- that unprocessed scans are clearly labelled and match the gels and western blots presented in figures.

-- that control panels for gels and western blots are appropriately described as loading on sample processing controls

-- all images in the paper are checked for duplication of panels and for splicing of gel lanes.

Nature Cell Biology is committed to improving transparency in authorship. As part of our efforts in this direction, we are now requesting that all authors identified as 'corresponding author' on published papers create and link their Open Researcher and Contributor Identifier (ORCID) with their account on the Manuscript Tracking System (MTS), prior to acceptance. ORCID helps the scientific community achieve unambiguous attribution of all scholarly contributions. You can create and link your ORCID from the home page of the MTS by clicking on 'Modify my Springer Nature account'. For more information please visit www.springernature.com/orcid.

This journal strongly supports public availability of data. Please place the data used in your paper into a public data repository, or alternatively, present the data as Supplementary Information. If data can only be shared on request, please explain why in your Data Availability Statement, and also in the correspondence with your editor. Please note that for some data types, deposition in a public repository is mandatory - more information on our data deposition policies and available repositories appears below.

Link Redacted

We hope that you will find our referees' comments and editorial guidance helpful. Please do not hesitate to contact me if there is anything you would like to discuss. Thank you again for considering the journal for your work.

Best wishes,

Melina

Melina Casadio, PhD
Senior Editor, Nature Cell Biology
ORCID ID: <https://orcid.org/0000-0003-2389-2243>

Reviewers' Comments:

Reviewer #1:

Remarks to the Author:

In this manuscript, Rawal et al. examined how epithelial cells exhibit distinct behaviors at convex versus concave edges, correlated with ER morphology. Cells at convex edges are inclined to develop lamellipodia, supported by orderly actin fibers and microtubules, with ER tubules extending in the direction of migration. Conversely, at concave edges, there is a reduction in lamellipodia, actin fibers show contraction, microtubules form dense bundles, and ER tubules coalesce into sheet-like structures. By integrating imaging, genetic perturbations, and computational models, the authors propose the connection between different ER organizations and the mechanical states of the cells.

They claim that ER morphology dictates the mode of epithelial migration, as indicated by the title and abstract. This potentially distinguishes this study. However, this claim is mainly addressed only in Figures 6 and 7, where the presence of lamellipodia and distributions of focal adhesions are examined in the context of transient overexpressing of Rtn4 and Climp63. These experiments are somewhat descriptive and also, as I detail below, lack the robustness and persuasiveness needed to support their central claim. Most of the preceding data (Figs. 1-5) primarily discuss how ER morphology is shaped by the interplay between cytoskeletal organization and the mechanical forces associated with the curvature of the cell periphery. While potentially significant, the presentation of their data needs significant improvements for providing strong evidence for their conclusions.

Here are some comments for suggested improvements:

1. The authors demonstrate that altering ER morphology from tubular to sheet-like structures through overexpression experiments influences lamellipodia formation in cells. Similar phenomenon has been reported previously in different cell lines (e.g., doi: 10.1038/srep35969). The results are not exactly the same but there is a conceptual similarity. Moving forward, a detailed elucidation of how ER morphology dictates cytoskeletal and cell morphology is necessary, supported by robust experimental evidence. Incorporating both experimental data and theoretical analysis would significantly strengthen the investigation of this point.
2. The authors utilize the Weka segmentation method to quantify tubules and sheets, yet they do not provide detailed methodology. This omission is significant to ensure their findings, and a thorough explanation of their segmentation process is essential. At least, they should describe the supervised information used for training the segmentation tool, the specifics of the training process, and how reliable it is.
3. In relation to my previous comment, videos 1-4, which are intended to illustrate ER dynamics, lack clarity. For instance, the claim that 'In contrast, at the concave edge, we observed an anterograde movement of ER sheets towards the front of the migrating cells' is difficult to verify in the videos because the location of the ER sheet is not clear. This issue comes from the experimental method that uses only Sec61beta as an ER marker. The clarity of the results could be enhanced by conducting simultaneous imaging with a general ER marker and a specific ER sheet marker, such as Climp63 or p180. This technical improvement should improve the clarity of the data throughout the paper. Videos 3 and 4 are of only 8 minutes and should have longer time frame to clearly provide the evidence.
4. The overexpression experiments in this study were conducted using transient plasmid expression, which inherently causes variability in the amount of plasmid DNA entering the cells and, consequently, in the expression levels. Such variability could undermine the argument regarding the balance between tubular and sheet-like ER morphologies, as cautioned by a previous study (doi: 10.1038/srep35969): "Finally, in the light of these results, using the NOGO-B/RTN4B overexpression constructs as general markers for ER tubules should be carefully considered as even small changes in protein levels affect sheet-tubule balance and thereby induce clear changes in the ER morphology. Recent development in

targeted genome editing technology provides solution as this approach can be used to generate stable cell lines expressing epitopically tagged versions of various ER membrane proteins at endogenous levels to avoid changes in sheet-tubule balance or expansion of the membrane.”. It is strongly advised that the authors establish a stable cell line ensuring consistent expression levels across the population. Alternatively, the authors should detail the methods they used to normalize expression levels during transient transfection experiments.

5. In addition to experimental evidence, theoretical analysis could elucidate how protrusive and contractile forces influence ER morphology. However, the manuscript lacks detailed mathematical explanations, which weakens the support for the conclusions. A notable omission is the absence of spatial distribution data for actin fiber η or the protrusive/contractile forces. Although Eq. 3 indicates the regulation of magnitude in η , its spatial distribution, crucial for understanding mechanical stress distribution, is not addressed (as the equation lacks spatial variables, the spatial distribution is given and fixed). This omission makes it difficult to understand some statements, for instance, how cells experience different bending moments and axial forces during contraction, as mentioned on page 7, line 9.

I question whether the parallel ER model adequately represents the distribution of ER sheets, as shown in Figure 5a. While the perpendicular tubular ER is a reasonable extreme for convex conditions, the justification for the parallel sheet assumption as its counterpart is not clear. The small difference in strain energy density between parallel and perpendicular orientations (Figs. 4c, f) suggests that this discrepancy might vanish under conditions that more closely resemble experimental observations. This renders the results unconvincing.

Moreover, the modeling approach is phenomenological, and the rationale behind Eq. 2 needs clarification. The manuscript employs stress-strain rate relationships for active stress, more indicative of viscoelastic rather than purely elastic material behavior, which includes time-dependent responses under load. The manuscript also contains ambiguities, such as the definition of σ in Eq. 4, where it is unclear whether it refers to either σ^a , σ^p , or Σ . At this stage, the whole part of mathematical modeling and analysis does not contribute to providing the strong evidence for the conclusion.

6. Necessary information is missing in some graphs.

- Fig. 2b and d: Origins and numbers on both axes in the kymographs are missing. Position of nucleus is required to show if the flow of intensity is not due to the shifting of cell body itself.
- Fig. 3b: There are 6 lines in the graph but no labels for each.
- Fig. 5b: no labels
- Fig. 6e-g: what are the dotted lines?

7. Statistics: Please provide sufficient information of statistical analysis such as p-values and replicates.

Reviewer #2:

Remarks to the Author:

Epithelial gaps of various sizes and geometries appear in all organisms, from single-cell extrusion to centimeter-sized wounds. For closure of gaps, epithelial cells invoke two orthogonal modes: lamellipodial crawling at the convex edge and purse-string-like movements at the concave edge. Mechanisms underlying these differential responses to geometric cues have remained elusive. Here, the authors perform an intracellular cartography to reveal that in both micropatterned and naturally arising gaps, and they suggest that the endoplasmic reticulum (ER) undergoes edge curvature-dependent morphological reorganizations with convex and concave edges promoting ER tubules and sheets, respectively. This reorganization depends on cytoskeleton-generated protrusive and contractile forces. Additionally, theoretical modeling predicts that the curvature-specific ER morphology leads to a lower strain energy state. ER tubules at the convex edge favor perpendicularly oriented focal adhesions, supporting lamellipodial crawling while ER sheets at the concave edge favor parallelly oriented focal adhesions, supporting purse-string-like movements. Altogether, ER emerges as a central player in cellular mechanotransduction, which orchestrates two orthogonal modes of cell migration by integrating signals from cytoskeletal networks.

This study makes a potentially impactful finding regarding the role of curvature-dependent ER morphology changes on wound closure. The topic is quite interesting. However, the identification of peripheral sheets and methods for expansion of peripheral sheet-like structures (and to a lesser extent, ER tubules) appears overly simplistic and is potentially confounded. Specific points:

1. Dense ER at the front of migrating cells at the concave edge are described as ER sheets and fenestrated ER sheets. These peripheral structures may also represent dense tubular matrices (see Nixon-Abell et al, Science 2016). How do these different possibilities affect the conclusions of this study (or the mathematical modeling)? In other words, what if these are not sheets at all, but instead dense networks of tubules?
2. Related to the above, the generation of sheets and tubules via protein overexpression described here may be overly simplistic. For instance, Climp63 binds microtubules and localizes to ER tubules as well, and may not have sheet-generating properties at all (for example, see Zheng et al., Nature 2022). Have the authors tried constructs of Climp63 defective for microtubule interaction? Also, what is the effect of Climp63 knock out? For ER tubule generation, Rtn4 is a particularly large reticulon; are similar findings seen upon overexpression of a smaller one? Also, some knockout/knockdown experiments for reticulons (though admittedly there are several forms), may be helpful.

Reviewer #3:

Remarks to the Author:

The manuscript authored by Sieran Rawal et al., titled "Curvature-dependent morphological reorganization of the endoplasmic reticulum determines the mode of epithelial migration," investigates the curvature-dependent morphological reorganization of the endoplasmic reticulum (ER) at convex and concave edges.

The study presents intriguing observations utilizing a potentially valuable system for the field. While the study offers original insights into curvature-dependent ER morphologies, a notable limitation is the lack of clear definition and characterization of in-plane (concave and convex) curvatures. In addition, curvatures are sometimes studied at the single scale or at the multicellular scale.

1- The authors utilized both micropatterned wounds of defined geometry in cultured epithelial monolayers and natural wounds with spontaneously arising convex and concave curved regions in mouse embryonic epidermis. Although the manuscript mentions the use of polydimethylsiloxane (PDMS) stencils of various shapes, the specific geometries of these stencils are not clearly presented. It is essential for the authors to provide images of the micropatterns obtained from the negative stencils and offer a time-lapse sequence of a few typical tissues at low magnification during the healing process.

2- This work is very close to the work by Tianchi Chen et al. in Nature Physics 2019 "Large-scale curvature sensing by directional actin flow drives cellular migration mode switching" (<https://europepmc.org/backend/ptpmcrender.fcgi?accid=PMC6456019&blobtype=pdf>) and the flower pattern could be very useful here to better control and standardize experiments on positive and negative curvature.

3- The authors claimed that lamellipodial protrusions or actin bundles emerge within 15-30 minutes. This assertion should be substantiated with immunostained experiments of micropatterned tissues at various time points ($t=0$, $t=10$, $t=20$, and $t=30$ min) or using live actin imaging in mouse embryonic epidermal cells. Additionally, the fraction of ER tubules in Fig. 1d and the fraction of Climp63 in 1e should be presented as a function of curvature for both convex and concave patterns.

4- While Fig. 2 illustrates a cell-scale curvature-dependent mechanism mainly related to lamellipodial shape, Fig. 3 appears to depict curvature using a few cells. The authors could consider using adhesive micropatterns to impose specific curvatures (concave vs. convex) on individual mouse embryonic epidermal cells while controlling the spreading area, which could enhance the support for Fig. 2.

5- The scale of curvatures in this study is challenging to appreciate and appears to be close to the lamellipodial size. The authors should provide a robust quantification of convex and concave curvature as well as their evolution over time. Demonstrating the independence of results from curvature size, in addition to curvature sign, is crucial. Dynamic micropatterning could aid in showing the evolution of tubule fraction concerning dynamic changes in lamellipodial curvature.

6- The fraction of ER tubules in Fig. 3d appears to be significantly lower on concave edges (red) than on convex edges (green). It is imperative to present a plot with associated statistical tests to confirm this observation.

7- If there is indeed a lower fraction of ER tubules on concave edges compared to convex edges (as noted in #5), the authors should reflect this difference in their modeling presented in Fig. 3. If the model does not currently account for this discrepancy, it should be adapted accordingly.

8- The modeling of strain energy minimization in Fig. 4 was conducted within a curvature range of $0-0.2 \mu\text{m}^{-1}$, which seems unsupported by experimental data on concave and convex curvature. This discrepancy undermines the credibility of the results and must be addressed.

9- The authors mentioned the high axial stiffness of the ER resulting in lower cell deformation and thereby reducing strain energy density. Clarification is needed regarding what is meant by "high axial stiffness" and the specific values of stiffness or bending modulus used in the simulations.

10- Given the observed correlation between ER organization and microtubules (Figs. 5 and 6), it is important to inquire whether the presence of microtubules was implemented in the simulations presented in Fig. 4. If not, this omission should be addressed and justified.

11- In Figs. 6 and 7 (and throughout the manuscript), it is recommended to present all data in box plots with individual data points superimposed, rather than using bars alone.

12- The methodology for focal adhesion orientation analysis, particularly in Figs. 7c, lacks clarity. The authors should provide detailed information about their analysis pipeline, including higher magnification images, the results of their thresholding process, and clear demonstrations of orientation at the single adhesion scale.

13- Is there a clear relationship between focal adhesions orientation, and directional actin flow such as observed by Tianchi Chen et al. in Nature Physics 2019 "Large-scale curvature sensing by directional actin flow drives cellular migration mode switching" (<https://europepmc.org/backend/ptpmcrender.fcgi?accid=PMC6456019&blobtype=pdf>) ? Additional experiments on actin

treadmilling could be very informative.

Methods should be written concisely, but should contain all elements necessary to allow interpretation and replication of the results. As a guideline, Methods sections typically do not exceed 3,000 words. The Methods should be divided into subsections listing reagents and techniques. When citing previous methods, accurate references should be provided and any alterations should be noted. Information must be provided about: antibody dilutions, company names, catalogue numbers and clone numbers for monoclonal antibodies; sequences of RNAi and cDNA probes/primers or company names and catalogue numbers if reagents are commercial; cell line names, sources and information on cell line identity and

authentication. Animal studies and experiments involving human subjects must be reported in detail, identifying the committees approving the protocols. For studies involving human subjects/samples, a statement must be included confirming that informed consent was obtained. Statistical analyses and information on the reproducibility of experimental results should be provided in a section titled "Statistics and Reproducibility".

All Nature Cell Biology manuscripts submitted on or after March 21 2016 must include a Data availability statement as a separate section after Methods but before references, under the heading "Data Availability". For Springer Nature policies on data availability see <http://www.nature.com/authors/policies/availability.html>; for more information on this particular policy see <http://www.nature.com/authors/policies/data/data-availability-statements-data-citations.pdf>. The Data availability statement should include:

- Accession codes for primary datasets (generated during the study under consideration and designated as "primary accessions") and secondary datasets (published datasets reanalysed during the study under consideration, designated as "referenced accessions"). For primary accessions data should be made public to coincide with publication of the manuscript. A list of data types for which submission to community-endorsed public repositories is mandated (including sequence, structure, microarray, deep sequencing data) can be found here <http://www.nature.com/authors/policies/availability.html#data>.
- Unique identifiers (accession codes, DOIs or other unique persistent identifier) and hyperlinks for datasets deposited in an approved repository, but for which data deposition is not mandated (see here for details <http://www.nature.com/sdata/data-policies/repositories>).
- At a minimum, please include a statement confirming that all relevant data are available from the authors, and/or are included with the manuscript (e.g. as source data or supplementary information), listing which data are included (e.g. by figure panels and data types) and mentioning any restrictions on availability.
- If a dataset has a Digital Object Identifier (DOI) as its unique identifier, we strongly encourage including this in the Reference list and citing the dataset in the Methods.

We recommend that you upload the step-by-step protocols used in this manuscript to [protocols.io](http://www.protocols.io). More details can be found at <https://www.protocols.io/help/publish-articles>.

All imaging data should be accompanied by scale bars, which should be defined in the legend.

Cropped images of gels/blots are acceptable, but need to be accompanied by size markers, and to retain visible background signal within the linear range (i.e. should not be saturated). The boundaries of panels with low background have to be demarked with black lines. Splicing of panels should only be considered if unavoidable, and must be clearly marked on the figure, and noted in the legend with a statement on whether the samples were obtained and processed simultaneously. Quantitative comparisons between samples on different gels/blots are discouraged; if this is unavoidable, it should only be performed for samples derived from the same experiment with gels/blots were processed in parallel, which needs to be stated in the legend.

- We do not recommend using Adobe Photoshop for designing figures, but we can accept Photoshop generated (.PSD or .TIFF) files only if each element included in the figure (text, labels, pictures, graphs, arrows and scale bars) are on separate

layers. All text should be editable in 'type layers' and line-art such as graphs and other simple schematics should be preserved and embedded within 'vector smart objects' - not flattened raster/bitmap graphics.

The total number of Supplementary Figures (not including the "unprocessed scans" Supplementary Figure) should not exceed the number of main display items (figures and/or tables (see our Guide to Authors and March 2012 editorial <http://www.nature.com/ncb/authors/submit/index.html#supinfo>; <http://www.nature.com/ncb/journal/v14/n3/index.html#ed>). No restrictions apply to Supplementary Tables or Videos, but we advise authors to be selective in including supplemental data.

GUIDELINES FOR EXPERIMENTAL AND STATISTICAL REPORTING

REPORTING REQUIREMENTS – We are trying to improve the quality of methods and statistics reporting in our papers. To that end, we are now asking authors to complete a reporting summary that collects information on experimental design and reagents. The Reporting Summary can be found here <https://www.nature.com/documents/nr-reporting-summary.pdf>. If you would like to reference the guidance text as you complete the template, please access these flattened versions at <http://www.nature.com/authors/policies/availability.html>.

STATISTICS – Wherever statistics have been derived the legend needs to provide the n number (i.e. the sample size used to derive statistics) as a precise value (not a range), and define what this value represents. Error bars need to be defined in the legends (e.g. SD, SEM) together with a measure of centre (e.g. mean, median). Box plots need to be defined in terms of minima, maxima, centre, and percentiles. Ranges are more appropriate than standard errors for small data sets. Wherever

statistical significance has been derived, precise p values need to be provided and the statistical test used needs to be stated in the legend. Statistics such as error bars must not be derived from $n < 3$. For sample sizes of $n < 5$ please plot the individual data points rather than providing bar graphs. Deriving statistics from technical replicate samples, rather than biological replicates is strongly discouraged. Wherever statistical significance has been derived, precise p values need to be provided and the statistical test stated in the legend.

Version 2:

Decision Letter:

Our ref: NCB-A53522B

27th May 2025

Dear Dr. Das,

Thank you for submitting your revised manuscript "Curvature-dependent morphological reorganization of the endoplasmic reticulum determines the mode of epithelial migration" (NCB-A53522B). It has now been seen by the original referees and their comments are below. The reviewers find that the paper has improved in revision, and therefore we'll be happy in principle to publish it in Nature Cell Biology, pending minor revisions to satisfy the referees' final requests and to comply with our editorial and formatting guidelines.

Thank you again for your interest in Nature Cell Biology Please do not hesitate to contact me if you have any questions.

Sincerely,

Angela R Parrish, PhD
Locum Senior Editor
Nature Cell Biology

Reviewer #1 (Remarks to the Author):

I appreciate the authors' comprehensive responses to all my previous comments. Their efforts to address each point are respectable, and the manuscript has improved overall.

I would like to highlight one remaining issue regarding figure presentation. Specifically, the scale bars in most figures are too small to be clearly visible, which may hinder proper interpretation of the data. Additionally, some panels are missing scale bars altogether (e.g., Figures 2h and 2i). The authors should revise the figure panels to ensure all scale bars are clearly legible and consistently presented across the figures.

Once this issue is addressed, I believe the manuscript will be suitable for publication.

Reviewer #1 (Remarks on code availability):

A detailed assessment was not possible because it requires Abaqus, which I do not have. The README file provides sufficient instructions.

Reviewer #2 (Remarks to the Author):

In this revised manuscript, the authors have performed numerous additional experiments and analyses. Each of the points that I raised in my original review has been addressed.

Version 3:

Decision Letter:

Dear Dr Das,

I am pleased to inform you that your manuscript, "Edge curvature drives endoplasmic reticulum reorganization and dictates epithelial migration mode", has now been accepted for publication in *Nature Cell Biology*.

Over the next few weeks, your paper will be copyedited to ensure that it conforms to *Nature Cell Biology* style. Once your paper is typeset, you will receive an email with a link to choose the appropriate publishing options for your paper and our Author Services team will be in touch regarding any additional information that may be required.

Publication is conditional on the manuscript not being published elsewhere and on there being no announcement of this work to any media outlet until the online publication date in *Nature Cell Biology*.

Please note that *Nature Cell Biology* is a Transformative Journal (TJ). Authors may publish their research with us through the traditional subscription access route or make their paper immediately open access through payment of an article-processing charge (APC). Authors will not be required to make a final decision about access to their article until it has been accepted. [Find out more about Transformative Journals](https://www.springernature.com/gp/open-research/transformative-journals)

Authors may need to take specific actions to achieve [compliance with funder and institutional open access mandates](https://www.springernature.com/gp/open-research/funding/policy-compliance-faqs). If your research is supported by a funder that requires immediate open access (e.g. according to [Plan S principles](https://www.springernature.com/gp/open-research/plan-s-compliance)) then you should select the gold OA route, and we will direct you to the compliant route where possible. For authors selecting the subscription publication route, the journal's standard licensing terms will need to be accepted, including [self-archiving policies](https://www.springernature.com/gp/open-research/policies/journal-policies). Those licensing terms will supersede any other terms that the author or any third party may assert apply to any version of the manuscript.

If you have not already done so, we strongly recommend that you upload the step-by-step protocols used in this manuscript to protocols.io (<https://protocols.io>), an open online resource that allows researchers to share their detailed experimental know-how. All uploaded protocols are made freely available and are assigned DOIs for ease of citation. Protocols and Nature Portfolio journal papers in which they are used can be linked to one another, and this link is clearly and prominently visible in the online versions of both. Authors who performed the specific experiments can act as primary authors for the Protocol as they will be best placed to share the methodology details, but the Corresponding Author of the present research paper should be included as one of the authors. By uploading your Protocols onto protocols.io, you are enabling researchers to more readily reproduce or adapt the methodology you use, as well as increasing the visibility of your protocols and papers. You can also establish a dedicated workspace to collect your lab Protocols. Further information can be found at <https://www.protocols.io/help/publish-articles>.

Nature Cell Biology encourages authors presenting evidence for cell, biological, molecular, and genetic interactions to consider communicating these findings using Biofactoid (<https://biofactoid.org/>). This tool helps users share a searchable representation of interactions (e.g. binding, gene expression, post-translational modification) between genes, gene products, or chemicals. Information added to Biofactoid, with author attribution, is shared on social media and public databases, such as Pathway Commons, where it can be discovered and analyzed in the context of a large and growing corpus of knowledge.

With kind regards,

Angela R Parrish, PhD
Locum Senior Editor
Nature Cell Biology

** Visit the Springer Nature Editorial and Publishing website at http://editorial-jobs.springernature.com?utm_source=ejp_NCB_email&utm_medium=ejp_NCB_email&utm_campaign=ejp_NCB for more information about our career opportunities. If you have any questions please click [here](mailto:editorial.publishing.jobs@springernature.com).

Response to Review Comments

Reviewer #1:

In this manuscript, Rawal et al. examined how epithelial cells exhibit distinct behaviors at convex versus concave edges, correlated with ER morphology. Cells at convex edges are inclined to develop lamellipodia, supported by orderly actin fibers and microtubules, with ER tubules extending in the direction of migration. Conversely, at concave edges, there is a reduction in lamellipodia, actin fibers show contraction, microtubules form dense bundles, and ER tubules coalesce into sheet-like structures. By integrating imaging, genetic perturbations, and computational models, the authors propose the connection between different ER organizations and the mechanical states of the cells. They claim that ER morphology dictates the mode of epithelial migration, as indicated by the title and abstract. This potentially distinguishes this study. However, this claim is mainly addressed only in Figures 6 and 7, where the presence of lamellipodia and distributions of focal adhesions are examined in the context of transient overexpressing of Rtn4 and Climp63. These experiments are somewhat descriptive and also, as I detail below, lack the robustness and persuasiveness needed to support their central claim. Most of the preceding data (Figs. 1-5) primarily discuss how ER morphology is shaped by the interplay between cytoskeletal organization and the mechanical forces associated with the curvature of the cell periphery. While potentially significant, the presentation of their data needs significant improvements for providing strong evidence for their conclusions.

Response: We thank the reviewer for the thoughtful and constructive comments, which significantly improved the depth and clarity of our study. In response, we introduced a second layer of perturbations by combining Climp63- and Rtn4a-overexpression with cytoskeletal drug treatments, revealing a mechanistic tug-of-war between ER structure and cytoskeletal dynamics. These experiments strongly demonstrated that ER sheets suppress lamellipodia formation, while ER tubules hinder the stabilization of contractile actin bundles, highlighting an instructive role for ER morphology in migration mode selection. To reduce variability, we generated stable cell lines expressing ER-shaping proteins at consistent levels. We also included additional live imaging, including super-resolution time-lapse microscopy, to track ER dynamics at wound edges. We further expanded our mathematical modeling to test alternative ER configurations, confirming the energetic favorability of ER sheets at concave edges. Importantly, we quantified curvature at both single-cell and tissue levels, validated our ER segmentation pipeline, and linked ER morphology to ER-PM contact distribution. We believe that these new additions together have significantly strengthened our conclusion that ER structure instructively governs curvature-guided epithelial migration.

Here are some comments for suggested improvements:

1. The authors demonstrate that altering ER morphology from tubular to sheet-like structures through overexpression experiments influences lamellipodia formation in cells. Similar phenomenon has been reported previously in different cell lines (e.g., doi: 10.1038/srep35969). The results are not exactly the same but there is a conceptual similarity. Moving forward, a detailed elucidation of how ER morphology dictates cytoskeletal and cell morphology is necessary, supported by robust experimental evidence. Incorporating both experimental data and theoretical analysis would significantly strengthen the investigation of this point.

Response: We thank the reviewer for their insightful comments. In this study, we have investigated how curvature-driven differential forces shape ER morphology, which in turn stabilizes cytoskeletal organization and dictates epithelial migration modes. Our previous results demonstrated that at convex wound edges, cells predominantly form ER tubules, supporting lamellipodial crawling, while at concave edges, they exhibit ER sheets, reinforcing actomyosin-driven contractility and purse-string closure. These results suggested that ER

morphology is not merely a downstream effect of cytoskeletal forces but actively influences migration mode. To further dissect this relationship, we introduced a second layer of perturbations using drug treatments in Climp63- and Rtn4a-overexpressing cells. Please see the table below, with highlighted rows being the most important ones. This set of dual perturbations allowed us to untangle the interplay between curvature, ER structure, and cytoskeletal organization.

- A. ER Structure as a Dominant Factor in Curvature-Specific Cytoskeletal Organization: Curvature normally dictates cytoskeletal architecture - lamellipodia at convex edges and contractile actin bundles at concave edges. However, overexpression of Climp63 (favoring ER sheets) at convex edges disrupted this pattern, leading to actin bundle formation instead of lamellipodia, while Rtn4a (favoring ER tubules) at concave edges increased lamellipodia, suppressing actin bundles (Figs. 6a-b and Supplementary Figs. 10b-e). These results confirmed that ER morphology is not just adapting to curvature-dependent cytoskeletal forces but actively shaping them.
- B. Introducing a Tug-of-War at the Convex Edge: ER Sheets vs. Branched Actin Polymerization: In wild-type cells, Blebbistatin treatment, which inhibits actomyosin contractility, promoted lamellipodia formation at the concave edge (Supplementary Fig. 7a). However, in Climp63-overexpressing cells, Blebbistatin failed to rescue lamellipodia, indicating that ER sheets override the effect of contractility inhibition, locking cells into a contractile state (Fig. 6d). This also suggests that ER tubules are necessary for branched actin polymerization and lamellipodia-driven migration. Blebbistatin treatment also failed to completely rescue microtubule rearrangements from parallel to perpendicular in Climp63 over-expressing cells (Supplementary Figs. 10i-j).
- C. Introducing a Tug-of-War at the Concave Edge: ER Tubules vs. Contractile Actin Bundles: In wild-type cells, CK666 treatment, which inhibits Arp2/3-mediated branched actin polymerization, suppressed lamellipodia and promoted actin bundle formation at the convex edge (Supplementary Fig. 7a). However, in Rtn4a-overexpressing cells, CK666 failed to fully restore actin bundles, leaving a few persistent protrusive structures at the edge and disrupting the normally continuous contractile actin architecture (Fig. 6d). This suggests that once ER tubules accumulate, they prevent full re-establishment of contractile actin structures, even when lamellipodia formation is inhibited. In contrast, CK666 treatment in Rtn4a-overexpressing cells led to a reorientation of microtubules from a perpendicular to a parallel arrangement, suggesting that microtubule organization adapts to the dominating actin architecture, and ER morphology may exert a more decisive influence on actin dynamics than on microtubules (Supplementary Figs. 10k-l).
- D. ER-Plasma Membrane (ER-PM) Contacts in Migration: A recent study (<https://pubmed.ncbi.nlm.nih.gov/38867038/>) using ER-PM (plasma membrane) contacts marker MAPPER demonstrated that ER-PM contacts polarize in single migrating cells, with Climp63-rich ER sheets at the rear showing more ER-PM contacts than Rtn4a-rich tubular ER at the front, influencing receptor tyrosine kinase signaling and phosphoinositide distribution. While their focus was on the front and back of cells during single-cell migration, our study examines curvature-dependent collective epithelial migration, where mechanical forces and ER organization are regulated at tissue scale. Despite these differences, we find an intriguing connection: curvature-driven ER morphology correlates with ER-PM contact distribution, with reduced ER-PM contacts at convex edges (ER tubule-enriched) and increased contacts at concave edges (ER sheet-enriched) (Supplementary Figs. 6c-d). This suggests that curvature may influence ER-PM contact site regulation through ER structural changes, potentially affecting signaling pathways, including receptor tyrosine kinase signaling and phosphoinositide distribution. Regulating phosphoinositide distribution can then regulate key cellular processes

including cytoskeletal dynamics, membrane trafficking, polarity, and receptor signaling, which are essential components cell migration. Rather than a direct parallel, we draw mechanistic inspiration from this study, extending the role of ER beyond intracellular organization to a possible regulator of curvature-dependent signaling in epithelial migration.

Together, by introducing dual perturbations - ER structural manipulation combined with cytoskeletal drugs - we created a mechanistic tug-of-war that revealed the critical influences of ER morphology on cytoskeletal organization: ER sheets stabilize actin bundles and suppress lamellipodia, while ER tubules promote branched actin polymerization and destabilize contractile bundles. The inability of cytoskeletal drugs to fully restore curvature-specific migration modes after ER perturbations strengthens this conclusion, highlighting the instructive role of ER structure in directing curvature-dependent cell migration. Furthermore, our findings reveal that curvature-driven ER organization influences ER-PM contact site distribution, which should control further downstream signaling. Rather than being a passive responder to curvature, ER emerges as a key integrator of mechanical and biochemical cues. By demonstrating that ER structure can override cytoskeletal responses to curvature cues, our study provides a new framework for understanding intracellular organelles as regulators of collective cell migration, while strengthening the mechanistic foundation of our work. Notably, nocodazole treatment, which disrupts microtubules, consistently promoted ER sheet formation along with thick, continuous actin bundles at the edge and increased stress fibers within the cell, in both Rtn4a- and Climp63-overexpressing cells (Supplementary Figs. 11a–b). This supports our earlier observation that microtubule-ER interactions facilitate ER tubule generation and suggests that microtubules contribute primarily to lamellipodia formation at convex edges but play a lesser role in actin bundling at concave edges.

We have reported these new results and described the new mechanistic insights in the revised manuscript, please see Page 6, Lines 18-31, the section “Manipulating ER morphology alters the cellular response to edge curvature” on Page 11-12, Figs. 6a-d, Fig. 7e, Supplementary Figs. 6c-d, and Supplementary Figs. 10b-l.

Rebuttal Table 1:

Curvature	Cytoskeletal changes	Over-expression	Cytoskeletal changes	Drug treatment	Cytoskeletal changes
Convex	Protrusive Lamellipodia; Perpendicular MT	Rtn4a	Protrusive Lamellipodia; Perpendicular MT	CK666	Bundled Actin with remnant protrusions and parallel MT
Convex	Protrusive Lamellipodia; Perpendicular MT	Rtn4a	Protrusive Lamellipodia; Perpendicular MT	Nocodazole	Actin bundle and stress fibres, depolymerised MT
Convex	Protrusive Lamellipodia; Perpendicular MT	Climp63	Actin bundling, parallel MT	Blebbistatin	Small protrusive actin structures, perpendicular as well as parallel MT
Convex	Protrusive Lamellipodia; Perpendicular MT	Climp63	Actin bundling, parallel MT	Nocodazole	Actin bundle and stress fibres, depolymerised MT
Concave	Actin Bundling, parallel MT	Rtn4a	Protrusive Lamellipodia; Perpendicular MT	CK666	Bundled actin with few persistent

					protrusions; mostly parallel MT
Concave	Actin Bundling, parallel MT	Rtn4a	Protrusive Lamellipodia; Perpendicular MT	Nocodazole	Actin bundle and stress fibres, depolymerised MT
Concave	Actin Bundling, parallel MT	Climp63	Actin bundling, parallel MT	Blebbistatin	Small protrusive actin structures, perpendicular as well as parallel MT
Concave	Actin Bundling, parallel MT	Climp63	Actin bundling, parallel MT	Nocodazole	Actin bundle and stress fibres, depolymerised MT

MT: Microtubules

Finally, to examine whether ER morphology influences cytoskeletal organization in non-migrating cells, we transiently overexpressed Rtn4a and Climp63 in MDCK monolayers to induce ER tubule- and sheet-enriched states, respectively. This mosaic approach allowed us to directly compare the cytoskeletal organization of transfected cells against neighboring wild-type cells within the same monolayer, serving as internal controls. Immunostaining for actin and microtubules revealed no significant differences in their organization in either condition (Supplementary Fig. 10f), suggesting that ER morphological changes do not overtly disrupt cytoskeletal architecture in non-migrating cells. To further investigate whether ER restructuring affects collective properties such as cell shape index (perimeter/(area)^{0.5}) (<https://www.nature.com/articles/nphys3471>), we used FACS and antibiotic selection to generate stable MDCK populations homogeneously expressing mCherry-Climp63 or Rtn4a-GFP. ZO-1 staining was used to delineate cell boundaries, and subsequent shape index analysis revealed slightly higher values in both Climp63- and Rtn4a-expressing monolayers compared to wild-type. While statistically significant, these changes were modest, and most cells retained a cobblestone-like morphology, suggesting limited physiological relevance from a shape-transition perspective (Supplementary Figs. 10g–h). Please see Page 11, Line 37 to Page 12, Line 5.

2. The authors utilize the Weka segmentation method to quantify tubules and sheets, yet they do not provide detailed methodology. This omission is significant to ensure their findings, and a thorough explanation of their segmentation process is essential. At least, they should describe the supervised information used for training the segmentation tool, the specifics of the training process, and how reliable it is.

Response: We have used trainable WEKA segmentation 2D (<https://pubmed.ncbi.nlm.nih.gov/28369169/>), a plugin in Fiji to segment ER tubules and sheet-like dense ER as has also been previously used to quantify ER tubules (<https://pubmed.ncbi.nlm.nih.gov/33328230/>). This machine-learning-based segmentation tool allows automated and reproducible classification of ER structures by training on specific image features. To train the model, we selected key image processing features, including Gaussian blur, Sobel filter, Hessian, difference of Gaussians, and membrane projections, which help distinguish ER tubules from sheets based on intensity, texture, and edge detection. A sample image, as shown in Rebuttal Fig. 1a, was manually annotated to define three structural categories or classes: 1. thin peripheral ER tubules, 2. dense bright sheet-like ER, and 3. background regions that included cytoplasmic spaces between ER structures and the extracellular area. The tool was iteratively trained by manually marking multiple areas in each category until the segmented output closely matched the original image. Once trained, the

model was saved and applied to multiple images to ensure consistency and minimize user bias in segmentation.

Following classification, the segmented image (Rebuttal Fig. 1a) was converted to an 8-bit grayscale format for further quantification. Using thresholding, we extracted binary masks for tubules, sheet-like dense ER, and total ER (Rebuttal Fig. 1b). The pixel area corresponding to each category was measured, providing quantitative data on the fraction of tubules per cell. This automated approach ensures an objective assessment of ER morphology across different experimental conditions.

To assess the reliability of our WEKA-based segmentation, we performed a validation by comparing automated segmentation outputs with manual annotations. A subset of images ($n = 30$) was manually segmented using Fiji's polygon selection tool to trace ER tubule and sheet regions. We then applied our trained WEKA model to the same images and quantified the corresponding pixel areas. The fraction of ER tubules obtained by both methods showed a strong linear correlation (Pearson $r = 0.99$, $p < 0.001$), indicating that the automated classification reliably replicates manual assessment (Rebuttal Fig. 1c). These results demonstrate the robustness and accuracy of our segmentation pipeline. All methodological details have been incorporated into the manuscript's Methods (Supplementary Material) section to ensure transparency and reproducibility.

Rebuttal Figure 1. Segmentation of ER. a. MDCK cell transfected with mApplesec61β used as sample image (left), classified image produced after training (right) b. Segmented ER tubules (left), segmented ER sheets (right). c. Quantification of ER tubules manual vs by Trainable Weka Segmentation (TWS) $n=30$.

3. In relation to my previous comment, videos 1-4, which are intended to illustrate ER dynamics, lack clarity. For instance, the claim that 'In contrast, at the concave edge, we observed an anterograde movement of ER sheets towards the front of the migrating cells. At the concave edge, there is an anterograde movement of ER sheets towards the front of migrating cells (line 3 page 5)' is difficult to verify in the videos because the location of the ER sheet is not clear. This issue comes from the experimental method that uses only Sec61beta as an ER marker. The clarity of the results could be enhanced by conducting simultaneous imaging with a general ER marker and a specific ER sheet marker, such as Climp63 or p180. This technical

improvement should improve the clarity of the data throughout the paper. Videos 3 and 4 are of only 8 minutes and should have longer time frame to clearly provide the evidence.

Response: We thank the reviewer for this valuable suggestion. As initially presented, we performed live confocal imaging of LifeAct MDCK cells expressing mApple-Sec61 β to examine ER dynamics during migration at different curvature zones (Figs. 2e–g). However, as the reviewer rightly pointed out, Sec61 β alone may not clearly distinguish ER sheets from tubules in standard-resolution imaging. To address this, we conducted super-resolution live imaging using SoRa microscopy in LifeAct MDCK cells expressing Sec61 β , which provided enhanced spatial clarity. This allowed us to clearly visualize dense ER sheet-like structures moving anterogradely toward the leading edge at concave curvatures, and retrogradely at convex curvatures, as shown in Supplementary Fig. 6e. To further confirm that the dense ER structures at the concave front are indeed sheets, we expressed mCherry-Climp63, a known ER sheet marker, in LAMMDCK cells. To minimize the possibility of overexpression-induced structural artifacts, we selected cells with low to moderate expression. Live imaging of these cells revealed anterograde movement of Climp63-positive ER sheets toward the leading edge at the concave wound edge (Fig. 2h), thereby supporting our interpretation.

As further suggested by the reviewer, we attempted simultaneous imaging using both a general ER marker (ER-mVenus) and an ER sheet marker (mCherry-Climp63). However, in this dual-transfected setup, cells failed to migrate and instead retracted shortly after the PDMS chamber was lifted, likely due to the stress of dual transfection combined with live imaging (Rebuttal Fig. 2). While technically challenging, this effort underscores our commitment to addressing the reviewer’s concern.

Regarding the video durations, Videos 3 and 4 were kept to 8 minutes intentionally, as they focus on short-timescale ER tubule dynamics, which occur rapidly and are difficult to follow over longer periods due to continuous remodeling. The longer-term ER behaviors, such as overall sheet/tubule positioning and directional flow, are captured in Videos 1 and 2. Notably, even within the short timescales shown in Videos 3 and 4, individual ER tubules demonstrate consistent retrograde movement at convex and anterograde movement at concave edges, aligning with the overall patterns observed.

Rebuttal Figure 2. Live imaging of MDCK co-transfected with ER-venus and mCherry Climp63, labelled with cell mask (Magenta).

4. The overexpression experiments in this study were conducted using transient plasmid expression, which inherently causes variability in the amount of plasmid DNA entering the cells and, consequently, in the expression levels. Such variability could undermine the argument regarding the balance between tubular and sheet-like ER morphologies, as cautioned by a previous study (doi: 10.1038/srep35969): “Finally, in the light of these results, using the NOGO-B/RTN4B overexpression constructs as general markers for ER tubules should be carefully considered as even small changes in protein levels affect sheet-tubule balance and thereby induce clear changes in the ER morphology. Recent development in targeted genome editing technology provides solution as this approach can be used to generate stable cell lines expressing epitopically tagged versions of various ER membrane proteins at endogenous levels to avoid changes in sheet-tubule balance or expansion of the membrane.”. It is strongly advised that the authors establish a stable cell line ensuring consistent expression levels across the population. Alternatively, the authors should detail the methods they used to normalize expression levels during transient transfection experiments.

Response: We thank the reviewer for this critical and valuable suggestion. The transient overexpression allows rapid screening of phenotypes and enables analysis across a range of expression levels, with internal controls, offering insight into dose-dependent effects. However, we fully recognize the limitations of transient overexpression, particularly the variability in expression levels across a cell population, which can affect the interpretation of morphological changes in ER structure. To address this, we established stable MDCK cell lines expressing Rtn4a-GFP or mCherry-Climp63 using conventional antibiotic-mediated clonal selection. However, as noted in previous studies and from our own experience, even monoclonal MDCK lines can develop spontaneous heterogeneity in expression over multiple passages. To ensure consistency and uniformity in expression levels during experiments, we implemented an additional step of fluorescence-activated cell sorting (FACS) before each experiment. This allowed us to select cells with homogeneous and comparable expression of the overexpressed proteins. These FACS-sorted, stable lines were then seeded around PDMS chambers to assess curvature-dependent migration. The details of this protocol have been clearly described in the Methods section of the revised manuscript (Please see the revised Supplementary Material). Using these well-controlled stable lines, we observed that Rtn4a-GFP-expressing cells formed prominent lamellipodial protrusions at both convex and concave curvatures (Fig. 6a). MDE (mean distribution from edge) analysis of actin showed increased protrusive activity at concave edges, consistent with a shift away from contractile bundle formation (Fig. 6b). In contrast, mCherry-Climp63-expressing cells exhibited prominent apical actin bundles at both curvatures, with reduced basal lamellipodia, as reflected in lower MDE values at convex edges (Fig. 6b). Time-lapse imaging further revealed that Rtn4a-GFP cells closed the wound gap faster than WT, with some cells migrating individually into the gap—suggestive of enhanced protrusive behavior linked to increased ER tubules (Fig. 6c). Interestingly, Climp63-expressing cells also exhibited accelerated gap closure, potentially reflecting a more contractile, bundle-driven migration mode (Fig. 6c). Together, these results underscore that ER morphology alterations directly influence both the mode and rate of epithelial migration, and that our use of FACS-sorted, stable lines provide a robust and reproducible system for dissecting these effects. Please see Page 11, Line 24-36.

5. In addition to experimental evidence, theoretical analysis could elucidate how protrusive and contractile forces influence ER morphology. However, the manuscript lacks detailed mathematical explanations, which weakens the support for the conclusions. A notable omission is the absence of spatial distribution data for actin fiber η or the protrusive/contractile forces. Although Eq. 3 indicates the regulation of magnitude in η , its spatial distribution, crucial for understanding mechanical stress distribution, is not addressed (as the equation lacks spatial

variables, the spatial distribution is given and fixed). This omission makes it difficult to understand some statements, for instance, how cells experience different bending moments and axial forces during contraction, as mentioned on page 7, line 9.

Response: We agree with the reviewer that a more detailed explanation of the mathematical model is useful. Our previously brief description of the model may have led to confusion about the correlation between the spherical coordinate system, where the actin fibre distribution (η) was defined and the Cartesian coordinates where mechanical equilibrium is solved. We have now added more details in the supplementary material, describing the conversion of active stress in the (ω, θ) coordinate system to the spatial rectangular coordinate system (X_1, X_2, X_3) (Supplementary Eq. (4,5)). In addition, we have also added a schematic to show how a scalar defined in the (ω, θ) coordinate system and the Cartesian (X_1, X_2, X_3) coordinate system overlap (Supplementary Fig. 8e). We have also added equation (8) to show how we convert back the strain rate evaluated in the Cartesian coordinate system to (ω, θ) coordinates.

To model the actin cortex, we assume that the fibres are uniformly distributed in the cortical region, resulting in uniform contractility in the cortex. For the purposes of numerical integration, we assume 20 equally spaced discrete angles in the ϕ, ω domains. The fibre concentration η , will be initially evaluated in the (ω, θ) coordinate system assuming a stress and strain-free initial conditions. Active stress (Supplementary Eq. (2)) will be evaluated and homogenised (Supplementary Eq. 5) to obtain the active stress tensor. The passive stress tensor will be evaluated, assuming linear elasticity, (Supplementary Eq. (6)) and added to the active stress tensor to get the total stress. The mechanical equilibrium (Supplementary Eq. (7)) will be solved using the finite element method with small strain formulation. The displacement will be obtained, and strain and stress will be evaluated. The strain rate will be further evaluated which will then be converted back to (ω, θ) coordinates to update active stress. The solution of mechanical equilibrium depends on the geometry of the structure and the boundary conditions, thereby regulating the strain, strain rate and therefore the active stress generated.

Using this model on the cell with concave and convex curvatures, a uniform active stress is generated in the actin cortex. In the case of cell with concave curvature, we constrain displacement in all the degrees of freedom on the bottom side of the cell (represented by the orange line with small dashes) and leave the remaining sides free (Rebuttal Fig 3a). The active contractile stress results in the deformation of the free ends of the cell (Rebuttal Fig 3b). The total stress in the actin cortex is also mapped on the deformed geometry (Rebuttal Fig 3b).

Rebuttal Figure 3. a) Schematic of displacement (u) boundary conditions on a cell with concave curvature. b) Deformation of actin cortex due to contractile stress generated in the actin cortex. Total stress (Von Mises) mapped on the deformed geometry. c) The Cartesian coordinate system used in a).

To model protrusion, we change the sign of active stress relative to the contraction model, and restrict the direction of protrusion of the sides marked in black with large dashes to only along the Y axis by constraining displacements along X, and Z axes (Rebuttal Fig 4a). Additionally,

we constrain all degrees of freedom on the bottom side of the cell (represented by an orange line with small dashes, Rebuttal Fig. 4a). The protrusive force results in deformation of the actin cortex as shown in Rebuttal Fig. 4b. The total stress in the actin cortex is also mapped on the deformed geometry (Rebuttal Fig. 4b).

Rebuttal Figure 4. a) Schematic of displacement (u) boundary conditions on a cell with convex curvature. b) Deformation of actin cortex due to protrusive stress generated in the actin cortex. Total stress (Von Mises) mapped on the deformed geometry. c) The Cartesian coordinate system used in a).

Thus, the active stress generated in the actin cortex coupled with the right boundary conditions will lead to protrusive forces in the case of convex curvature and to contractile forces for concave curvature. We have added schematics to show the direction of forces generated during contraction and protrusion (Supplementary Figs. 8a-b), along with the corresponding deformations observed.

I question whether the parallel ER model adequately represents the distribution of ER sheets, as shown in Figure 5a. While the perpendicular tubular ER is a reasonable extreme for convex conditions, the justification for the parallel sheet assumption as its counterpart is not clear. The small difference in strain energy density between parallel and perpendicular orientations (Figs. 4c, f) suggests that this discrepancy might vanish under conditions that more closely resemble experimental observations. This renders the results unconvincing.

Response: We thank the reviewer for the insightful feedback. We agree that the parallel ER model did not completely simulate the sheet ER morphology. Hence, we have updated the model with additional morphology representing a sheet (Fig. 4a) instead of several beams arranged parallel to each other. This model gave an even higher difference between tubular and sheet morphologies in terms of strain energy density (Figs. 4e and h). Hence, we propose that parallel ER could be considered as the case of a cluster of tubules arranged parallel to each other. Our major conclusion from the model is unchanged: the ER morphologies resulting in minimum strain energy in either the concave, contractile case or the convex, protrusive case are the ones predominantly observed in the experiments.

Moreover, the modeling approach is phenomenological, and the rationale behind Eq. 2 needs clarification. The manuscript employs stress-strain rate relationships for active stress, more indicative of viscoelastic rather than purely elastic material behavior, which includes time-dependent responses under load. The manuscript also contains ambiguities, such as the definition of σ in Eq. 4, where it is unclear whether it refers to either σ^a , σ^p , or Σ . At this stage, the whole part of mathematical modeling and analysis does not contribute to providing the strong evidence for the conclusion.

Response: We agree the model is, in part, phenomenological, to simplify some of the complex biophysical and molecular processes that regulate contractility, protrusions, ER and the

cytoskeleton. A full mechano-chemical model that can capture the molecular and mechanical dynamics of the underlying processes is currently beyond reach, as many critical parameters (such as chemical rate constants of ER regulating molecules, and how they vary with mechanical and geometrical stimuli) are not experimentally identified.

The main aim of the model was to explain the mechanism regulating the curvature and force-dependent preference of ER morphology. We, therefore, derived the simplest possible model that is able to provide this explanation. The use of a strain-rate dependent model for the actin cortex is motivated by the use of similar models to simulate internal active stress growth which is dictating processes such as cell reorientation and substrate-topography responses [Pathak et al, 2008, *J. R. Soc. Interface*.**5**: 507–524; Tommaso et al. 2016, *Biophysical Journal*, 111,10: 2274 - 2285]. The model has shown exceptional similarities to the viscoelastic behaviour of actin stress fibre contractility. Hence, we decided to use a similar model to represent the active behaviour of the actin cortex. We assume that other components of the actin cortex, ER, and cytoplasm behave in a passive manner, and hence, for simplicity, a linear elastic model for these components is assumed. We thank the reviewer for spotting some ambiguities in our model description, e.g. Eq. (4), and have now fixed this.

6. Necessary information is missing in some graphs. - Fig. 2b and d: Origins and numbers on both axes in the kymographs are missing. Position of nucleus is required to show if the flow of intensity is not due to the shifting of cell body itself.

- Fig. 3b: There are 6 lines in the graph but no labels for each.

- Fig. 5b: no labels

- Fig. 6e-g: what are the dotted lines?

Response: We apologize for missing out on these details and thank the reviewer for pointing them out. We have labelled the graphs in Fig 3b and 5b properly. The graphs to represent velocity of the mutants (Figs. 6e-g) have now been replaced by the data from homogeneously expressing Rtn4 and Climp63 MDCK cells (Fig. 6c). Origin and scale have now been added to the kymographs. Sec61 β labels the nuclear periphery as well along with the whole ER, the position of the nucleus has been specified in the kymographs.

7. Statistics: Please provide sufficient information of statistical analysis such as p-values and replicates.

Response: We thank the reviewer for the suggestion. We have now replaced all graphs with representation of the p-values; statistical analysis and the number of replicates has been mentioned in each figure legend.

Reviewer #2:

Remarks to the Author:

Epithelial gaps of various sizes and geometries appear in all organisms, from single-cell extrusion to centimeter-sized wounds. For closure of gaps, epithelial cells invoke two orthogonal modes: lamellipodial crawling at the convex edge and purse-string-like movements at the concave edge. Mechanisms underlying these differential responses to geometric cues have remained elusive. Here, the authors perform an intracellular cartography to reveal that in both micropatterned and naturally arising gaps, and they suggest that the endoplasmic reticulum (ER) undergoes edge curvature-dependent morphological reorganizations with convex and concave edges promoting ER tubules and sheets, respectively. This reorganization depends on cytoskeleton-generated protrusive and contractile forces. Additionally, theoretical modeling predicts that the curvature-specific ER morphology leads to a lower strain energy state. ER tubules at the convex edge favor perpendicularly oriented focal adhesions, supporting lamellipodial crawling while ER sheets at the concave edge favor parallelly oriented focal adhesions, supporting purse-string-like movements. Altogether, ER emerges as a central player in cellular mechanotransduction, which orchestrates two orthogonal modes of cell migration by integrating signals from cytoskeletal networks.

This study makes a potentially impactful finding regarding the role of curvature-dependent ER morphology changes on wound closure. The topic is quite interesting. However, the identification of peripheral sheets and methods for expansion of peripheral sheet-like structures (and to a lesser extent, ER tubules) appears overly simplistic and is potentially confounded.

Response: We thank the reviewer for the insightful comments. To distinguish ER sheets from dense tubule matrices, we validated Climp63 and p180 as reliable sheet markers and confirmed their enrichment at concave edges via super-resolution imaging and live dynamics. We further supported this with ER-PM contact site data and modeling of alternative ER architectures. To test functional roles, we used Climp63-3E mutants and siRNA knockdown of Climp63 and Rtn4, showing that ER structure, independent of microtubule interaction, instructively regulates migration. Additionally, only Rtn4b among smaller reticulons mimicked Rtn4a in promoting lamellipodia. These results robustly link ER morphology to curvature-dependent migration.

Specific points:

1. Dense ER at the front of migrating cells at the concave edge are described as ER sheets and fenestrated ER sheets. These peripheral structures may also represent dense tubular matrices (see Nixon-Abell et al, Science 2016). How do these different possibilities affect the conclusions of this study (or the mathematical modeling)? In other words, what if these are not sheets at all, but instead dense networks of tubules?

Response: We thank the reviewer for this important and insightful comment. We agree that distinguishing between ER sheets and dense tubular matrices is crucial, especially in the context of dynamic cellular processes such as migration. Our goal was to determine the identity of the dense ER structures observed at the front of migrating cells at concave wound edges, and whether they more closely resemble ER sheets or tubule clusters, as discussed in prior work (e.g., Nixon-Abell et al., Science, 2016). While Climp63 and p180 are widely recognized markers of ER sheets (<https://pubmed.ncbi.nlm.nih.gov/21111237/> and <https://pubmed.ncbi.nlm.nih.gov/31751826/>), we sought to directly benchmark their localization in our system to confirm how faithfully they label ER sheets in contrast to ER tubules. Please see the section “Characterization of dense sheet-like ER at the concave edge” on Page 5-6 of the revised manuscript.

To do this, we transfected MDCK cells with the general ER marker Sec61 β , allowed them to form confluent monolayers, and performed super-resolution SORA imaging after

immunostaining for Climp63 or p180. In non-migrating cells, both Climp63 and p180 localized exclusively to dense, perinuclear ER structures, consistent with their known enrichment in ER sheets (Supplementary Fig. 6a). These markers were absent from the peripheral ER, where tubules predominate, including potential dense tubular networks as described in previous studies. This validation step confirmed that Climp63 and p180 reliably distinguish ER sheets from tubules in our system. Having established this benchmark, we then turned our attention to the leading edge of migrating cells, particularly those at concave wound curvatures. There, we observed strong enrichment of Climp63 and p180 in dense ER structures specifically at the concave edge (Figs. 2a-b), but not at convex edges. Quantitative analysis showed that the fraction of Climp63 and p180 signal over total ER (Sec61 β) was significantly higher at concave than convex fronts (Figs. 2c-d), supporting the conclusion that the dense ER at concave edges is largely composed of sheet-like structures. Furthermore, live-cell imaging of mCherry-Climp63 in cells with low-to-moderate expression revealed anterograde movement of Climp63-positive structures toward the concave front (Fig. 2h), indicative of dynamic ER sheets involved in front-directed processes.

To further investigate the composition of the dense ER structures at the concave edge, we examined the localization of Rtn4b, a tubule-associated curvature-stabilizing protein also known to localize at the edges of ER sheets. Using dual labeling with Sec61 β and Rtn4b followed by super-resolution imaging (*Right most panel*, Supplementary Fig. 6b), we observed that while Rtn4b was most prominently enriched at the rims of dense ER sheets, with only little presence along distinct tubules and little to no signal within the sheet interiors. This preferential localization to sheet boundaries was more pronounced and supports the view that the dense ER at the concave edge is primarily composed of Climp63-positive sheets, with minimal contribution from tubular elements.

Further supporting this interpretation, our new findings on ER-PM contact site distribution provide additional evidence in favor of ER sheets at the concave front. A recent study (<https://pubmed.ncbi.nlm.nih.gov/38867038/>) using ER-PM (plasma membrane) contacts marker MAPPER demonstrated that ER-PM contacts polarize in single migrating cells, with Climp63-rich ER sheets at the rear showing more ER-PM contacts than Rtn4a-rich tubular ER at the front, influencing receptor tyrosine kinase signaling and phosphoinositide distribution. While that study focused on front-rear polarity in single-cell migration, our study addresses curvature-dependent collective migration, where mechanical forces act at the tissue scale. Nonetheless, we observe a consistent pattern: ER-PM contact density (measured using GFP-MAPPER) is elevated at concave edges, which are enriched in ER sheets and reduced at convex edges, where ER tubules dominate (Supplementary Figs. 6c-d). This suggests that curvature-driven ER morphology may regulate ER-PM contact site formation, potentially influencing signaling mechanisms critical for migration. These observations further support the conclusion that Climp63-enriched ER sheets are the dominant dense ER structure at concave wound fronts.

Altogether, multiple lines of evidence, from marker-based immunostaining, super-resolution imaging, and live-cell dynamics to ER-PM contact site distribution, converge to support the conclusion that ER sheets are the predominant component of the dense ER structure at the concave edge. Please note that in the flow of the manuscript, we initially referred to the dense ER as sheet-like, and only designated it as ER sheets after establishing multiple lines of supporting evidence. However, we fully acknowledge the complexity of ER morphology, and the possibility that clusters of dense tubules are also present and may contribute to the observed dynamics. Considering this possible structural heterogeneity, we have incorporated this consideration into our mathematical modeling framework. Specifically, we simulated two alternative configurations at the concave edge: (1) ER sheets and (2) parallel-aligned tubule clusters. Although ER sheets and parallel tubules both have lower energy compared to

perpendicular tubules, making them favorable at the concave edge, the model shows that ER sheets have the lowest energy overall, and thus represent the most energetically favorable configuration under concave, contractile conditions. However, parallel tubule arrays also lower the energy relative to perpendicular tubules, and thus a mixture of structures intermediate between sheets and parallel tubules may also be favored over perpendicular ER tubules, depending on local force balance and protein distribution. At the convex edge, perpendicular ER tubules remain energetically favorable due to protrusive dynamics and lower compressive stress. These results demonstrate that, while ER sheets are energetically dominant at concave zones, the model is robust to the inclusion of dense tubule networks and can accommodate the structural complexity observed experimentally. We are grateful to the reviewer for raising this nuanced and important point. We have revised the manuscript to include this issue in the discussion (please see Page 14, Line 32-45), explicitly referencing the relevant literature (including Nixon-Abell *et al.*, Science 2016) and noting the structural possibilities and modeling considerations that arise from the diversity of ER morphologies.

2. Related to the above, the generation of sheets and tubules via protein overexpression described here may be overly simplistic. For instance, Climp63 binds microtubules and localizes to ER tubules as well, and may not have sheet-generating properties at all (for example, see Zheng *et al.*, Nature 2022). Have the authors tried constructs of Climp63 defective for microtubule interaction? Also, what is the effect of Climp63 knock out? For ER tubule generation, Rtn4 is a particularly large reticulon; are similar findings seen upon overexpression of a smaller one? Also, some knockout/knockdown experiments for reticulons (though admittedly there are several forms), may be helpful.

Response: We thank the reviewer for raising these valuable and insightful points. It is well established that Climp63 overexpression leads to ER sheet expansion in various systems (<https://pubmed.ncbi.nlm.nih.gov/21111237/>, <https://pubmed.ncbi.nlm.nih.gov/36044336/>), though the precise mechanism remains unclear. In our system as well, super-resolution imaging of mCherry-Climp63-expressing cells confirmed expansion of dense ER sheets (Supplementary Fig. 10a). To address the reviewer's concern regarding Climp63's microtubule-binding role, we overexpressed Climp63-3E, a phosphomimetic mutant defective in microtubule interaction (<https://pubmed.ncbi.nlm.nih.gov/15703217/>). Interestingly, Climp63-3E-expressing cells, despite being defective in microtubule binding, still exhibited impaired lamellipodia formation, similar to cells overexpressing wildtype Climp63 (Rebuttal Fig. 5). These cells also showed increased ER sheet accumulation, suggesting that Climp63-induced suppression of lamellipodia at the convex edge is primarily driven by its ability to promote ER sheet formation, rather than its interaction with microtubules. We thank the reviewer for this suggestion, which prompted us to rigorously test the role of microtubule binding in this context.

To further validate the functional roles of ER structure, we conducted siRNA-mediated knockdown of Climp63 and Rtn4 in Eph4 epithelial cells. Knockdown efficiency was confirmed by western blotting (Supplementary Fig. 11c). Upon Rtn4 knockdown, we observed a loss of ER tubules (anti-KDEL staining, Supplementary Fig. 11d) and a corresponding decrease in lamellipodia formation, with increased polarized actin bundles at the wound edge. This was reflected in lower MDE values for actin (Fig. 6e, g), consistent with a shift toward contractile behavior at convex curvatures. Conversely, Climp63 knockdown led to expansion of sheet-like ER (Supplementary Fig. 11e) and formation of actin bundles even at convex edges (Fig. 6f, h), again supporting the idea that ER sheets promote contractile migration modes and inhibit protrusive activity.

As also suggested by the reviewer, we tested whether smaller reticulons have similar effects by overexpressing Rtn4b, Rtn2b, and Rtn3S in MDCK cells. Among these, only Rtn4b

overexpression led to enhanced lamellipodia formation and suppression of actin bundles, including at concave edges (Supplementary Fig. 12), mirroring the phenotype observed with Rtn4a overexpression. All reticulons (Rtn1–4) have been shown to localize on ER tubules and promote membrane curvature generation through a conserved reticulon homology domain (RHD) at the C-terminus, which inserts into the membrane as a hairpin to induce curvature (<https://pubmed.ncbi.nlm.nih.gov/20955502/>). Prior studies have shown that Rtn4a and Rtn4b overexpression shifts ER morphology toward tubules in mammalian cells (<https://pubmed.ncbi.nlm.nih.gov/27786289/>). Our findings further align with this, showing that like Rtn4a, Rtn4b promotes lamellipodia formation at both convex and concave curvatures. Moreover, Rtn4b has previously been implicated in wound healing, where it is required for macrophage migration to the wound site in mouse models (<https://pubmed.ncbi.nlm.nih.gov/19805174/>). Based on these observations and prior literature, we hypothesize that Rtn4a and Rtn4b promote high protrusive activity during migration by generating ER tubules, thereby contributing to curvature-dependent epithelial remodeling.

Collectively, these results show that altering ER morphology using multiple, independent approaches, including overexpression, knockdown, and mutant constructs, consistently affects the mode of epithelial migration. While Climp63's role is complex and may extend beyond sheet generation, the functional outcome across all manipulations supports a model where ER structure, especially at the concave edge, regulates the cytoskeletal state and migration behavior in a curvature-dependent manner. Please see Page 12, Line 41 to Page 13, Line 20.

Rebuttal Figure 5. *Climp63-3E mutant migration. MDCK cells transfected with Climp63-3E migrating at convex (upper panel) and concave (lower panel) edges.*

Reviewer #3:

Remarks to the Author:

The manuscript authored by Sieran Rawal et al., titled "Curvature-dependent morphological reorganization of the endoplasmic reticulum determines the mode of epithelial migration," investigates the curvature-dependent morphological reorganization of the endoplasmic reticulum (ER) at convex and concave edges.

The study presents intriguing observations utilizing a potentially valuable system for the field. While the study offers original insights into curvature-dependent ER morphologies, a notable limitation is the lack of clear definition and characterization of in-plane (concave and convex) curvatures. In addition, curvatures are sometimes studied at the single scale or at the multicellular scale.

Response: We thank the reviewer for raising this point. As mentioned by the reviewer, accurately quantifying curvature at both single-cell and tissue levels is crucial for understanding how geometric constraints influence cellular behavior, ensuring that observed migration patterns and ER organization are directly linked to curvature-driven effects rather than secondary remodeling events. To this end, we calculated both single-cell curvature and tissue-level curvature to assess how individual cells respond to geometric constraints at the wound edge. The single-cell curvature (k_{cell}) was determined by fitting a circle to the leading edge of each cell and computing its curvature as $k_{cell} = (R_{cell})^{-1}$, where R_{cell} is the radius of the fitted circle. Similarly, the tissue-level curvature (k_{tissue}) was derived by fitting a smooth curvature-spline to the overall wound boundary and computing $k_{tissue} = (R_{tissue})^{-1}$ where R_{tissue} represents the radius of the wound edge at a given point. To account for variability in how much of a cell is directly interacting with the wound edge, we computed the Fraction of Cell Perimeter Exposed to the Wound Edge (P_{exp}) as $P_{exp} = (L_{edge}/L_{total}) \times 100$, where L_{edge} is the length of the cell perimeter in direct contact with the wound and L_{total} is the total perimeter of the cell. The extent of a cell's perimeter exposed to the wound edge is a critical parameter in curvature analysis, as it determines whether the cell is truly experiencing and responding to the local geometric cue. Without sufficient edge exposure, cellular behavior is more likely to be influenced by neighboring cells than by curvature itself, leading to confounding effects in the analysis. In this respect, at first, plotting ER tubule fractions as functions of P_{exp} , we observed that ER tubule fraction decreased with increasing P_{exp} in the concave region and increased with increasing P_{exp} in the convex region (Supplementary Figs. 2a-b). Next, plotting Climp63 fractions as functions of P_{exp} , we observed that Climp63 fraction increased with increasing P_{exp} in the concave region and decreased with increasing P_{exp} in the convex region (Supplementary Figs. 2c-d). These results suggested that the extent of a cell's exposure to the wound edge (P_{exp}) modulates its ability to sense and respond to curvature, with higher exposure reinforcing curvature-specific ER morphologies - tubules at convex and sheets at concave edges. These analyses together underscore the importance of direct geometric input in driving intracellular reorganization.

Next to find a correlation between k_{tissue} and k_{cell} , we plotted them against each other at both curvatures. We observed that the k_{tissue} and k_{cell} had direct correlation for most values at both curvatures with some off data points (Supplementary Figs. 2e-f). We then checked whether these off-data points represented cells with low curvature exposure. To this end, when we included only those cells with $P_{exp} > 20\%$, most of the off-data points got removed (Supplementary Figs. 2g-h). In the cleaned dataset, cell and tissue curvatures emerged to show a stronger one-to-one correspondence (Supplementary Figs. 2g-h) than before. Taking this analysis into consideration, for subsequent analyses of our experimental data, we included only those cells with $P_{exp} > 20\%$ to remove the ambiguity in interpreting cell behavior and establish a robust basis for analyzing curvature-dependent responses. This approach ensured that our curvature measurements accurately reflected the behavior of cells at the migration front,

providing a robust basis for analyzing curvature-dependent cellular responses. It also confirmed that individual cells sense and respond to the large-scale geometric constraints imposed by the wound edge if they have a certain threshold exposure to the wound edge. We have included this detailed discussion on curvature estimation in the revised Supplemental Material (*section B, Page 6*).

Relevantly, both our study and that of Chen *et al.* (2019) (<https://www.nature.com/articles/s41567-018-0383-6>) explore how epithelial cells respond to curvature cues, highlighting the importance of curvature sensing in guiding migration. Chen *et al.* used adhesive stamps and micropatterns to generate subcellular and multicellular curvatures, demonstrating representative responses across a range of radii (8–100 μm), and emphasized how cells distinguish between positive and negative curvature. Similarly, we investigated curvature sensing across scales by combining micropatterned gaps with naturally occurring convex and concave curvatures in sub-confluent colonies. While Chen *et al.* focused on qualitative distinctions, we extended this by quantitatively analysing both tissue-level and single-cell curvature and mapping the graded cellular response within a defined curvature range (0.005–0.1 μm^{-1}). Instead of adhesive stamps, we used spontaneous curvature in colony edges, capturing physiological contexts. Together, both approaches reveal complementary aspects of how epithelial cells sense and respond to curvature across spatial scales.

1. The authors utilized both micropatterned wounds of defined geometry in cultured epithelial monolayers and natural wounds with spontaneously arising convex and concave curved regions in mouse embryonic epidermis. Although the manuscript mentions the use of polydimethylsiloxane (PDMS) stencils of various shapes, the specific geometries of these stencils are not clearly presented. It is essential for the authors to provide images of the micropatterns obtained from the negative stencils and offer a time-lapse sequence of a few typical tissues at low magnification during the healing process

Response: We thank the reviewer for this important suggestion. In response, we have now included low-magnification time-lapse images of MDCK cells migrating into micropatterned gaps generated using PDMS stencils with defined convex and concave geometries, as shown in Figure 1a. Additionally, we have added time-lapse imaging of MDCK monolayers closing flower-shaped wounds to illustrate collective migration in more complex geometries (Supplementary Figs. 1a-b, supplementary videos 1 and 2). These experiments show that cells close the gaps within approximately 6–8 hours, demonstrating the robustness and reproducibility of the micropatterned wound system. While our study primarily focuses on early stages of migration (within 30 minutes) to capture the initial cellular response to curvature, we hope these additional datasets provide a clearer visualization of the stencil design, wound geometry, and overall healing dynamics.

2. This work is very closed to the work by Tianchi Chen *et al.* in Nature Physics 2019 "Large-scale curvature sensing by directional actin flow drives cellular migration mode switching" (<https://europepmc.org/backend/ptpmcrender.fcgi?accid=PMC6456019&blobtype=pdf>) and the flower pattern could be very useful here to better control and standardize experiments on positive and negative curvature.

Response: We thank the reviewer for highlighting the relevant work by Tianchi Chen *et al.* (Nature Physics, 2019). As suggested, we have also employed flower-shaped wound patterns to better standardize and control regions of positive and negative curvature. Consistent with our main findings, we observed actin bundling at concave edges and lamellipodial protrusions at convex edges, along with corresponding changes in ER morphology - tubular ER enrichment at convex edges and dense sheet-like ER at concave edges (Supplementary Fig. 3e). These

results further validate our conclusions and demonstrate that our observations are robust across different geometric configurations, including the well-established flower pattern system.

3. The authors claimed that lamellipodial protrusions or actin bundles emerge within 15-30 minutes. This assertion should be substantiated with immunostained experiments of micropatterned tissues at various time points ($t=0$, $t=10$, $t=20$, and $t=30$ min) or using live actin imaging in mouse embryonic epidermal cells. Additionally, the fraction of ER tubules in Fig. 1d and the fraction of Climp63 in 1e should be presented as a function of curvature for both convex and concave patterns.

Response: We thank the reviewer for this constructive comment. To directly support our claim that lamellipodial protrusions and actin bundles emerge within 15–30 minutes, we performed time-lapse imaging in both micropatterned epithelial monolayers and embryonic mouse epidermal tissues. In LifeAct-MDCK cells seeded around PDMS micropatterns, we observed the formation of actin bundles at concave edges and lamellipodial protrusions at convex edges within 30 minutes (Supplementary Fig. 1c, supplementary videos 3 and 4). Similarly, in mouse embryonic skin explants, we labelled the tissue with SiR-actin, introduced controlled incision wounds, and conducted live imaging. We observed accumulation of actin at concave wound edges and emergence of small protrusive structures at convex edges within the same 30-minute window (Supplementary Fig. 5a). These results confirm that the curvature-dependent cytoskeletal responses we report are established within the early phase of migration.

In response to the reviewer's second suggestion, we have now included quantitative plots showing the fraction of ER tubules and Climp63 intensity at the cell front as a function of curvature (Figs. 1d and h). These data reveal that both ER tubule fraction and Climp63 intensity exhibit a clear curvature-dependent response at convex wound edges: the ER tubule fraction increases, while Climp63 staining decreases with increasing positive curvature. In contrast, at concave edges, both parameters appear relatively insensitive to curvature magnitude, showing little variation across the range of negative curvatures. Importantly, when comparing convex and concave zones overall, we observe marked differences in average ER tubule and Climp63 levels, consistent with distinct ER organization and cytoskeletal states between the two curvature regimes.

4. While Fig. 2 illustrates a cell-scale curvature-dependent mechanism mainly related to lamellipodial shape, Fig. 3 appears to depict curvature using a few cells. The authors could consider using adhesive micropatterns to impose specific curvatures (concave vs. convex) on individual mouse embryonic epidermal cells while controlling the spreading area, which could enhance the support for Fig. 2.

Response: We thank the reviewer for this thoughtful suggestion. Our study focuses on how epithelial cells respond to large-scale curvature, spanning single to multiple cells, during collective migration. While Figure 2 (of the previous version of the manuscript) specifically highlighted ER dynamics within a single cell, this was done to clearly visualize the anterograde and retrograde flows of ER at concave and convex edges, respectively, which required high-resolution, zoomed-in imaging. In contrast, the rest of the figures in the manuscript demonstrate that curvature-dependent responses, particularly ER morphology and cytoskeletal organization, are consistently observed across all cells at the migration front, whether the curvature spans one or several neighboring cells (Rebuttal Fig. 6). Thus, while subcellular resolution was necessary to capture ER dynamics in Figure 2, our overall focus remains on the cellular to multicellular scale of curvature, which better reflects physiological curvature sensing along wound boundaries in both micropatterned *in vitro* models and embryonic epidermal tissues.

As the reviewer rightly highlights, accurately quantifying curvature is essential to ensure that the observed migration patterns and ER organization are truly curvature-driven,

rather than artifacts of local tissue remodeling or cell-cell variability. To this end, we developed a robust framework to quantify both single-cell curvature and tissue-level curvature. Single-cell curvature (k_{cell}) was determined by fitting a circle to the leading edge of each cell and computing $k_{cell} = (R_{cell})^{-1}$. Similarly, tissue-level curvature (k_{tissue}) was calculated by fitting a smooth spline to the overall wound edge and using $k_{tissue} = (R_{tissue})^{-1}$. To account for how much of the cell directly interacts with the wound edge, we also calculated the Fraction of Cell Perimeter Exposed to the Wound Edge (P_{exp}) as $(L_{edge}/L_{total}) \times 100$, and included only those cells with $P_{exp} > 20\%$ to focus on true curvature-sensing cells, excluding those primarily constrained by neighboring cells. This comprehensive approach revealed a strong one-to-one correspondence between single-cell and tissue-level curvature (Supplementary Fig 2), confirming that individual cells faithfully sense and respond to the larger-scale geometric constraints of the wound edge. Importantly, as shown in Figures 1d and h, we observed significant differences in ER morphology as a function of curvature sign, and these differences held consistently across a range of geometries. Thus, while adhesive micropatterns imposing curvature on isolated cells may be a valuable approach in the future, our current system captures physiologically relevant curvature sensing in collective contexts and provides quantitative, geometry-linked insight into how ER organization and cytoskeletal state are coordinated during epithelial migration. Importantly, while ER morphology shows a graded response with curvature magnitude, we consistently observe robust tubule-to-sheet transitions at all scales of convex and concave curvatures, provided that single-cell and tissue-level curvatures align (i.e. $k_{cell} = k_{tissue}$).

Rebuttal Figure 6. Large scale curvature depicted by MDCK cells exposed to different scales of curvature (increasing from left to right).

5. The scale of curvatures in this study is challenging to appreciate and appears to be close to the lamellipodial size. The authors should provide a robust quantification of convex and concave curvature as well as their evolution over time. Demonstrating the independence of results from curvature size, in addition to curvature sign, is crucial. Dynamic micropatterning could aid in showing the evolution of tubule fraction concerning dynamic changes in lamellipodial curvature.

Response: We thank the reviewer for this thoughtful comment regarding curvature scale. In our study, the curvatures explored typically span tens of microns, corresponding to 1–4 cell lengths - a regime we refer to as finite-scale curvature, which is highly relevant in epithelial tissues where cells encounter geometric constraints during collective migration. To further test

whether our findings depend on the artificiality of imposed geometries or the scale of curvature, we analyzed cells at the periphery of finite-sized epithelial colonies, where curvature emerges spontaneously without external patterning. These naturally occurring convex and concave curvatures reflect the physiological range experienced by epithelial cells. Relevantly, our micropatterned curvatures exhibit similar spatial scales, enabling direct comparison between controlled and native settings. In this setup, MDCK cells expressing mApple-Sec61 β were allowed to form small colonies, and actin organization was visualized using phalloidin staining. We observed that cells along convex regions formed lamellipodial protrusions and showed enrichment of ER tubules, whereas cells at concave zones exhibited actin bundles and dense ER sheets (Supplementary Fig. 3f). Quantification of ER tubule fraction as a function of curvature confirmed a significantly higher tubule content at convex edges compared to concave ones (Supplementary Fig. 3g), reinforcing the idea that ER organization is shaped by local curvature cues. Please see Page 4, Line 40 – Page 5, Line 7.

In addition to these experiments, as mentioned above, we performed a systematic analysis of ER morphology (ER tubule fraction and Climp63 intensity) as a function of both curvature sign and magnitude (Figs. 1d and h). This analysis revealed that at convex edges, the ER tubule fraction increases and Climp63 intensity decreases with increasing positive curvature, indicating a graded response. In contrast, at concave edges, both parameters remain relatively independent of curvature magnitude. As stated above, we have ensured our analysis captures cells genuinely influenced by curvature. Together, these findings demonstrate that our observed curvature-dependent ER reorganization is not limited to stencil-defined geometries or specific scales, but rather reflects a robust and generalizable epithelial response to local 2D curvature cues. The consistency of this behavior across both patterned and spontaneously emerging curvatures highlights that this is an intrinsic cellular mechanism, tightly linked to geometric sensing and cytoskeletal state during collective migration.

One of the challenges in analyzing curvature-dependent cell behavior is that curvature evolves dynamically during migration, as cells remodel the wound edge over time. Since our interest was to look at the cell's initial response to curvature, we calculated how the curvature evolves over a period of 30 mins (i.e. our time of interest). As observed in Rebuttal Fig. 7 the magnitude of tissue curvature did not change significantly in 30 mins.

Rebuttal Figure 7. Evolution of concave (upper panel) and convex (lower panel) curvature with time as seen in LifeAct MDCK cells, LifeAct-GFP (green)

6. The fraction of ER tubules in Fig. 3d appears to be significantly lower on concave edges (red) than on convex edges (green). It is imperative to present a plot with associated statistical tests to confirm this observation.

Response: We thank the reviewer for this important observation. In response, we have now included a quantitative plot with statistical analysis to compare the fraction of ER tubules at convex and concave edges under different drug treatment conditions (Fig. 3d). As also shown in Figure 1, the fraction of ER tubules is significantly lower at concave edges compared to convex edges under control conditions, reflecting the predominance of ER sheet morphology at concave zones. Upon treatment with CK666, which inhibits Arp2/3-mediated branched actin polymerization, cells form contractile actin bundles at both convex and concave edges. This contractile state promotes sheet-like ER morphology across both curvature types, leading to a marked reduction in ER tubule content at the front, as shown in the quantification (Fig. 3d). In contrast, Blebbistatin treatment, which inhibits actomyosin contractility, induces protrusive activity at both curvatures. This shift promotes ER tubule formation even at concave edges, resulting in a higher tubule fraction under Blebbistatin compared to CK666 treatment. These quantifications reinforce our core finding that ER morphology is tightly coupled to local cytoskeletal states, which are in turn influenced by curvature and force balance and confirm that the differences in tubule content across curvature types are both statistically and biologically significant.

Also, please see the response to Comment 1 of Reviewer 1 for the mechanistic analysis that we have included in the revised manuscript. Briefly, our results demonstrate that curvature-driven forces shape ER morphology, which in turn plays an active, instructive role in organizing cytoskeletal architecture and determining migration mode. Through combined genetic and pharmacological perturbations, we show that ER sheets suppress lamellipodia even under conditions favoring protrusion, while ER tubules disrupt contractile actin bundles, indicating a mechanistic tug-of-war between ER structure and cytoskeletal dynamics that ultimately governs curvature-specific epithelial migration.

7. If there is indeed a lower fraction of ER tubules on concave edges compared to convex edges (as noted in #5), the authors should reflect this difference in their modeling presented in Fig. 3. If the model does not currently account for this discrepancy, it should be adapted accordingly.

Response: The main aim of the model is to explain the mechanism regulating the curvature and force-dependent preference of ER morphology distribution in a cell. Accordingly, we assumed that a given cell could have only one type of ER morphology and considered cells with either sheet, parallel or perpendicular tubular ER morphologies. By comparing the strain energy density of the cells, we predicted which ER morphology is preferred for a given curvature. That is, our model predicted that we expect to see a higher fraction of ER tubules in convex curvature and a higher fraction of ER sheets in concave curvature because the combination leads to lowered strain energy density. To achieve such a prediction and have a fair comparison, we initialised the model with the same volume of ER, irrespective of the morphology, for a given curvature. Initialising a cell geometry by explicitly adding more tubules in the convex case and less in the concave case would introduce a bias in the model. This could imply that the model's output will always prefer the initial configuration that was imposed, irrespective of the curvature and their corresponding forces. Hence, we assume equal volume of ER morphology and allow the model to predict which ER morphology is preferred for a given curvature.

8. The modeling of strain energy minimization in Fig. 4 was conducted within a curvature range of 0-0.2 μm^{-1} , which seems unsupported by experimental data on concave and convex curvature. This discrepancy undermines the credibility of the results and must be addressed.

Response: The curvatures of the cell geometry used in the mathematical model was decided based on the values used in in vitro experiments. Both in experiments and modelling, the curvatures varied from 0-0.2 μm^{-1} (Figs. 4 e,g,h, and j). Further, we have added a section in the supplementary material to describe the evaluation of curvature of the cell geometry used in the mathematical model (Supplementary Fig. 8f).

9. The authors mentioned the high axial stiffness of the ER resulting in lower cell deformation and thereby reducing strain energy density. Clarification is needed regarding what is meant by "high axial stiffness" and the specific values of stiffness or bending modulus used in the simulations.

Response: The term "high axial stiffness" is borrowed from structural mechanics, and we thank the reviewer for highlighting the need to explain better how we used it. We have added Supplementary Figures 8a-b to explain the forces acting on ER. In the case of protrusive forces, the axial forces is simply the one due to the protrusion, and the axial stiffness is the corresponding resistance to this force. Hence, to avoid the confusion, we have rephrased "axial stiffness" with "resist protruding forces". We have also added a table in the supplementary with the stiffness values used in the model along with literature references.

10. Given the observed correlation between ER organization and microtubules (Figs. 5 and 6), it is important to inquire whether the presence of microtubules was implemented in the simulations presented in Fig. 4. If not, this omission should be addressed and justified.

Response: Since the main aim of the model was to explain the mechanics behind the curvature and force-dependent ER morphology distribution, we did not model microtubules explicitly. We found that even without any coupling to the microtubules, our model can explain the experimentally observed curvature-dependent organisation of the ER. However, our model can explain the response of passive mechanical structures to protrusive and contractile forces.

Rebuttal Figure 8. (a) Normalised strain energy density variation with curvature during protrusion. b) Normalised strain energy density variation with curvature during contraction.

By simulating microtubules (MT) instead of ER and following the strain energy density minimisation hypothesis, we find that the strain energy density of tubular or perpendicular MT is lower than that of parallel MT during protrusion (Rebuttal Fig. 8a). Further, the strain energy

density of parallel MT is lower than that of tubular MT in case of contraction (Rebuttal Fig 8b). In addition, we also see that the response of parallel and tubular MT with varying curvatures is similar to that observed in ER (Figs. 4e and h). This supports the experimentally observed correlation between ER and microtubules organisation (Figs. 5 and 6). Note that the Young's modulus of microtubules is $1.2e9$ Pa [Gittes et al, *J Cell Biol* 1993; 120 (4): 923–934].], while that of ER is $19e6$ Pa [Georgiades, P. et al, *Sci Rep*, 2017, 7,16474].]. The explicit modelling of very stiff microtubules, without a detailed complex molecular level modelling of their interaction with the ER would result in a model where the overall mechanical state is dominated by the microtubules. However, since our major novel contribution is to explain the organisation of the ER and the resulting effect on collective cell migration, and because a full coupled ER-microtubule model would depend on intricate molecular links between the two that are still not fully known, we decided to only model the ER. We have mentioned this assumption in the revised manuscript (Please see Page 9, Lines 7-10).

11. In Figs. 6 and 7 (and throughout the manuscript), it is recommended to present all data in box plots with individual data points superimposed, rather than using bars alone.

Response: We thank the reviewer for this suggestion. We have now represented all plots with individual data points superimposed and indicated the p-values. Statistical analysis and number of replicates have been added to the Figure legends.

12. The methodology for focal adhesion orientation analysis, particularly in Figs. 7c, lacks clarity. The authors should provide detailed information about their analysis pipeline, including higher magnification images, the results of their thresholding process, and clear demonstrations of orientation at the single adhesion scale.

Response: We have used directionality plugin in Fiji to quantify the orientation of focal adhesions. This plugin is used to quantify the preferred orientation of the structures in the input image. We crop the image of the front of the cell, labelled with anti-paxillin antibody, clearly marking focal adhesions and provide this as the input image to the plugin. We use the Fourier components method, 6 Nbins from 0-90 degrees. It computes and generate a histogram representing the frequency of structures (focal adhesions) at a particular orientation. The data over many cells at different curvatures is collected and averaged to finally plot the focal adhesion orientation as in Fig. 7c. The pipeline of the analysis is demonstrated in the Rebuttal Fig. 9. The details of this analysis have now been added to the methods section.

Rebuttal Figure 9. Directionality analysis Pipeline

13. Is there a clear relationship between focal adhesions orientation, and directional actin flow such as observed by Tianchi Chen et al. in Nature Physics 2019 "Large-scale curvature sensing by directional actin flow drives cellular migration mode switching" (<https://europepmc.org/backend/ptpmcrender.fcgi?accid=PMC6456019&blobtype=pdf>) ?

Additional experiments on actin treadmilling could be very informative.

Response: We thank the reviewer for this interesting suggestion. In our unpublished results, we have observed some differential dynamics of YFP-Paxillin in MDCK cells at convex and concave curvatures, which could be correlated to the directional actin flow as suggested by the reviewer. However, further studies will be required to confirm this hypothesis which is out of scope of the current study. Here, we demonstrate and emphasize on the role of ER morphology changes at the front of cells migrating at different curvatures and how these changes are ultimately important for deciding the mode of cell migration. We do observe retrograde and anterograde flow of ER at convex and concave edges respectively (Figs. 2e-g) which could be correlated to the differential actin flow at the two curvatures as reported in Chen *et al.* 2019.

Rebuttal Letter

Reviewer #1 (Remarks to the Author):

I appreciate the authors' comprehensive responses to all my previous comments. Their efforts to address each point are respectable, and the manuscript has improved overall.

I would like to highlight one remaining issue regarding figure presentation. Specifically, the scale bars in most figures are too small to be clearly visible, which may hinder proper interpretation of the data. Additionally, some panels are missing scale bars altogether (e.g., Figures 2h and 2i). The authors should revise the figure panels to ensure all scale bars are clearly legible and consistently presented across the figures.

Once this issue is addressed, I believe the manuscript will be suitable for publication.

Reviewer #1 (Remarks on code availability):

A detailed assessment was not possible because it requires Abaqus, which I do not have. The README file provides sufficient instructions.

Response: We thank the reviewer for the valuable feedback and contribution to the manuscript and for pointing out the error in consistent presentation of the scale bars in the figures. We have now added visible and clear scale bars in all images, including Figs. 2h and 2i.

Reviewer #2 (Remarks to the Author):

In this revised manuscript, the authors have performed numerous additional experiments and analyses. Each of the points that I raised in my original review has been addressed.

Response: We thank the reviewer for important suggestions, which helped make the manuscript more thorough and comprehensive.